# SCALING-UP DIVERSE ORTHOGONAL CONVOLUTIONAL NETWORKS BY A PARAUNITARY FRAMEWORK

## ABSTRACT

Enforcing orthogonality in convolutional neural networks is an antidote for gradient vanishing/exploding problems, sensitivity to perturbation, and bounding generalization errors. Many previous approaches for orthogonal convolutions enforce orthogonality on its flattened kernel, which, however, do not lead to the orthogonality of the operation. Some recent approaches consider orthogonality for standard convolutional layers and propose specific classes of their realizations. In this work, we propose a theoretical framework that establishes the equivalence between diverse orthogonal convolutional layers in the spatial domain and the paraunitary systems in the spectral domain. Since 1D paraunitary systems admit a complete factorization, we can parameterize any separable orthogonal convolution as a composition of spatial filters. As a result, our framework endows high expressive power to various convolutional layers while maintaining their exact orthogonality. Furthermore, our layers are memory and computationally efficient for deep networks compared to previous designs. Our versatile framework, for the first time, enables the study of architectural designs for deep orthogonal networks, such as choices of skip connection, initialization, stride, and dilation. Consequently, we scale up orthogonal networks to deep architectures, including ResNet and ShuffleNet, substantially outperforming their shallower counterparts. Finally, we show how to construct residual flows, a flow-based generative model that requires strict Lipschitzness, using our orthogonal networks.

## 1 INTRODUCTION

Convolutional neural networks (CNNs), whose deployment has witnessed extensive empirical success, still exhibit a range of limitations that are not thoroughly studied. Firstly, deep convolutional networks are in general difficult to learn, and their high performance heavily relies on techniques that are not fully understood, such as skip-connections (He et al., 2016a), batch normalization (Ioffe & Szegedy, 2015), specialized initialization (Glorot & Bengio, 2010). Secondly, they are notoriously sensitive to imperceptible perturbations, including adversarial attacks (Goodfellow et al., 2014) or geometric transformations (Azulay & Weiss, 2019). Finally, a precise characterization of their generalization property is still under active investigation (Neyshabur et al., 2018; Jia et al., 2019).

Orthogonal networks, which have a "flat" spectrum with all singular values of the layer being 1 (each layer's output norm $\|\boldsymbol{y}\|$ equals its input norm $\|\boldsymbol{x}\|$, $\forall \boldsymbol{x}$), alleviate all problems above. As shown in recent works, by enforcing orthogonality in neural nets, we obtain *(1) easier optimization* (Zhang et al., 2018a; Qi et al., 2020): since each orthogonal layer preserves the gradient norm during backpropagation, an orthogonal network is free from gradient vanishing/ exploding problems; *(2) robustness against adversarial perturbation* (Anil et al., 2019; Li et al., 2019b; Trockman & Kolter, 2021): since each orthogonal layer is 1-Lipschitz, an orthogonal network can not amplify any perturbation to the input to flip the output prediction; *(3) better generalizability* as proved in (Jia et al., 2019): a network's generalization error is positively related to the standard deviation of each linear layer's singular values, thus encouraging orthogonality in the network lowers its generalization error.

Despite the benefits, enforcing orthogonality in convolutional networks is challenging. To avoid the strict constraint, orthogonal initialization (dynamical isometry) (Pennington et al., 2017; Xiao et al., 2018) and orthogonal regularization (Wang et al., 2019; Qi et al., 2020) are adopted to address the gradient vanishing/exploding problems. However, these methods are not suitable for applications that

require strict Lipschitzness, such as adversarial robustness (Anil et al., 2019) and residual flows (Chen et al., 2019), as they do not enforce strict orthogonality (and Lipschitzness) during training.

Our goal is to enforce exact orthogonality in state-of-the-art convolutional networks without expensive computations. We identify three main challenges in doing so. **Challenge I: Achieving exact orthogonality throughout the training process.** Prior works such as *orthogonal regularization* (Wang et al., 2019) and *reshaped kernel orthogonality* (Jia et al., 2017; Cisse et al., 2017), while enjoying algorithmic simplicity, do not meet the requirement of exact orthogonality. Note that enforcing orthogonality during training is necessary as a post-training orthogonalization of the weights can ruin the performance. **Challenge II: Avoiding expensive computations.** An efficient algorithm is crucial for scalability to large networks. Existing work based on projected gradient descent (Sedghi et al., 2019), however, requires expensive projection after each update. For instance, the projection step in Sedghi et al. (2019) computes an SVD and flattens the spectrum to enforce orthogonality, which costs $O(\text{size}(\text{feature}) \cdot \text{channels}^3)$ for a convolutional layer. **Challenge III: Scaling-up to state-of-the-art deep convolutional networks.** There are many variants to the standard convolutional layer, including dilated, strided, group convolutions, which are essential for state-of-the-art deep convolutional networks. However, none of the existing methods proposes mechanisms to orthogonalize these variants. The lack of techniques, as a result, limits the broad applications of orthogonal convolutional layers to state-of-the-art deeper convolutional networks.

We resolve **challenges I, II, and III** by proposing *complete parameterizations* for *orthogonal 1D-convolutions* and *separable orthogonal 2D-convolutions*. First, using the convolution theorem (Oppenheim et al., 1996) (convolution in the spatial domain is equivalent to multiplication in the spectral domain), we reduce the problem of designing orthogonal convolutions to constructing unitary matrices for all frequencies, i.e., a paraunitary system (Vaidyanathan, 1993). Therefore, we obtain a parameterization for *all* separable orthogonal 2D-convolutions, guaranteeing exact orthogonality and high expressive power. Note that there is no previous approach that achieves exact orthogonality (up to machine precision). For the first time, our versatile framework enables the study in the designs of deep orthogonal networks, such as skip connection, initialization, stride, and dilation. Consequently, we scale orthogonal networks to deep architectures, including ResNet and ShuffleNet, substantially outperforming their shallower counterparts. Our proposed method serves as gaining insight into the trade-off between orthogonality and expressiveness. We observe that exact orthogonality is essential in these deeper architectures with more than 10 layers (detailed discussion of exact orthogonality in Appendix E.3). Finally, we show how to deploy our orthogonal networks in Residual Flow (Chen et al., 2019), a flow-based generative model that requires strict Lipschitzness (Appendix F).

**Summary of Contributions:**
1. We establish the equivalence between orthogonal convolutions in the spatial domain and paraunitary systems in the spectral domain. Consequently, we can interpret existing approaches for orthogonal convolutions as implicit designs of paraunitary systems.
2. Based on a complete factorization of 1D paraunitary systems, we propose the first exact and complete design for orthogonal 1D-convolutions as well as separable orthogonal 2D-convolutions, ensuring exact orthogonality and high expressive power. None of the existing works achieve a complete parameterization of all orthogonal 2D-convolutions.
3. We prove that orthogonality for various convolutional layers (strided, dilated, group) are also entirely characterized by paraunitary systems. Consequently, our design easily extends to these variants, ensuring both completeness and exactness of the orthogonal convolutions.
4. We study the design considerations for orthogonal networks (choices of skip connection, initialization, depth, width, and kernel size), and show that orthogonal networks can scale to deep architectures including ResNet, WideResNet, and ShuffleNet.

## 2 ORTHOGONAL CONVOLUTIONS VIA PARAUNITARY SYSTEMS

Designing orthogonal convolutional layer $\{\boldsymbol{h}_{t,s} : \boldsymbol{y}_t = \boldsymbol{h}_{t,s} * \boldsymbol{x}_s\}_{t=1,s=1}^{T,S}$ ($s, t$ index input/output channels) in the spatial domain is challenging. When a convolution layer is written as a matrix-vector product, where the matrix is block-circulant and its $(t, s)^{\text{th}}$ takes a circulant form $\text{Cir}(\boldsymbol{h}_{t,s})$ as:

$$\text{Cir}(\boldsymbol{h}_{t,s}) = \begin{bmatrix} h_{t,s}[1] & h_{t,s}[N] & \cdots & h_{t,s}[2] \\ h_{t,s}[2] & h_{t,s}[1] & h_{t,s}[N] & \cdots \\ \vdots & \ddots & \ddots & \vdots \\ h_{t,s}[N] & \cdots & h_{t,s}[2] & h_{t,s}[1] \end{bmatrix} \in \mathbb{R}^{N \times N}. \tag{2.1}$$

Therefore, the layer is orthogonal if the block-circulant matrix $[\mathsf{Cir}\,(\boldsymbol{h}_{t,s})]_{t=1,s=1}^{T,S}$ is orthogonal. However, it is not obvious how to enforce orthogonality in a block-circulant matrix.

## 2.1 Achieving Orthogonal Convolutions by Paraunitary Systems

We propose a novel design of orthogonal convolutions from a spectral domain perspective, motivated by the *convolution theorem* (Theorem 2.1). For simplicity, we group the entries at the same locations into a vector/matrix, e.g., we denote $\{x_s[n]\}_{s=1}^S$ as $\boldsymbol{x}[n] \in \mathbb{R}^S$ and $\{h_{t,s}[n]\}_{t=1,s=1}^{T,S}$ as $\boldsymbol{h}[n] \in \mathbb{R}^{T \times S}$.

**Theorem 2.1** (Convolution theorem (Oppenheim et al., 1996))**.** *For a standard convolution layer* $\boldsymbol{h}$*:* $\boldsymbol{y}[i] = \sum_n \boldsymbol{h}[n]\boldsymbol{x}[i-n]$*, the convolution in the spatial domain is equivalent to a matrix-vector multiplication in the spectral domain, i.e.,* $\boldsymbol{Y}(z) = \boldsymbol{H}(z)\boldsymbol{X}(z), \forall z \in \mathbb{C}$*. Here,* $\boldsymbol{X}(z) = \sum_{n=0}^{N-1} \boldsymbol{x}[n]z^{-n}$*,* $\boldsymbol{Y}(z) = \sum_{n=0}^{N-1} \boldsymbol{y}[n]z^{-n}$*,* $\boldsymbol{H}(z) = \sum_{n=-\underline{L}}^{\overline{L}} \boldsymbol{h}[n]z^{-n}$ *denote the z-transforms of input, output, kernel respectively, where* $N$ *is the length of* $\boldsymbol{x}, \boldsymbol{y}$ *and* $[-\underline{L}, \overline{L}]$ *is the span of the filter* $\boldsymbol{h}$*.*

The convolution theorem states that a standard convolution layer is a matrix-vector multiplication in the spectral domain. As long as the *transfer matrix* $\boldsymbol{H}(z)$ is unitary at $z = e^{j\omega}$ for all frequencies $\forall \omega \in \mathbb{R}$ (j is the imaginary unit), the convolutional layer $\boldsymbol{h}$ is orthogonal.

Therefore, as a major novelty of our paper, we design orthogonal convolutions through designing unitary transfer matrix $\boldsymbol{H}(e^{j\omega})$ at all frequencies $\omega \in \mathbb{R}$, which is known as a *paraunitary systems* (Vaidyanathan, 1993; Strang & Nguyen, 1996). We prove in Theorem B.5 that a convolutional layer is orthogonal in the spatial domain if and only if it is paraunitary in the spectral domain.

**Benefits through paraunitary systems. (1)** The spectral representation simplifies the designs of orthogonal convolutions and avoids the analysis of block-circulant structure. **(2)** Since paraunitary systems are necessary and sufficient for orthogonal convolutions, it is *impossible* to find an orthogonal convolution whose transfer matrix is *not* paraunitary. **(3)** There exists a complete factorization of paraunitary systems: any paraunitary $\boldsymbol{H}(z)$ can be realized through multiplications in the spectral domain, as will show in Equation (2.2a). **(4)** Since multiplications in spectral domain correspond to convolutions in spatial domain, *any* orthogonal convolution can be realized as cascaded convolutions of multiple sub-layers, each parameterized by an orthogonal matrix. **(5)** There are mature methods that represent orthogonal matrices via unconstrained parameters. Consequently, we can learn orthogonal convolutions using standard optimizers on a model parameterized via our design.

**Interpretation of existing methods.** Since paraunitary system is a necessary and sufficient condition for orthogonal convolution, all existing approaches, including *singular value clipping and masking (SVCM)* (Sedghi et al., 2019), *block convolution orthogonal parameterization (BCOP)* (Li et al., 2019b), *Cayley Convolution (CayleyConv)* (Trockman & Kolter, 2021), *skew orthogonal convolution (SOC)* design paraunitary systems implicitly. We discuss these interpretations in Appendix C.3.3, and show that our SC-Fac has the lowest computational complexity among all approaches.

## 2.2 Realizing Paraunitary Systems via Re-parameterization

After reducing the problem of orthogonal convolutions to paraunitary systems, we are left with the question of how to realize paraunitary systems. We use a complete factorization form of paraunitary systems to realize any paraunitary systems. According to Theorem C.7, any paraunitary system $\boldsymbol{H}(z)$ can be written as the form in Equation (2.2a) and any $\boldsymbol{H}(z)$ in this form is a paraunitary system:

$$\boldsymbol{H}(z) = \boldsymbol{V}(z; \boldsymbol{U}^{(-\underline{L})}) \cdots \boldsymbol{V}(z; \boldsymbol{U}^{(-1)})\boldsymbol{Q}\boldsymbol{V}(z^{-1}; \boldsymbol{U}^{(1)}) \cdots \boldsymbol{V}(z^{-1}; \boldsymbol{U}^{(\overline{L})}), \tag{2.2a}$$

$$\text{where } \boldsymbol{V}(z; \boldsymbol{U}^{(\ell)}) = (\boldsymbol{I} - \boldsymbol{U}^{(\ell)}\boldsymbol{U}^{(\ell)\top}) + \boldsymbol{U}^{(\ell)}\boldsymbol{U}^{(\ell)\top}z, \ \forall \ell \in \{-\underline{L}, \cdots, -1\} \cup \{1, \cdots, \overline{L}\}. \tag{2.2b}$$

Here $\boldsymbol{Q}$ is an orthogonal matrix and each $\boldsymbol{U}^{(\ell)}$ is a column-orthogonal matrix whose number of columns is sampled uniformly from $\{1, \cdots, T\}$. As multiplications in the spectral domain are equivalent to convolutions in the spatial domain, this complete spectral factorization of paraunitary systems in Equation (2.2a) allows us to parameterize *any* orthogonal convolution in the spatial domain as cascaded convolutions of $\boldsymbol{V}(z; \boldsymbol{U}^{(\ell)})$'s spatial counterparts and the orthogonal matrix $\boldsymbol{Q}$.

**Model design in the spatial domain.** We obtain a *complete design of orthogonal 1D-convolutions:* using *learnable (column)-orthogonal matrices* $(\{\boldsymbol{U}^{(\ell)}\}_{\ell=-\underline{L}}^{-1}, \boldsymbol{Q}, \{\boldsymbol{U}^{(\ell)}\}_{\ell=1}^{\overline{L}})$, we parameterize a size $(\underline{L} + \overline{L} + 1)$ convolution as cascaded convolutions of the following filters in the spatial domain

$$\left\{\left[\boldsymbol{I} - \boldsymbol{U}^{(\ell)}\boldsymbol{U}^{(\ell)\top}, \ \boldsymbol{U}^{(\ell)}\boldsymbol{U}^{(\ell)\top}\right]\right\}_{\ell=-\underline{L}}^{-1}, \ \boldsymbol{Q}, \ \left\{\left[\boldsymbol{U}^{(\ell)}\boldsymbol{U}^{(\ell)\top}, \boldsymbol{I} - \boldsymbol{U}^{(\ell)}\boldsymbol{U}^{(\ell)\top}\right]\right\}_{\ell=1}^{\overline{L}}. \tag{2.3}$$

Figure 1 provides a visualization of our proposed design of orthogonal convolution layers; each block denotes a convolution and the form of the filter is displayed in each of the block. In practice, we compose all $(\underline{L} + \overline{L} + 1)$ filters into one for orthogonal convolution, which not only increases the computational parallelism but also avoids storing intermediate outputs between filters.

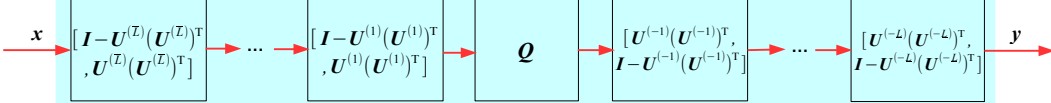

Figure 1: Complete design of orthogonal convolutional layer as a cascade of convolutions, whose filter coefficients are depicted in each block. The matrix $Q$ is orthogonal and $U^{(\ell)}$'s are column-orthogonal.

Using this 1D-convolution, we obtain a complete design for *separable orthogonal 2D-convolutions*, which can be represented as a convolution of two orthogonal 1D-convolutional layers. With separability, we obtain a design of orthogonal 2D-convolutions by composing two complete designs of orthogonal 1D-convolutions. As a result, we can parameterize separable orthogonal 2D-convolution with filter size $(\overline{L}_1 + \underline{L}_1 + 1) \times (\overline{L}_2 + \underline{L}_2 + 1)$ as a convolution of two orthogonal 1D-convolutions with learnable (column-)orthogonal matrices $(\{U_1^{(\ell)}\}_{\ell=-\underline{L}_1}^{-1}, Q_1, \{U_1^{(\ell)}\}_{\ell=1}^{\overline{L}_1})$ and $(\{U_2^{(\ell)}\}_{\ell=-\underline{L}_2}^{-1}, Q_2, \{U_2^{(\ell)}\}_{\ell=1}^{\overline{L}_2})$. With a complete factorization of paraunitary systems (1D and separable 2D), we reduce the problem of designing orthogonal convolutions to the one for orthogonal matrices.

**Parameterization for orthogonal matrices.** In Appendix C.4, we perform a comparative study on different parameterizations of orthogonal matrices, including the *Björck orthogonalization* (Anil et al., 2019; Li et al., 2019b), the *Cayley transform* (Helfrich et al., 2018; Maduranga et al., 2019), and the *exponential map* (Lezcano-Casado & Martínez-Rubio, 2019). In our implementation, we adopt a modified exponential map due to its efficiency, exactness, and completeness. The exponential map is a *surjective* mapping from a skew-symmetry matrix $A$ to a special orthogonal matrix $U$ (i.e., $\det(U) = 1$) with $U = \exp(A) = I + A + A^2/2 + \cdots$, where the infinite series is computed up to machine-precision (Higham, 2009). To parameterize all orthogonal matrices, we introduce an additional orthogonal matrix $Q$ in $U = Q \exp(A)$, where $Q$ is (randomly) generated at initialization and fixed during training (Lezcano Casado, 2019) .

Now we have an end-to-end pipeline for orthogonal convolutions as shown in Figure 2 (See Algorithm 1 in Appendix E for a pseudo implementation). Since our method relies on separability and complete factorization for 1D paraunitary systems, we call it *Separable Complete Factorization (SC-Fac)*.

| Ortho. Conv. $h[n]$ | $H(z)=\sum h[n]z^{-n}$ | Paraunitary $H(z)$ | $H(z)=f(Q,\{U^{(l)}\})$ | Ortho. Factors $Q,\{U^{(l)}\}$ | $Q=\exp(A)$ $U^{(l)}=\exp(A^{(l)})$ | Model Params. $A,\{A^{(l)}\}$ |
|---|---|---|---|---|---|---|

Figure 2: **SC-Fac: A pipeline for designing orthogonal convolutional layer. (1)** An orthogonal convolution $h[n]$ is equivalent a paraunitary system $H(z)$ in the spectral domain (Theorem 2.1). **(2)** The paraunitary system $H(z)$ is multiplications of factors characterized by (column-)orthogonal matrices $(\{U^{(\ell)}\}_{\ell=-\underline{L}}^{-1}, Q, \{U^{(\ell)}\}_{\ell=1}^{\overline{L}})$ (Equation (2.2a), Theorem B.5). **(3)** These orthogonal matrices are parameterized by skew-symmetric matrices using exponential map.

## 3 UNIFYING ORTHOGONAL CONVOLUTIONS VARIANTS

Various convolutional layers (strided, dilated, and group convolution) are widely used in neural networks. However, it is not apparent how to enforce their orthogonality, as the convolution theorem (Theorem 2.1) only holds for *standard* convolutions. Previous approaches only deal with standard convolutions (Sedghi et al., 2019; Li et al., 2019b; Trockman & Kolter, 2021), thus orthogonality for state-of-the-art architectures has never been studied before.

We address this limitation by modifying convolution theorem for each variant of convolution layer, which allows us to design these variants using paraunitary systems.

**Theorem 3.1** (Convolution and paraunitary theorems for various convolutions). *Strided, dilated, and group convolutions can be unified in the spectral domain as* $\underline{Y}(z) = \underline{H}(z)\underline{X}(z)$, *where* $\underline{Y}(z)$, $\underline{H}(z)$, $\underline{X}(z)$ *are modified Z-transforms of* $y$, $h$, $x$. *We instantiate* $\underline{H}(z)$ *for strided convolutions in Proposition C.4, dilated convolution in Proposition C.5, and group convolution in Proposition C.6. Furthermore, a convolution is orthogonal if and only if* $\underline{H}(z)$ *is paraunitary.*

In Table 1, we formulate strided, dilated, and group convolutions in the spatial domain, interpreting them as up-sampled or down-sampled variants of a standard convolution. Now, we introduce the concept of up-sampling and down-sampling precisely below.

Given a sequence $\boldsymbol{x}$, we introduce its *up-sampled sequence* $\boldsymbol{x}^{\uparrow R}$ with sampling rate $R$ as $\boldsymbol{x}^{\uparrow R}[n] \triangleq \boldsymbol{x}[n/R]$ for $n \equiv 0 \pmod{R}$. On the other hand, its *(r,R)-polyphase component* $\boldsymbol{x}^{r|R}$ indicates the $r$-th down-sampled sequence with sampling rate $R$, defined

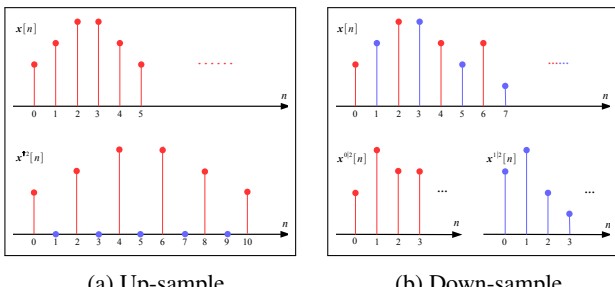

(a) Up-sample      (b) Down-sample

Figure 3: **Up and down sampling.** In **(a)**, the sequence $\boldsymbol{x}[n]$ is up-sampled into $\boldsymbol{x}^{\uparrow 2}[n]$. In **(b)**, $\boldsymbol{x}[n]$ is down-sampled into $\boldsymbol{x}^{0|2}[n]$ with even entries (red) and $\boldsymbol{x}^{1|2}[n]$ with odd entries(blue).

as $\boldsymbol{x}^{r|R}[n] \triangleq \boldsymbol{x}[nR + r]$. We illustrated an example of $\boldsymbol{x}^{\uparrow R}$ and $\boldsymbol{x}^{r|R}$ in Figure 3 when sampling rate $R = 2$. The Z-transforms of $\boldsymbol{x}^{\uparrow R}, \boldsymbol{x}^{r|R}$ are denoted as $\boldsymbol{X}^{\uparrow R}(z)$, $\boldsymbol{X}^{r|R}(z)$ respectively. Their relations to $\boldsymbol{X}(z)$ are studied in Appendix C.1.

Table 1: **Variants of convolutions.** We present the modified $Z$-transforms, $\underline{\boldsymbol{Y}}(z)$, $\underline{\boldsymbol{H}}(z)$, and $\underline{\boldsymbol{X}}(z)$ for each convolution such that $\underline{\boldsymbol{Y}}(z) = \underline{\boldsymbol{H}}(z)\underline{\boldsymbol{X}}(z)$ holds. In the table, $\boldsymbol{X}^{[R]}(z) \triangleq [\boldsymbol{X}^{0|R}(z)^{\top}, \dots, \boldsymbol{X}^{R-1|R}(z)^{\top}]^{\top}$ and $\widetilde{\boldsymbol{X}}^{[R]}(z) = [\boldsymbol{X}^{-0|R}(z), \dots, \boldsymbol{X}^{-(R-1)|R}(z)]$. For group convolution, $\boldsymbol{h}^g$ is the filter for the $g^{\text{th}}$ group with $\boldsymbol{H}^g(z)$ being its Z-transform, and $\mathsf{blkdiag}(\cdot)$ stacks multiple matrices into a block-diagonal matrix.

| Convolution Type | Spatial Representation | Spectral Representation | | |
| --- | --- | --- | --- | --- |
| | | $\underline{\boldsymbol{Y}}(z)$ | $\underline{\boldsymbol{H}}(z)$ | $\underline{\boldsymbol{X}}(z)$ |
| Standard | $\boldsymbol{y}[i] = \sum_{n \in \mathbb{Z}} \boldsymbol{h}[n]\boldsymbol{x}[i - n]$ | $\boldsymbol{Y}(z)$ | $\boldsymbol{H}(z)$ | $\boldsymbol{X}(z)$ |
| $R$-Dilated | $\boldsymbol{y}[i] = \sum_{n \in \mathbb{Z}} \boldsymbol{h}^{\uparrow R}[n]\boldsymbol{x}[i - n]$ | $\boldsymbol{Y}(z)$ | $\boldsymbol{H}(z^R)$ | $\boldsymbol{X}(z)$ |
| $\downarrow R$-Strided | $\boldsymbol{y}[i] = \sum_{n \in \mathbb{Z}} \boldsymbol{h}[n]\boldsymbol{x}[Ri - n]$ | $\boldsymbol{Y}(z)$ | $\widetilde{\boldsymbol{H}}^{[R]}(z)$ | $\boldsymbol{X}^{[R]}(z)$ |
| $\uparrow R$-Strided | $\boldsymbol{y}[i] = \sum_{n \in \mathbb{Z}} \boldsymbol{h}[n]\boldsymbol{x}^{\uparrow R}[i - n]$ | $\boldsymbol{Y}^{[R]}(z)$ | $\boldsymbol{H}^{[R]}(z)$ | $\boldsymbol{X}(z)$ |
| $G$-Group | $\boldsymbol{y}[i] = \sum_{n \in \mathbb{Z}} \mathsf{blkdiag}(\{\boldsymbol{h}^g[n]\})\boldsymbol{x}[i - n]$ | $\boldsymbol{Y}(z)$ | $\mathsf{blkdiag}(\{\boldsymbol{H}^g(z)\})$ | $\boldsymbol{X}(z)$ |

Now we are ready to interpret convolution variants. *(1) Strided convolution* is used to adjust the feature resolution: a strided convolution ($\downarrow R$-strided) decreases the resolution by down-sampling after a standard convolution, while a transposed strided convolutional layer ($\uparrow R$-strided) increases the resolution by up-sampling before a standard convolution. *(2) Dilated convolution* increases the receptive field of a convolution without extra parameters: an $R$-dilated convolution up-samples its filters before convolution with the input. *(3) Group convolution* reduces the parameters and computations, thus widely used by efficient architectures: a $G$-group convolution divides the input/output channels into $G$ groups and restricts the connections within each group. In Appendix C.2, we prove that a convolution is orthogonal if and only if its modified Z-transform $\underline{\boldsymbol{H}}(z)$ is paraunitary.

## 4   LEARNING DEEP ORTHOGONAL NETWORKS WITH LIPSCHITZ BOUNDS

In this section, we switch our focus from layer design to network design. In particular, we aim to study how to scale-up deep orthogonal networks with Lipschitz bounds.

**Lipschitz networks** (Anil et al., 2019; Li et al., 2019b; Trockman & Kolter, 2021), whose Lipschitz bounds are imposed by their architectures, are proposed as competitive candidates to guarantee robustness in deep learning. A Lipschitz network consists of *orthogonal layers* and *GroupSort activations* (Anil et al., 2019) — both are 1-Lipschitz and gradient norm preserving. See Appendix D for more discussions on the properties of GroupSort and Lipschitz networks. Given a Lipschitz constant $L$, a Lipschitz network $f$ can compute a certified radius for each input from its output margin. Formally, denote the output margin of an input $\boldsymbol{x}$ with label $c$ as

$$\mathcal{M}_f(x) \triangleq \max(0, f(\boldsymbol{x})_c - \max_{i \neq c} f(\boldsymbol{x})_i), \tag{4.1}$$

i.e., the difference between the correct logit and the second largest logit. Then the output is robust to perturbation such that $f(\boldsymbol{x} + \boldsymbol{\epsilon}) = f(\boldsymbol{x}) = c, \forall \epsilon : \|\boldsymbol{\epsilon}\| < \mathcal{M}_f(\boldsymbol{x})/\sqrt{2}L$.

Despite the benefit, existing architectures for Lipschitz networks remain simple and shallow, and a Lipschitz network is typically an interleaving cascade of orthogonal layers and GroupSort activations (Li et al., 2019b). More advanced architectures, such as ResNet and ShuffleNet, are still out of reach. While orthogonal layers supposedly substitute the role of *batch normalization* (Pennington et al., 2017; Xiao et al., 2018; Qi et al., 2020), other critical factors, including *skip-connections* (He et al., 2016a;b) and *proper initialization* (Glorot & Bengio, 2010) are lacking. In this section, we explore skip-connections and initialization methods toward addressing this problem.

**Skip-connections.** Two general types of skip-connections are widely used in deep networks, one based on *addition* and another on *concatenation*. The addition-based connection is proposed in ResNet (He et al., 2016a), and adopted in SE-Net (Hu et al., 2018) and EffcientNet (Tan & Le, 2019). The concatenation-based connection is proposed in flow-based generative models (Dinh et al., 2014; 2016; Kingma & Dhariwal, 2018), and adopted in DenseNet (Huang et al., 2017) and ShuffleNet (Zhang et al., 2018b; Ma et al., 2018). In what follows, we propose Lipschitz skip-connections with these two mechanisms, illustrated in Figure 6 (in Appendix D).

**Proposition 4.1** (Lipschitzness of residual blocks). *Suppose $f^1$, $f^2$ are L-Lipschitz (for residual/shortcut branches) and $\alpha \in [0, 1]$ is a learnable scalar, then an additive residual block $f : f(\boldsymbol{x}) \triangleq \alpha f^1(\boldsymbol{x}) + (1 - \alpha) f^2(\boldsymbol{x})$ is L-Lipschitz. Alternatively, suppose $g^1$, $g^2$ are L-Lipschitz and $\boldsymbol{P}$ is a channel permutation, then a concatenative residual block $g : g(\boldsymbol{x}) \triangleq \boldsymbol{P} \left[ g^1(\boldsymbol{x}^1); g^2(\boldsymbol{x}^2) \right]$ is L-Lipschitz, where $[\cdot; \cdot]$ denotes channel concatenation, and $\boldsymbol{x}$ is split into $\boldsymbol{x}_1$ and $\boldsymbol{x}_2$, i.e., $\boldsymbol{x} = [\boldsymbol{x}_1, \boldsymbol{x}_2]$.*

**Initialization.** Proper initialization is crucial in training deep networks (Glorot & Bengio, 2010; He et al., 2016a). Various methods are proposed to initialize orthogonal matrices, including the identical/permutation and torus initialization in the context of orthogonal RNNs (Henaff et al., 2016; Helfrich et al., 2018; Lezcano-Casado & Martínez-Rubio, 2019). However, initialization of orthogonal convolutions was not systematically studied, and all previous approaches inherit the initialization from the underlying parameterization (Li et al., 2019b; Trockman & Kolter, 2021). In Proposition D.1, we study the condition when a paraunitary system (represented in Equation (2.2a)) reduces to an orthogonal matrix. This reduction allows us to apply the initialization methods for orthogonal matrices (e.g., uniform, torus) to orthogonal convolutions.

In the experiments, we will evaluate the impact of different choices of skip-connections and initialization methods to the performance of deep Lipschitz networks.

## 5 RELATED WORK

**Dynamical isometry** (Pennington et al., 2017; 2018; Chen et al., 2018; Pennington et al., 2018) aims to address the gradient vanishing/exploding problems in deep vanilla networks with *orthogonal initialization*. These works focus on understanding the interplay between initialization and various nonlinear activations. However, these approaches do not guarantee Lipschitzness after training, thus are unsuitable for applications that require strict characterization of Lipschitz constants, such as adversarial robustness (Anil et al., 2019) and residual flows (Behrmann et al., 2019).

**Learning orthogonality** has three typical approaches: *regularization*, *parameterization* (representing the feasible set with unconstrained parameters), and *projected gradient descent* (PGD) / *Riemannian gradient descent* (RGD). While regularization is approximate, the latter two learn exact orthogonality. **(1)** For **orthogonal matrices**, various regularizations are proposed in Xie et al. (2017); Bansal et al. (2018). Alternatively, numerous parameterizations exist, including *Householder reflections* (Mhammedi et al., 2017), *Given rotations* (Dorobantu et al., 2016), *Cayley transform* (Helfrich et al., 2018), *matrix exponential* (Lezcano-Casado & Martínez-Rubio, 2019), and *algorithmic unrolling* (Anil et al., 2019; Huang et al., 2020). Lastly, Jia et al. (2017) propose PGD via *singular value clipping*, and Vorontsov et al. (2017); Li et al. (2019a) consider RGD. **(2)** For **orthogonal convolutions**, some existing works learn orthogonality for the *flattened matrix* (Jia et al., 2017; Cisse et al., 2017; Bansal et al., 2018) or each *output channel* (Liu et al., 2021). However, these methods do not lead to orthogonality (norm preserving) of the operation. Sedghi et al. (2019) propose to use PGD via *singular value clipping and masking*, which is expensive and can result in inexact orthogonality. Recent works adopt parameterizations, using *block convolutions* (Li et al., 2019b), *Cayley transform* (Trockman & Kolter, 2021), or *convolution exponential* (Singla & Feizi, 2021). Note that *network deconvolution* (Ye et al., 2020) aims to whiten the activations (i.e., orthogonalize their distribution), but the added whitening operations do not necessarily preserve orthogonality.

**Paraunitary systems** are extensively studied in filter banks and wavelets (Vaidyanathan, 1993; Strang & Nguyen, 1996; Lin & Vaidyanathan, 1996). Classic theory shows that 1D-paraunitary systems are fully characterized by a spectral factorization (see Chapter 14 of Vaidyanathan (1993)), but not all MD-paraunitary systems admit a factorized form (see Chapter 8 of Lin & Vaidyanathan (1996)). While the complete characterization of MD-paraunitary systems is known in theory (which incurs a difficult problem of solving a system of nonlinear equations) (Venkataraman & Levy, 1995; Zhou, 2005), most practical constructions use separable paraunitary systems (Lin & Vaidyanathan, 1996) and special classes of non-separable paraunitary systems (Hurley & Hurley, 2012). The equivalence between orthogonal convolutions and paraunitary systems thus opens the opportunities to apply these classic theories in designing orthogonal convolutions.

## 6 EXPERIMENTS

In the experiments, we achieve the following goals. **(1)** We demonstrate in Section 6.1 that our separable complete factorization (SC-Fac) achieves precise orthogonality (up to machine-precision), resulting in more accurate orthogonal designs than previous ones (Sedghi et al., 2019; Li et al., 2019b; Trockman & Kolter, 2021). **(2)** Despite the differences in preciseness, we show in Section 6.2 that different realizations of paraunitary systems only have a minor impact on the adversarial robustness of Lipschitz networks. **(3)** Due to the versatility of our convolutional layers and architectures, in Section 6.3, we explore the best strategy to scale Lipschitz networks to wider/deeper architectures. **(4)** In Appendix F, we further demonstrate in a successful application of orthogonal convolutions in residual flows (Chen et al., 2019). Training details are provided in Appendix E.1.

### 6.1 EXACT ORTHOGONALITY

We evaluate the orthogonality of our SC-Fac layer verse previous approaches, including Cayley-Conv (Trockman & Kolter, 2021), BCOP (Li et al., 2019b), SVCM (Sedghi et al., 2019), RKO (Cisse et al., 2017), OSSN (Miyato et al., 2018). Our experiments are based on a convolutional layer with $64$ input channels and $16 \times 16$ input size. We orthogonalize the layer using each approach, and evaluate it with Gaussian inputs. For our SC-Fac layer, We initialize all orthogonal matrices uniformly, while we use built-in initialization for others. We evaluate the difference between $1$ and the ratio of the output norm to the input norm — a layer is exactly orthogonal if the number is close to $0$.

Table 2: **(Left) Orthogonality evaluation of different designs for standard convolution.** The number $\|\text{Conv}(x)\|/\|x\| - 1$ indicates the difference between the output and input norms of a layer. A layer is more precisely orthogonal if the number is closer to 0. As shown, our SC-Fac achieves orders of magnitude more orthogonal on standard convolution. **(Right) Orthogonality evaluation of our SC-Fac design for various convolutions.** The numbers $\|\text{Conv}(x)\|/\|x\| - 1$ displayed are in the magnitude of $10^{-8}$. As shown, our SC-Fac achieves machine epsilon orthogonality on variants of convolution.

| Conv. | $\|\text{Conv}(x)\|/\|x\| - 1$ |
|---|---|
| **SC-Fac** | $(\mathbf{+3.14 \pm 7.38}) \times \mathbf{10^{-8}}$ |
| CayleyConv | $(+2.88 \pm 1.90) \times 10^{-4}$ |
| BCOP | $(+2.59 \pm 6.14) \times 10^{-3}$ |
| SVCM | $-0.429 \pm 3.31 \times 10^{-3}$ |
| RKO | $-0.666 \pm 1.74 \times 10^{-3}$ |
| OSSN | $-0.422 \pm 3.44 \times 10^{-3}$ |

| Type | Groups | 1 | 4 | 16 |
|---|---|---|---|---|
| $R$-Dilated | 1 | $+3.14 \pm 7.38$ | $+1.94 \pm 6.87$ | $+1.44 \pm 6.29$ |
| | 2 | $+3.65 \pm 7.87$ | $+1.41 \pm 6.77$ | $+1.02 \pm 6.46$ |
| | 4 | $+3.18 \pm 7.46$ | $+1.79 \pm 6.87$ | $+1.54 \pm 6.21$ |
| $\downarrow R$-Strided | 2 | $-4.69 \pm 5.10$ | $+4.38 \pm 6.30$ | $+1.79 \pm 5.78$ |
| | 4 | $+10.39 \pm 5.15$ | $+6.35 \pm 6.04$ | $+3.05 \pm 5.79$ |
| $\uparrow R$-Strided | 2 | $+3.67 \pm 7.96$ | $+1.38 \pm 6.70$ | $+1.43 \pm 6.23$ |
| | 4 | $+3.86 \pm 7.09$ | $+1.12 \pm 6.81$ | N/A |

**(1) Standard convolution.** We show in Table 2 (Left) that our SC-Fac is orders of magnitude more precise than all other approaches. The SC-Fac layer is in fact exactly orthogonal up to machine epsilon, which is $2^{-24} \approx 5.96 \times 10^{-8}$ for 32-bits floats. While RKO and OSSN are known not to be orthogonal, we surprisingly find that SVCM is far from orthogonal due to its masking step. **(2) Convolutions variants.** In Section 3, we show that various orthogonal convolutions can be constructed using paraunitary systems. We verify our theory in Table 2 (Right): our SC-Fac layers are exactly orthogonal (up to machine precision) for various convolutions.

### 6.2 ADVERSARIAL ROBUSTNESS

In this subsection, we evaluate the adversarial robustness of Lipschitz networks. Following the setup in (Trockman & Kolter, 2021), we adopt KW-Large, ResNet9, WideResNet10-10 as the backbone architectures, and evaluate their robust accuracy on CIFAR-10 with different designs of orthogonal

convolutions. We extensively perform a hyper-parameter search and choose the best hyper-parameters for each approach based on the robust accuracy. The details of the hyper-parameter search is in Appendix E. We run each model with 5 different seeds and report the best accuracy.

**(1) Certified robustness.** Following Li et al. (2019b), we use the raw images (without normalization) for network input to achieve the best certified accuracy. As shown in Table 3 (Top), different realizations of paraunitary systems, SC-Fac, CayleyConv and BCOP achieve comparable performance — CayleyConv performs $< 1\%$ better in clean accuracy, but the difference in robust accuracy are negligible. **(2) Practical robustness.** Trockman & Kolter (2021) shows that the certified accuracy is too conservative, and it is possible to increase the practical robustness (against PGD attacks) with a standard input normalization. Notice that the normalization increases the Lipschitz bound, thus lower the certified accuracy. Our experiments in Table 3 (Bottom) are based on ResNet9, WideResNet10-10 (Trockman & Kolter, 2021) and a deeper WideResNet22. For the shallow architectures (ResNet9, WideResNet10-10), our SC-Fac, CayleyConv, and BCOP again achieve comparable performance — CayleyConv is slightly ahead in robust accuracy. **For the deeper architecture, our SC-Fac has a clear advantage in both clean and robustness accuracy**, and the clean accuracy to only $5\%$ lower than a traditional ResNet 32 trained with batch normalization. Surprisingly, we find that RKO also performs well in robust accuracy while not exactly orthogonal. In summary, our experiments show that various paraunitary realizations provide different impacts on certified and practical robustness. While exact orthogonality provides tight Lipschitz bound, there is a trade-off between the exact orthogonality and the practical robustness (especially with the shallow architectures).

Table 3: **(Top) Certified robustness for plain convolutional networks** (without input normalization). We use KW-Large introduced by Wong et al. (2018). The results for RKO, OSSN, and SVCM are produced by Trockman & Kolter (2021). **(Bottom) Practical robustness for residual networks** (with input normalization). For 22 layers, the width of SC-Fac is multiplied with 10, CayleyConv with 6, and BCOP and RKO with 8. We are unable to scale CayleyConv, BCOP, and RKO due to memory constraint. As shown, deeper architectures perform better than shallow ones for all orthogonal convolution types, and our SC-Fac has a clear advantage.

| | | | KW-Large | | | | | |
|---|---|---|---|---|---|---|---|---|
| $\epsilon$ | Test Acc. | SC-Fac | CayleyConv | BCOP | RKO | OSSN | SVCM |
| 0 | Clean | 74.69 | 75.57 | 74.81 | 74.47 | 71.69 | 72.43 |
| $\frac{36}{255}$ | Certified | 58.68 | 59.03 | 58.83 | 57.50 | 55.71 | 52.11 |
| | PGD | 67.72 | 67.78 | 67.47 | 68.32 | 65.13 | 66.43 |

| | | ResNet9 | | | | WideResNet10-10 | | | | WideResNet22-max | | | |
|---|---|---|---|---|---|---|---|---|---|---|---|---|---|
| $\epsilon$ | Test Acc. | SC-Fac | CayleyConv | BCOP | RKO | SC-Fac | CayleyConv | BCOP | RKO | SC-Fac | CayleyConv | BCOP | RKO |
| 0 | Clean | 82.19 | **84.26** | 83.20 | 84.07 | 84.09 | 82.99 | 84.29 | **84.51** | **87.82** | 85.85 | 84.50 | 84.55 |
| $\frac{36}{255}$ | PGD | 71.21 | 73.47 | 73.05 | **75.03** | 74.29 | 76.02 | 74.60 | **77.14** | **76.46** | 74.81 | 75.00 | 76.41 |

## 6.3 Scaling-up Deep Orthogonal Networks with Lipschitz Bounds

All previously proposed Lipschitz networks (Li et al., 2019b; Trockman & Kolter, 2021) consider only shallow architectures ($\leq 10$ layers). In this part, we investigate various factors to scale Lipschitz networks to deeper architectures: skip-connection, depth-width, receptive field, and down-sampling.

**(1) Skip-connections.** Conventional wisdom suggests that skip-connections mainly address the gradient vanishing/exploding problem; thus, they are not needed for orthogonal networks. To understand their role, we perform an experiment that trains deep Lipschitz networks without skip-connection and with skip-connections based on addition/concatenation (see Section 4). As shown in Table 4 (left), the network with additive skip-connection substantially outperforms the other two, and the one without skip-connections performs the worst. Therefore, we empirically show that (additive) skip-connection is crucial in deep Lipschitz networks. **(2) Depth and width.** Exact orthogonality is criticized for harming the expressive power of neural networks. We show that the decrease of expressive power can be alleviated by increasing the network depth/width. In Table 3 (Bottom) and Table 7 (Appendix E), we observe that deeper/wider architectures increase both the clean and robust accuracy. **(3) Initialization methods.** We try different initialization methods, including identical, permutation, uniform, and torus (Henaff et al., 2016; Helfrich et al., 2018). We find that identical initialization works the best for deep Lipschitz networks ($> 10$ layers), while all methods perform similarly in shallow networks as shown in Table 6 (Appendix E). **(4) Receptive field and down-sampling.** Previous works (Li et al., 2019b; Trockman & Kolter, 2021) use larger kernel

Table 4: **(Left) Comparisons of various skip connection types** on WideResNet22-10 (kernel size 5). **(Right) Comparisons of various receptive field and down-sampling types** on WideResNet10-10. The symbols ✓, ✗ indicate whether average pooling or strided convolution is used for down-sampling. For "slim" in strided convolution, we set kernel_size = stride; and for for "wide", kernel_size = stride * kernel_size' (where kernel_size' is the kernel size for the main branch.

| Skip type | Test Acc. | |
| --- | --- | --- |
| | Clean | PGD |
| ConvNet (w/o skip) | 69.59 | 59.22 |
| ShuffleNet (concat) | 75.21 | 66.00 |
| ResNet (add) | **87.82** | **76.46** |

| Receptive Field | | Down-Sampling | | Test Acc. | |
| --- | --- | --- | --- | --- | --- |
| Kernel | Dilation | Pool | Stride | Clean | PGD |
| 3 | 1 | ✗ | slim | 80.70 | 68.81 |
| 3 | 1 | ✗ | wide | 82.36 | 70.36 |
| 3 | 1 | ✓ | ✗ | 84.54 | 71.71 |
| 3 | 2 | ✓ | ✗ | 81.53 | 70.07 |
| 5 | 1 | ✓ | ✗ | **84.09** | **74.29** |
| 5 | 2 | ✓ | ✗ | 81.28 | 70.58 |

size and no stride for Lipschitz networks. In Table 4 (Right), we perform a study on the effects of kernel/dilation size and down-sampling types for the orthogonal convolutions. We find that an average pooling as down-sampling consistently outperforms strided convolutions. Furthermore, a larger kernel size helps to boost the performance. **(5) Run-time and memory comparison.** We find that previously proposed orthogonal convolutions such as CayleyConv, BCOP, and RKO require more GPU memory and computation time than SC-Fac. Therefore, we could not to scale them due to memory constraints (for 22 and 32 layers using Tesla V100 32G). In order to scale up Lipschitz networks, economical implementation of orthogonal convolution is crucial. As shown in Figure 4, for deep and wide architectures, our SC-Fac is the most computationally and memory efficient method and the only method that scales to a width increase of 10 on WideResNet22. Missing numbers in Figure 4 and Table 7 (Appendix E) are due to the large memory requirement.

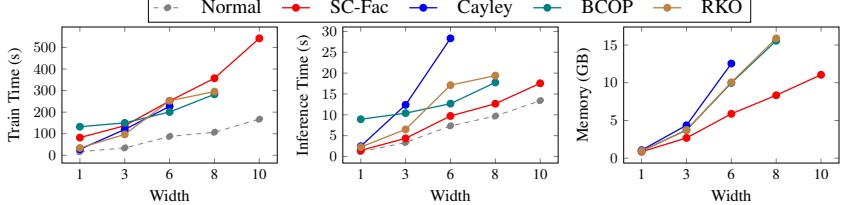

Figure 4: **Run-time and memory comparison** using WideResNet22 on Tesla V100 32G. x-axis indicates the width factor (channels = base_channels × factor). Our SC-Fac is the most computationally and memory-efficient for wide architectures and is the only method that scales to width factor to 10 on WideResNet22. We also compare with an ordinary network with regular convolutions and ReLU activations. Note that SC-Fac has the same inference speed as a regular convolution — the overhead is from the GroupSort activations.

In summary, additive skip-connections are still essential for learning deep orthogonal networks. Due to the orthogonal constraints, it is helpful to increase the depth/width of the network. However, this significantly increases the memory requirement; thus, a cheap implementation (like SC-Fac) is desirable. Finally, we find that a larger kernel size and down-sampling based on average pooling is helpful, unlike standard practices in deep networks.

# 7 CONCLUSION

In this paper, we present a paraunitary framework for orthogonal convolutions. Specifically, we establish the equivalence between orthogonal convolutions in the spatial domain and paraunitary systems in the spectral domain. Therefore, any design for orthogonal convolutions is implicitly constructing paraunitary systems. We further show that the orthogonality for variants of convolution (strided, dilated, and group convolutions) is also fully characterized by paraunitary systems. In summary, paraunitary systems are all we need to ensure orthogonality for diverse types of convolutions.

Based on the complete factorization of 1D paraunitary systems, we develop the first exact and complete design of separable orthogonal 2D-convolutions. Our versatile design allows us to study the design principles for orthogonal convolutional networks. Consequently, we scale orthogonal networks to deeper architectures, substantially outperforming their shallower counterparts. In our experiments, we observe that exact orthogonality plays a crucial role in learning deep Lipschitz networks. In the future, we plan to investigate other use cases that exact orthogonality is essential.

## ETHICS STATEMENT

Our work lies in the foundational research on neural information processing. Specifically, we establish the equivalence between orthogonal convolutions in neural networks and paraunitary systems in signal processing. Furthermore, our presented orthogonal convolutional layers are plug-and-play modules that can replace various convolutional layers in neural networks. Consequently, our modules are applicable in Lipschitz networks for adversarial robustness, recurrent networks for learning long-term dependency, or flow-based networks for effortless reversibility.

The vulnerability of neural networks raises concerns about their deployment in security-sensitive scenarios, such as healthcare systems or self-driving cars. In our experiment, we demonstrate a successful application of orthogonal convolutions in learning robust networks. Furthermore, these networks achieve higher robust accuracy without additional techniques such as adversarial training or randomized smoothing. Therefore, our research contributes to the robust learning of neural networks and potentially leads to their broader deployment.

Furthermore, we reduce the inference cost of our layer to be the same as a traditional convolution layer, significantly lowering the expense for deployment compared with other methods for orthogonal networks. However, our layers are memory and computationally more expensive during training than traditional layers. The overhead to the already expensive training cost exacerbates the concerns on the efficacy of learning neural networks. Therefore, balancing between robustness and efficiency is an important research topic that requires more research in the future. We develop more efficient implementation than previous approaches, narrowing the gap between these two conflicting goals.

## REPRODUCIBILITY STATEMENT

We provide detailed proofs to our theoretical claims in Appendices B to D. We describe experimental setups and training details of our models in Appendix E. We include our code for experiments in the supplementary material.

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

# Appendices: Scaling-up Diverse Orthogonal Convolutional Networks by a Paraunitary Framework

**Notations.** We use non-bold letters for *scalar* (e.g., $x$) and bold ones for *vectors* or *matrices* (e.g., $\boldsymbol{x}$). We denote sequences in the *spatial domain* using lower-case letters (e.g., $x[n]$, $\boldsymbol{x}[n]$) and their *spectral representations* using upper-case letters (e.g., $X(z)$, $\boldsymbol{X}(z)$). For a positive integer, say $R \in Z^+$, we abbreviate the set $\{0, 1, \cdots, R-1\}$ as $[R]$, and whenever possible, we use its lower-case letter, say $r \in [R]$, as the corresponding iterator.

**Assumptions.** For simplicity, we assume all sequences are $\mathcal{L}^2$ with range $\mathbb{Z} = \{0, \pm 1, \pm 2, \cdots\}$ (a sequence $\boldsymbol{x} = \{\boldsymbol{x}[n], n \in \mathbb{Z}\}$ is $\mathcal{L}^2$ if $\sum_{n \in \mathbb{Z}} \|\boldsymbol{x}[n]\|^2 < \infty$). Such assumption is common in the literature, which avoids boundary conditions in signal analysis. To deal with periodic sequences (finite sequences with circular padding), one can either adopt the Dirac function in the spectral domain or use discrete Fourier transform to compute the spectral representations. In our implementation, we address the boundary condition case by case for each convolution type, with which we achieve exact orthogonality in the experiments (Section 6.1).

## A   PSEUDO CODE FOR OUR SC-FAC ALGORITHM

We include the pseudo-code for *separable complete factorization* (Section 2) in Algorithm 1 and *diverse orthogonal convolutions* (Section 3) in Algorithm 2. The pseudo-code in Algorithm 1 consists of three parts: **(1)** First, we obtain orthogonal matrices from skew-symmetric matrices using matrix exponential. We use GeoTorch library (Lezcano Casado, 2019) for the function matrix_exp in our implementation; **(2)** Subsequently, we construct two 1D paraunitary systems using these orthogonal matrices; **(3)** Lastly, we compose two 1D paraunitary systems to obtain one 2D paraunitary systems The pseudo-code in Algorithm 2 consists of two parts: **(1)** First, we reshape each paraunitary system into an orthogonal convolution depending on the stride; and **(5)** second, we concatenate the orthogonal kernels for different groups and return the output.

## B   ORTHOGONAL CONVOLUTIONS VIA PARAUNITARY SYSTEMS

In this section, we prove the convolution theorem and Parseval's theorem for standard convolutional layers. Subsequently, we prove the paraunitary theorem which establishes the equivalence between orthogonal convolutional layers and paraunitary systems.

### B.1   SPECTRAL ANALYSIS OF STANDARD CONVOLUTION LAYERS

**Standard convolutional layers** are the default building blocks for convolutional neural networks. One such layer consists of a filter bank with $T \times S$ filters $\boldsymbol{h} = \{h_{ts}[n], n \in \mathbb{Z}\}_{t \in [T], s \in [S]}$, where $S, T$ are the number of input and output channels respectively. The layer maps an $S$-channel input $\boldsymbol{x} = \{x_s[i], i \in \mathbb{Z}\}_{s \in [S]}$ to a $T$-channel output $\boldsymbol{y} = \{y_t[i], i \in \mathbb{Z}\}_{t \in [T]}$ according to

$$y_t[i] = \sum_{s \in [S]} \sum_{n \in \mathbb{Z}} h_{ts}[n] x_s[i - n], \tag{B.1}$$

where $i$ indexes the output location to be computed, and $n$ indexes the filter coefficients. Alternatively, we can rewrite Equation (B.1) in matrix-vector form as

$$\boldsymbol{y}[i] = \sum_{n \in \mathbb{Z}} \boldsymbol{h}[n] \boldsymbol{x}[i - n], \tag{B.2}$$

where each $\boldsymbol{h}[n] \in \mathbb{R}^{T \times S}$ is a matrix, and each $\boldsymbol{x}[i - n] \in \mathbb{R}^S$ or $\boldsymbol{y}[i] \in \mathbb{R}^T$ is a vector.

Notice that in Equation (B.2), we group entries from *all channels* into a vector or matrix (e.g., from $\{x_0[n]\}_{s \in [S]}$ to $\boldsymbol{x}[n]$), different from a common notation that groups entries from *all locations* into a vector or matrix (e.g., from $\{x_s[n], n \in \mathbb{Z}\}$ into $\boldsymbol{x}_s$). In the matrix-vector form, a standard convolutional layer computes a vector sequence $\boldsymbol{y} = \{\boldsymbol{y}[i] \in \mathbb{R}^T, i \in \mathbb{Z}\}$ with a convolution between a matrix sequence $\boldsymbol{h} = \{\boldsymbol{h}[n] \in \mathbb{R}^{T \times S}, n \in \mathbb{Z}\}$ and a vector sequence $\boldsymbol{x} = \{\boldsymbol{x}[i] \in \mathbb{R}^S, i \in \mathbb{Z}\}$.

---

**Algorithm 1:** Separable Complete Factorization (SC-Fac)

---

**Input:** Number of channels $C$, kernel size $K = 2L + 1$, and
Skew-symmetric matrices $\{A_d^{(\ell)}\}$ with $A_d^{(\ell)} \in \mathbb{R}^{C \times C}, \forall \ell \in [-L, L], d \in \{1, 2\}$.

**Output:** A paraunitary system $\mathcal{H} \in \mathbb{R}^{C \times C \times K \times K}$.

**Initialization:** Sample $N_d^{(\ell)}$ from $\{1, \cdots, C\}$ uniformly $\forall \ell \in [-L, L], d \in \{1, 2\}$

```
/* Iterate for vertical/horizontal dimensions                          */
```
**for** $d = 1$ **to** $2$ **do**
    ```/* 1) Compute orthogonal matrices from skew-symmetric matrices   */```
    ```/* Iterate for filter locations                                 */```
    **for** $\ell = -L$ **to** $L$ **do**
        **if** $\ell = 0$ **then**
            $Q_d \leftarrow \mathsf{matrix\_exp}(A_d^{(0)})$ `// use matrix_exp() in GeoTorch`
            `(Lezcano Casado, 2019)`
        **else**
            $U_d^{(\ell)} \leftarrow \mathsf{select}(\mathsf{matrix\_exp}(A_d^{(\ell)}), \mathsf{cols} = N_d^{(\ell)})$ `// selects the first cols`
            `columns of the matrix`
        **end if**
    **end for**
    ```/* 2) Compose 1D paraunitary systems from orthogonal matrices    */```
    $\mathcal{H}_d \leftarrow Q_d$
    **for** $\ell = 1$ **to** $L$ **do**
        $\mathcal{H}_d \leftarrow \mathsf{conv1d}\left(\mathcal{H}_d, \left[U_d^{(\ell)} U_d^{(\ell)^\top}, I - U_d^{(\ell)} U_d^{(\ell)^\top}\right]\right)$
        $\mathcal{H}_d \leftarrow \mathsf{conv1d}\left(\left[I - U_d^{(-\ell)} U_d^{(-\ell)^\top}, U_d^{(-\ell)} U_d^{(-\ell)^\top}\right], \mathcal{H}_d\right)$
    **end for**
**end for**
```/* 3) Compose a 2D paraunitary systems from two 1D paraunitary   */```
$\mathcal{H} \leftarrow \mathsf{Compose}(\mathcal{H}_1, \mathcal{H}_2)$ `// i.e.,` $\mathcal{H}_{:,:,i,j} = (\mathcal{H}_2)_{:,:,j}(\mathcal{H}_1)_{:,:,i}$ `where the 1D`
   `paraunitary systems` $\mathcal{H}_1$ `and` $\mathcal{H}_2$ `are of size` $C \times C \times K$
**return** $\mathcal{H}$

---

**Algorithm 2:** Construct Diverse Orthogonal Convolutions from Paraunitary Systems

---

**Input:** Number of base channels $C$, kernel size $K = R(2L + 1)$,
stride $R$, dilation $D$, number of groups $G$

**Output:** An orthogonal kernel $\mathcal{W} \in \mathbb{R}^{T \times S \times K \times K}$

Set $K' \leftarrow K/R$, number of input channels $S \leftarrow GC/R^2$ and output channels $T \leftarrow GC$

**for** $g = 0$ **to** $G - 1$ **do**
    ```/* 1) Construct orthogonal convolutions from paraunitary systems */```
    Initialize skew-symmetric matrices $\{\{A_d^{(\ell,g)}\}_{\ell=-L}^{L}\}_{d=1}^{2}$ for the current $g$
    $\mathcal{H}^g \leftarrow$ Algorithm 1: SC-Fac$(C, K', \{\{A_d^{(\ell,g)}\}_{\ell=-L}^{L}\}_{d=1}^{2})$
    $\mathcal{H}^g \leftarrow \mathsf{reshape}(\mathcal{H}^g, (C, C, K', K') \rightarrow (C/R^2, C, K, K))$
**end for**
```/* 2) Concatenate orthogonal convolutions from different groups  */```
$\mathcal{W} \leftarrow \mathsf{concatenate}(\{\mathcal{H}^g\}_{g=0}^{G-1}, \mathsf{dim} = 0)$
**return** $\mathcal{W}$ (where the filter for input channel $s$ and output channel $t$ is $\mathcal{W}_{t,s,:,:} \in \mathbb{R}^{K \times K}$)

---

Let us first define the **Z-transform** and various types of **Fourier transform** in Definition B.1 before proving the convolution theorem (Theorem 2.1).

**Definition B.1** (Z-transform and Fourier transforms). *For a sequence (of scalars, vectors, or matrices)* $\boldsymbol{x} = \{\boldsymbol{x}[n], n \in \mathbb{Z}\}$, *its Z-transform* $\boldsymbol{X}(z)$ *is defined as*

$$\boldsymbol{X}(z) = \sum_{n \in \mathbb{Z}} \boldsymbol{x}[n] z^{-n}, \tag{B.3}$$

*where* $z \in \mathbb{C}$ *is a complex number such that the infinite sum is convergent. If* $z$ *is restricted to the unit circle* $z = e^{\mathrm{j}\omega}$ *(i.e.,* $|z| = 1$*), the z-transform* $\boldsymbol{X}(z)$ *reduces to a discrete-time Fourier transform (DTFT)* $\boldsymbol{X}(e^{\mathrm{j}\omega})$. *If* $\omega$ *is further restricted to a finite set* $\omega \in \{2\pi k/N, k \in [N]\}$, *the DTFT* $\boldsymbol{X}(e^{\mathrm{j}\omega})$ *reduces to an* $N$*-points discrete Fourier transform (DFT)* $\boldsymbol{X}(e^{\mathrm{j}2\pi k/N})$.

The celebrated **convolution theorem** states that the convolution in spatial domain (Equation (B.2)) is equivalent to a multiplication in the spectral domain, i.e., $\boldsymbol{Y}(z) = \boldsymbol{H}(z)\boldsymbol{X}(z), \forall z \in \mathbb{C}$.

*Proof of Theorem 2.1.* The proof follows directly from the definitions of standard convolution (Equation (B.2)) and Z-transform (Equation (B.3)).

$$\boldsymbol{Y}(z) = \sum_{i \in \mathbb{Z}} \boldsymbol{y}[i] z^{-i} \tag{B.4}$$

$$= \sum_{i \in \mathbb{Z}} \left( \sum_{n \in \mathbb{Z}} \boldsymbol{h}[n]\boldsymbol{x}[i-n] \right) z^{-i} \tag{B.5}$$

$$= \sum_{n \in \mathbb{Z}} \boldsymbol{h}[n] z^{-n} \left( \sum_{i \in \mathbb{Z}} \boldsymbol{x}[i-n] z^{-(i-n)} \right) \tag{B.6}$$

$$= \left( \sum_{n \in \mathbb{Z}} \boldsymbol{h}[n] z^{-n} \right) \left( \sum_{k \in \mathbb{Z}} \boldsymbol{x}[k] z^{-k} \right) \tag{B.7}$$

$$= \boldsymbol{H}(z)\boldsymbol{X}(z), \tag{B.8}$$

where Equations (B.4) and (B.8) use the definition of Z-transform, Equation (B.5) uses the definition of convolution, and Equation (B.7) makes a change of variable $k = i - n$. □

Next, we introduce the concepts of **inner product** and **Frobenius norm** for sequences. We then prove **Parseval's theorem**, which allows us to compute the sequence norm in the spectral domain.

**Definition B.2** (Inner product and norm for sequences). *Given two sequences* $\boldsymbol{x} = \{\boldsymbol{x}[n], n \in \mathbb{Z}\}$ *and* $\boldsymbol{y} = \{\boldsymbol{y}[n], n \in \mathbb{Z}\}$ *with* $\boldsymbol{x}[n], \boldsymbol{y}[n]$ *having the same dimension for all* $n$, *the inner product of these two sequences is defined as*

$$\langle \boldsymbol{x}, \boldsymbol{y} \rangle \triangleq \sum_{n \in \mathbb{Z}} \langle \boldsymbol{x}[n], \boldsymbol{y}[n] \rangle \tag{B.9}$$

*where* $\langle \boldsymbol{x}[n], \boldsymbol{y}[n] \rangle$ *denotes the Frobenius inner product between* $\boldsymbol{x}[n]$ *and* $\boldsymbol{y}[n]$. *Subsequently, we can define the Frobenius norm of a sequence using inner product as*

$$\|\boldsymbol{x}\| \triangleq \sqrt{\langle \boldsymbol{x}, \boldsymbol{x} \rangle} \tag{B.10}$$

**Theorem B.3** (Parsavel's theorem). *Given a sequence* $\boldsymbol{x} = \{\boldsymbol{x}[n], n \in \mathbb{Z}\}$, *its sequence norm* $\|\boldsymbol{x}\|$ *can be computed by* $\boldsymbol{X}(e^{\mathrm{j}\omega})$ *in the spectral domain as*

$$\|\boldsymbol{x}\|^2 = \sum_{n \in \mathbb{Z}} \|\boldsymbol{x}[n]\|^2 = \frac{1}{2\pi} \int_{-\pi}^{\pi} \|\boldsymbol{X}(e^{\mathrm{j}\omega})\|^2 d\omega, \tag{B.11}$$

*where* $\|\boldsymbol{X}(e^{\mathrm{j}\omega})\|^2 = \boldsymbol{X}(e^{\mathrm{j}\omega})^\dagger \boldsymbol{X}(e^{\mathrm{j}\omega})$ *is an inner product between two complex arrays.*

*Proof of Theorem B.3.* The theorem follows from the definitions of convolution and discrete-time Fourier transform (DTFT).

$$\frac{1}{2\pi} \int_{-\pi}^{\pi} \left\| \mathbf{X}(e^{j\omega}) \right\|^2 d\omega = \frac{1}{2\pi} \int_{-\pi}^{\pi} \left\langle \mathbf{X}(e^{j\omega}), \mathbf{X}(e^{j\omega}) \right\rangle d\omega \tag{B.12}$$

$$= \frac{1}{2\pi} \int_{-\pi}^{\pi} \left\langle \sum_{n \in \mathbb{Z}} \mathbf{x}[n] e^{-j\omega n}, \sum_{m \in \mathbb{Z}} \mathbf{x}[m] e^{-j\omega m} \right\rangle d\omega \tag{B.13}$$

$$= \sum_{n \in \mathbb{Z}} \sum_{m \in \mathbb{Z}} \langle \mathbf{x}[n], \mathbf{x}[m] \rangle \frac{1}{2\pi} \int_{-\pi}^{\pi} e^{-j\omega(m-n)} d\omega \tag{B.14}$$

$$= \sum_{n \in \mathbb{Z}} \sum_{m \in \mathbb{Z}} \langle \mathbf{x}[n], \mathbf{x}[m] \rangle \, \mathbb{1}_{m=n} \tag{B.15}$$

$$= \sum_{n \in \mathbb{Z}} \langle \mathbf{x}[n], \mathbf{x}[n] \rangle = \sum_{n \in \mathbb{Z}} \left\| \mathbf{x}[n] \right\|^2 , \tag{B.16}$$

where Equation (B.14) is due to the bi-linearity of inner products, and Equation (B.15) makes uses of the fact that $\int_{-\pi}^{\pi} e^{-j\omega k} d\omega = 0$ for $k \neq 0$ and $\int_{-\pi}^{\pi} e^{-j\omega k} d\omega = \int_{-\pi}^{\pi} d\omega = 2\pi$ for $k = 0$. $\square$

## B.2 Equivalence between Orthogonal Convolutions and Paraunitary Systems

With the sequence norm introduced earlier, we formally define orthogonality for convolutional layers.

**Definition B.4** (Orthogonal convolutional layer). *A convolution layer is orthogonal if the input norm $\|\mathbf{x}\|$ is equal to the output norm $\|\mathbf{y}\|$ for arbitrary input $\mathbf{x}$, that is*

$$\|\mathbf{y}\| \triangleq \sqrt{\sum_{n \in \mathbb{Z}} \|\mathbf{y}[n]\|^2} = \sqrt{\sum_{n \in \mathbb{Z}} \|\mathbf{x}[n]\|^2} \triangleq \|\mathbf{x}\|, \tag{B.17}$$

*where $\|\mathbf{x}\|$ (or $\|\mathbf{y}\|$) is defined as the squared root of $\sum_{n \in \mathbb{Z}} \|\mathbf{x}[n]\|^2$ (or $\sum_{n \in \mathbb{Z}} \|\mathbf{y}[n]\|^2$).*

This definition of orthogonality not only applies to standard convolutions in Equation (B.2) but also variants of convolutions in Appendix C.3. In this section, however, we first establish the equivalence between *orthogonality for standard convolutions* and *paraunitary systems*.

**Theorem B.5** (Paraunitary theorem). *A standard convolutional layer (in Equation (B.2)) is orthogonal (by Definition B.4) if and only if its transfer matrix $\mathbf{H}(z)$ is paraunitary, i.e.,*

$$\mathbf{H}(z)^{\dagger} \mathbf{H}(z) = \mathbf{I}, \ \forall |z| = 1 \iff \mathbf{H}(e^{j\omega})^{\dagger} \mathbf{H}(e^{j\omega}) = \mathbf{I}, \ \forall \omega \in \mathbb{R}. \tag{B.18}$$

*In other words, the transfer matrix $\mathbf{H}(e^{j\omega})$ is unitary for all frequencies $\omega \in \mathbb{R}$.*

*Proof of Theorem B.5.* We first prove that a convolutional layer is orthogonal if its transfer matrix $\mathbf{H}(z)$ is paraunitary (i.e., $\mathbf{H}(e^{j\omega})$ is unitary for any frequency $\omega \in \mathbb{R}$).

$$\|\mathbf{y}\|^2 = \frac{1}{2\pi} \int_{-\pi}^{\pi} \left\| \mathbf{Y}(e^{j\omega}) \right\|^2 d\omega \tag{B.19}$$

$$= \frac{1}{2\pi} \int_{-\pi}^{\pi} \left\| \mathbf{H}(e^{j\omega}) \mathbf{X}(e^{j\omega}) \right\|^2 d\omega \tag{B.20}$$

$$= \frac{1}{2\pi} \int_{-\pi}^{\pi} \left\| \mathbf{X}(e^{j\omega}) \right\|^2 d\omega \tag{B.21}$$

$$= \|\mathbf{x}\|^2 \tag{B.22}$$

where Equations (B.19) and (B.22) are due to Parseval's theorem (Theorem B.3), Equations (B.19) and (B.20) follows from the convolution theorem (Theorem 2.1), and Equation (B.21) utilizes that $\mathbf{H}(e^{j\omega})$ is unitary for any frequency $\omega \in \mathbb{R}$ (thus $\|\mathbf{H}(e^{j\omega})\mathbf{X}(e^{j\omega})\| = \|\mathbf{X}(e^{j\omega})\|$ for any $\mathbf{X}(e^{j\omega})$).

The 'only if' part also holds in practice. We here prove by contradiction using periodic inputs (e.g., finite inputs with circular padding). Suppose there exists a frequency $\omega$ and $\mathbf{H}(e^{j\omega})$ is not unitary.

Since $\boldsymbol{H}(e^{\mathrm{j}\omega})$ is continuous (due to $h \in \mathcal{L}^2$ and dominated convergence theorem), there exist integers $N, k$, such that $\omega \approx 2k\pi/N$ and $\boldsymbol{H}(e^{\mathrm{j}2\pi k/N})$ is also not unitary. As a result, there exists a complex vector $\boldsymbol{u}$ such that $\boldsymbol{v} = \boldsymbol{H}(e^{\mathrm{j}2\pi k/N})\boldsymbol{u}$ while $\|\boldsymbol{v}\| \neq \|\boldsymbol{u}\|$. Therefore, we can construct two periodic sequences $\boldsymbol{x} = \{\boldsymbol{x}[n], n \in [N]\}$ and $\boldsymbol{y} = \{\boldsymbol{y}[n], n \in [N]\}$ such that

$$\boldsymbol{x}[n] = \boldsymbol{u}e^{\mathrm{j}2\pi nk/N} \implies \boldsymbol{y}[n] = \boldsymbol{v}e^{\mathrm{j}2\pi nk/N}. \tag{B.23}$$

Now the input norm $\|\boldsymbol{x}\| = \sqrt{\sum_{n \in [N]} \|\boldsymbol{x}[n]\|^2} = \sqrt{N}\|\boldsymbol{u}\|$ is not equal to the output norm $\|\boldsymbol{y}\| = \sqrt{\sum_{n \in [N]} \|\boldsymbol{y}[n]\|^2} = \sqrt{N}\|\boldsymbol{v}\|$, i.e., the layer is not orthogonal, which leads to a contradiction. $\quad\square$

## C  A PARAUNITARY FRAMEWORK FOR ORTHOGONAL CONVOLUTIONS

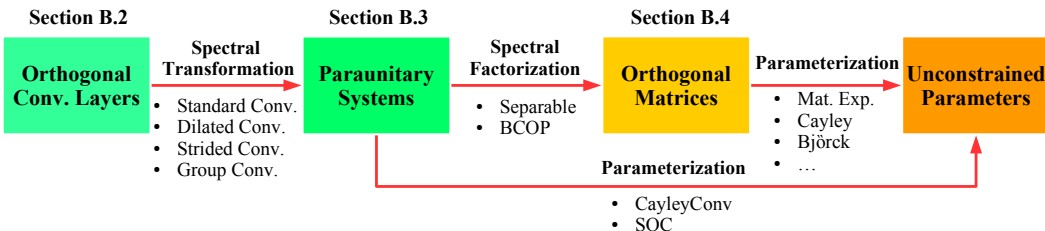

Figure 5: **A framework for designing orthogonal convolutional layers.** In Appendix C.2, we unify variants of orthogonal convolutions in the spectral domain and show that their designs reduce to constructing paraunitary systems. In Appendix C.3, we show that a paraunitary system can be constructed with different approaches: our approach and BCOP (Li et al., 2019b) represent the paraunitary using orthogonal matrices, while CayleyConv (Trockman & Kolter, 2021) and SOC (Singla & Feizi, 2021) directly parameterizes it using unconstrained parameters. In Appendix C.4, we investigate various parameterizations for orthogonal matrices, such as matrix exponential, Cayley transform, and Björck orthogonalization.

### C.1  MULTI-RESOLUTION ANALYSIS

Multi-resolution operations are essential in various convolutional layers, in particular strided and dilated convolutions. In order to define and analyze these convolutions rigorously, we first review the concepts of *up-sampling*, *down-sampling*, and *polyphase components*.

**(1) Up-sampling.** Given a sequence (of scalars, vectors, matrices) $\boldsymbol{x} = \{\boldsymbol{x}[n], n \in \mathbb{Z}\}$, its *up-sampled sequence* $\boldsymbol{x}^{\uparrow R} = \{\boldsymbol{x}^{\uparrow R}[n], n \in \mathbb{Z}\}$ is defined as

$$\boldsymbol{x}^{\uparrow R}[n] \triangleq \begin{cases} \boldsymbol{x}[n/R] & n \equiv 0 \,(\mathrm{mod}\, R) \\ 0 & \mathrm{otherwise} \end{cases}, \tag{C.1}$$

where $R \in \mathbb{Z}^+$ is the up-sampling rate. Accordingly, we denote the Z-transform of $\boldsymbol{x}^{\uparrow R}$ as

$$\boldsymbol{X}^{\uparrow R}(z) = \sum_{n \in \mathbb{Z}} \boldsymbol{x}^{\uparrow R}[n]z^{-n}. \tag{C.2}$$

The following proposition shows that $\boldsymbol{X}^{\uparrow R}(z)$ is easily computed from $\boldsymbol{X}(z)$.

**Proposition C.1** (Z-transform of up-sampled sequence). *Given a sequence $\boldsymbol{x}$ and its up-sampled sequence $\boldsymbol{x}^{\uparrow R}$, their Z-transforms $\boldsymbol{X}(z)$ and $\boldsymbol{X}^{\uparrow R}(z)$ are related by*

$$\boldsymbol{X}^{\uparrow R}(z) = \boldsymbol{X}(z^R). \tag{C.3}$$

*Proof of Proposition C.1.* The proof makes use of the definition of Z-transform (Equation (B.3)).

$$\boldsymbol{X}^{\uparrow R}(z) = \sum_{n \in \mathbb{Z}} \boldsymbol{x}^{\uparrow R}[n] z^{-n} \tag{C.4}$$

$$= \sum_{m \in \mathbb{Z}} \boldsymbol{x}^{\uparrow R}[mR] z^{-mR} \tag{C.5}$$

$$= \sum_{m \in \mathbb{Z}} \boldsymbol{x}[m](z^R)^{-m} = \boldsymbol{X}(z^R), \tag{C.6}$$

where Equation (C.5) makes a change of variables $m = n/R$ since $\boldsymbol{x}^{\uparrow R}[n] = 0, \forall n \neq mR$. □

**(2) Down-sampling and polyphase components.** Different from the up-sampled sequence, there exist multiple down-sampled sequences, depending on the *phase* of down-sampling. These sequences are known as the *polyphase components*. Specifically, given a sequence (of scalars, vectors, or matrices) $\boldsymbol{x} = \{\boldsymbol{x}[n], n \in \mathbb{Z}\}$, its $r^{\text{th}}$ polyphase component $\boldsymbol{x}^{r|R} = \{\boldsymbol{x}^{r|R}[n], n \in \mathbb{Z}\}$ is defined as

$$\boldsymbol{x}^{r|R}[n] \triangleq \boldsymbol{x}[nR + r], \tag{C.7}$$

where $R \in \mathbb{Z}^+$ is the down-sampling rate. We further denote the Z-transform of $\boldsymbol{x}^{r|R}$ as

$$\boldsymbol{X}^{r|R}(z) = \sum_{n \in \mathbb{Z}} \boldsymbol{x}^{r|R}[n] z^{-n}, \tag{C.8}$$

Note that $r \in \mathbb{Z}$ is an arbitrary integer, which does not necessarily take values from $[R]$. In fact, we have $\boldsymbol{x}^{(r+kR)|R}[n] = \boldsymbol{x}^{r|R}[n+k]$ and $\boldsymbol{X}^{r+kR|R}(z) = z^k \boldsymbol{X}^{r|R}(z)$. In Proposition C.2, we establish the relation between $\boldsymbol{H}(z)$ and $\{\boldsymbol{H}^{r|R}(z)\}_{r \in [R]}$, i.e., to represent $\boldsymbol{H}(z)$ in terms of $\{\boldsymbol{H}^{r|R}(z)\}_{r \in [R]}$.
**Proposition C.2** (Polyphase decomposition)**.** *Given a sequence $\boldsymbol{x}$ and its polyphase components $\boldsymbol{x}^{r|R}$'s, the Z-transform $\boldsymbol{X}(z)$ can be represented by $\{\boldsymbol{X}^{r|R}(z)\}_{r \in [R]}$ as*

$$\boldsymbol{X}(z) = \sum_{r \in [R]} \boldsymbol{X}^{r|R}(z^R) z^{-r}. \tag{C.9}$$

*Proof of Proposition C.2.* We start with $\boldsymbol{X}(z)$, and try to decompose it into its polyphase components $\boldsymbol{X}^{0|R}(z), \cdots, \boldsymbol{X}^{R-1|R}(z)$.

$$\boldsymbol{X}(z) = \sum_{n \in \mathbb{Z}} \boldsymbol{x}[n] z^{-n} \tag{C.10}$$

$$= \sum_{r \in [R]} \sum_{m \in \mathbb{Z}} \boldsymbol{x}[mR + r] z^{-(mR+r)} \tag{C.11}$$

$$= \sum_{r \in [R]} \left( \sum_{m \in \mathbb{Z}} \boldsymbol{x}[mR + r] z^{-mR} \right) z^{-r} \tag{C.12}$$

$$= \sum_{r \in [R]} \left( \sum_{m \in \mathbb{Z}} \boldsymbol{x}^{r|R}[m](z^R)^{-m} \right) z^{-r} \tag{C.13}$$

$$= \sum_{r \in [R]} \boldsymbol{X}^{r|R}(z^R) z^{-r}, \tag{C.14}$$

where Equation (C.11) makes a change of variables $n = mR + r$, and Equation (C.13) is the definition of polyphase components $\boldsymbol{x}^{r|R}[m] = \boldsymbol{x}[mR + r]$. □

For simplicity, we stack $R$ consecutive polyphase components into **polyphase matrices** as

$$\boldsymbol{X}^{[R]}(z) = \begin{bmatrix} \boldsymbol{X}^{0|R}(z) \\ \vdots \\ \boldsymbol{X}^{R-1|R}(z) \end{bmatrix}, \quad \widetilde{\boldsymbol{X}}^{[R]}(z) = \left[ \boldsymbol{X}^{-0|R}(z); \cdots; \boldsymbol{X}^{-(R-1)|R}(z) \right]. \tag{C.15}$$

The following proposition extends the Parseval's theorem in Theorem B.3 and shows that the sequence norm $\|\boldsymbol{x}\|$ can also be computed in terms of the polyphase matrix $\boldsymbol{X}^{[R]}(z)$ (or $\widetilde{\boldsymbol{X}}^{[R]}(z)$).

**Proposition C.3** (Parseval's theorem for polyphase matrices). *Given a sequence $\boldsymbol{x}$, its sequence norm $\|\boldsymbol{x}\|$ can be computed by $\boldsymbol{X}^{[R]}(e^{j\omega})$ (or $\widetilde{\boldsymbol{X}}^{[R]}(e^{j\omega})$) in the spectral domain as*

$$\|\boldsymbol{x}\|^2 = \frac{1}{2\pi} \int_{-\pi}^{\pi} \left\| \boldsymbol{X}^{[R]}(e^{j\omega}) \right\|^2 d\omega = \frac{1}{2\pi} \int_{-\pi}^{\pi} \left\| \widetilde{\boldsymbol{X}}^{[R]}(e^{j\omega}) \right\|^2 d\omega. \tag{C.16}$$

*Proof of Proposition C.3.* The proof follows the standard Parseval's theorem in Theorem B.3. We only prove the first part of the proposition (using $\boldsymbol{X}^{[R]}(e^{j\omega})$) as follows.

$$\|\boldsymbol{x}\|^2 = \sum_{n \in \mathbb{Z}} \|\boldsymbol{x}[n]\|^2 \tag{C.17}$$

$$= \sum_{r \in [R]} \sum_{m \in \mathbb{Z}} \|\boldsymbol{x}[mR + r]\|^2 \tag{C.18}$$

$$= \sum_{r \in [R]} \sum_{m \in \mathbb{Z}} \left\| \boldsymbol{x}^{r|R}[m] \right\|^2 \tag{C.19}$$

$$= \frac{1}{2\pi} \sum_{r \in [R]} \int_{-\pi}^{\pi} \left\| \boldsymbol{X}^{r|R}(e^{j\omega}) \right\|^2 d\omega \tag{C.20}$$

$$= \frac{1}{2\pi} \int_{-\pi}^{\pi} \left\| \boldsymbol{X}^{[R]}(e^{j\omega}) \right\|^2 d\omega, \tag{C.21}$$

where Equation (C.17) follows the definition of sequence norm, Equation (C.18) changes variables as $n = mR + r$, Equation (C.19) is the definition of polyphase components, and Equation (C.20) applies Parseval's theorem to $\boldsymbol{x}^{r|R}$'s. The second part (using $\widetilde{\boldsymbol{X}}^{[R]}(e^{j\omega})$) can be proved similarly. □

## C.2 UNIFYING VARIOUS CONVOLUTIONAL LAYERS IN THE SPECTRAL DOMAIN

In Appendix B, the convolution theorem states that a standard convolutional layer is a matrix-vector product $\boldsymbol{Y}(z) = \boldsymbol{H}(z)\boldsymbol{X}(z)$ in the spectral domain, and the layer is orthogonal if and only if $\boldsymbol{H}(z)$ is paraunitary (Theorem B.5). However, *the canonical convolution theorem does not hold for variants of convolutions, thus enforcing a paraunitary $\boldsymbol{H}(z)$ may not lead to orthogonal convolution*. In this subsection, we address this limitation by showing that various convolutions can be uniformly written as $\underline{\boldsymbol{Y}}(z) = \underline{\boldsymbol{H}}(z)\underline{\boldsymbol{X}}(z)$, where $\underline{\boldsymbol{Y}}(z)$, $\underline{\boldsymbol{H}}(z)$, $\underline{\boldsymbol{X}}(z)$ are some spectral representations of $\boldsymbol{y}$, $\boldsymbol{h}$, $\boldsymbol{x}$. Subsequently, we prove that any of these layers is orthogonal if and only if its $\underline{\boldsymbol{H}}(z)$ is paraunitary.

**(1) Strided convolutional layers** are widely used in neural networks to adjust the feature resolution: a strided convolution layer decreases the resolution by down-sampling after a standard convolution, while a transposed convolution increases the resolution by up-sampling before a standard convolution.

Formally, a strided convolutional layer with stride $R$ (abbrev. as $\downarrow R$-strided convolution) computes its output following

$$\boldsymbol{y}[i] = \sum_{n \in \mathbb{Z}} \boldsymbol{h}[n]\boldsymbol{x}[Ri - n]. \tag{C.22}$$

In contrast, a transposed strided convolutional layer with stride $R$ (abbrev. as $\uparrow R$-strided convolution) computes its output according to

$$\boldsymbol{y}[i] = \sum_{n \in \mathbb{Z}} \boldsymbol{h}[n]\boldsymbol{x}^{\uparrow R}[i - n]. \tag{C.23}$$

**Proposition C.4** (Orthogonality of strided convolutional layers). *For a $\downarrow R$-strided convolution, the spatial convolution in Equation (C.22) leads to the following spectral representation:*

$$\boldsymbol{Y}(z) = \widetilde{\boldsymbol{H}}^{[R]}(z)\boldsymbol{X}^{[R]}(z) \tag{C.24}$$

*And for an $\uparrow R$-strided convolution, the spatial convolution is represented in spectral domain as:*

$$\boldsymbol{Y}^{[R]}(z) = \boldsymbol{H}^{[R]}(z)\boldsymbol{X}(z) \tag{C.25}$$

*Furthermore, a $\downarrow R$-strided convolution is orthogonal if and only if $\widetilde{\boldsymbol{H}}^{[R]}(z)$ is paraunitary, and an $\uparrow R$-strided convolution is orthogonal if and only if $\boldsymbol{H}^{[R]}[z]$ is paraunitary.*

*Proof of Proposition C.4.* **(1a)** $\downarrow R$**-strided convolutions.** We first prove the spectral representation of $\downarrow R$-strided convolution in Equation (C.24).

$$\boldsymbol{Y}(z) = \sum_{i\in\mathbb{Z}} \boldsymbol{y}[i]z^{-i} = \sum_{i\in\mathbb{Z}}\left(\sum_{n\in\mathbb{Z}}\boldsymbol{h}[n]\boldsymbol{x}[Ri-n]\right)z^{-i} \tag{C.26}$$

$$= \sum_{i\in\mathbb{Z}}\left(\sum_{r\in[R]}\sum_{m\in\mathbb{Z}}\boldsymbol{h}[mR-r]\boldsymbol{x}[(i-m)R+r]\right)z^{-i} \tag{C.27}$$

$$= \sum_{r\in[R]}\left(\sum_{m\in\mathbb{Z}}\boldsymbol{h}[mR-r]z^{-m}\left(\sum_{i}\boldsymbol{x}[(i-m)R+r]z^{-(i-m)}\right)\right) \tag{C.28}$$

$$= \sum_{r\in[R]}\left(\sum_{m\in\mathbb{Z}}\boldsymbol{h}[mR-r]z^{-m}\right)\left(\sum_{i'\in\mathbb{Z}}\boldsymbol{x}[i'R+r]z^{-i'}\right) \tag{C.29}$$

$$= \sum_{r\in[R]}\boldsymbol{H}^{-r|R}(z)\boldsymbol{X}^{r|R}(z), \tag{C.30}$$

where Equation (C.26) follows from the definitions of the $\downarrow R$-strided convolution (Equation (C.22)) and the Z-transform (Equation (B.3)), Equation (C.27) makes a change of variables $n = mR - r$, Equation (C.29) further changes $i' = i - m$, and Equation (C.30) is due to the definition of polyphase components (Equation (C.7)). Now We rewrite the last equation concisely as

$$\boldsymbol{Y}(z) = \underbrace{\left[\boldsymbol{H}^{-0|R}(z); \cdots; \boldsymbol{H}^{-(R-1)}(z)\right]}_{\widetilde{\boldsymbol{H}}^{[R]}(z)}\underbrace{\begin{bmatrix}\boldsymbol{X}^{0|R}(z)\\ \vdots \\ \boldsymbol{X}^{R-1|R}(z)\end{bmatrix}}_{\boldsymbol{X}^{[R]}(z)}, \tag{C.31}$$

which is the spectral representation of $\downarrow R$-strided convolutions in Equation (C.24).

Now we prove the orthogonality condition for $\downarrow R$-strided convolutions.

$$\|\boldsymbol{y}\|^2 = \frac{1}{2\pi}\int_{-\pi}^{\pi}\left\|\boldsymbol{Y}(e^{j\omega})\right\|^2 d\omega \tag{C.32}$$

$$= \frac{1}{2\pi}\int_{-\pi}^{\pi}\left\|\widetilde{\boldsymbol{H}}^{[R]}(e^{j\omega})\boldsymbol{X}^{[R]}(e^{j\omega})\right\|^2 d\omega \tag{C.33}$$

$$= \frac{1}{2\pi}\int_{-\pi}^{\pi}\left\|\boldsymbol{X}^{[R]}(e^{j\omega})\right\|^2 d\omega \tag{C.34}$$

$$= \|\boldsymbol{x}\|^2, \tag{C.35}$$

where Equations (C.32) and (C.35) are due to Parseval's theorems (Theorem B.3 and Proposition C.3), Equation (C.33) follows from the spectral representation of the $\downarrow R$-strided convolution (Equation (C.24)), and Equation (C.34) utilizes that the transfer matrix is unitary at each frequency. The "only if" part can be proved by contradiction similar to Theorem B.5.

**(1b)** $\uparrow R$**-strided convolutions.** According to Proposition C.1, the Z-transform of $\boldsymbol{x}^{\uparrow R}$ is $\boldsymbol{X}(z^R)$. Therefore, an application of the convolution theorem (Theorem 2.1) on Equation (C.23) leads us to

$$\boldsymbol{Y}(z) = \boldsymbol{H}(z)\boldsymbol{X}(z^R) \tag{C.36}$$

Expanding $\boldsymbol{Y}(z)$ and $\boldsymbol{H}(z)$ using polyphase decomposition (Proposition C.2), we have

$$\sum_{r\in[R]}\boldsymbol{Y}^{r|R}(z^R)z^{-r} = \left(\sum_{r\in[R]}\boldsymbol{H}^{r|R}(z^R)z^{-r}\right)\boldsymbol{X}(z^R) \tag{C.37}$$

$$\sum_{r\in[R]}\boldsymbol{Y}^{r|R}(z^R)z^{-r} = \sum_{r\in[R]}\left(\boldsymbol{H}^{r|R}(z^R)\boldsymbol{X}(z^R)\right)z^{-r} \tag{C.38}$$

$$\boldsymbol{Y}^{r|R}(z^R) = \boldsymbol{H}^{r|R}(z^R)\boldsymbol{X}(z^R), \ \forall r \in [R] \tag{C.39}$$

$$\boldsymbol{Y}^{r|R}(z) = \boldsymbol{H}^{r|R}(z)\boldsymbol{X}(z), \ \forall r \in [R], \tag{C.40}$$

where Equation (C.39) is due to the uniqueness of Z-transform, and Equation (C.40) changes the variables from $z^R$ to $z$. Again,,we can rewrite the last equation in concisely as

$$\underbrace{\begin{bmatrix} \boldsymbol{Y}^{0|R}(z) \\ \boldsymbol{Y}^{1|R}(z) \\ \vdots \\ \boldsymbol{Y}^{R-1|R}(z) \end{bmatrix}}_{\boldsymbol{Y}^{[R]}(z)} = \underbrace{\begin{bmatrix} \boldsymbol{H}^{0|R}(z) \\ \boldsymbol{H}^{1|R}(z) \\ \vdots \\ \boldsymbol{H}^{R-1|R}(z) \end{bmatrix}}_{\boldsymbol{H}^{[R]}(z)} \boldsymbol{X}(z), \tag{C.41}$$

which is the spectral representation of $\uparrow R$-strided convolutions in Equation (C.25).

Lastly, we prove the orthogonality condition for $\uparrow R$-strided convolutions.

$$\|\boldsymbol{y}\|^2 = \frac{1}{2\pi} \int_{-\pi}^{\pi} \left\| \boldsymbol{Y}^{[R]}(e^{j\omega}) \right\|^2 d\omega \tag{C.42}$$

$$= \frac{1}{2\pi} \int_{-\pi}^{\pi} \left\| \boldsymbol{H}^{[R]}(e^{j\omega}) \boldsymbol{X}(e^{j\omega}) \right\|^2 d\omega \tag{C.43}$$

$$= \frac{1}{2\pi} \int_{-\pi}^{\pi} \left\| \boldsymbol{X}(e^{j\omega}) \right\|^2 d\omega \tag{C.44}$$

$$= \|\boldsymbol{x}\|^2, \tag{C.45}$$

where Equations (C.42) and (C.45) are due to Parseval's theorems (Theorem B.3 and Proposition C.3), Equation (C.43) follows from the spectral representation of the $\uparrow R$-strided convolution (Equation (C.25)), and Equation (C.44) uses the fact that the transfer matrix is unitary for each frequency. The "only if" part can be proved by contradiction similar to Theorem B.5. $\square$

**(2) Dilated convolutional layer** is proposed to increase the receptive field of a convolutional layer without extra parameters and computation. The layer up-samples its filter bank before convolution with the input. $R$-dilated convolutional layer) computes its output with the following equation:

$$\boldsymbol{y}[i] = \sum_{n \in \mathbb{Z}} \boldsymbol{h}^{\uparrow R}[n] \boldsymbol{x}[i - n] \tag{C.46}$$

**Proposition C.5** (Orthogonality of dilated convolutional layer)**.** *For an $R$-dilated convolution, the spatial convolution in Equation (C.46) leads to a spectral representation as*

$$\boldsymbol{Y}(z) = \boldsymbol{H}(z^R)\boldsymbol{X}(z), \tag{C.47}$$

*Furthermore, an $R$-dilated convolutional layer is orthogonal if and only if $\boldsymbol{H}(z^R)$ is paraunitary.*

*Proof of Proposition C.5.* According to Proposition C.1, the Z-transform of $\boldsymbol{h}^{\uparrow R}$ is $\boldsymbol{H}(z^R)$. Therefore, the "if" part follows directly from the convolution theorem. The "only if" part can be proved by constructing a counterexample similar to Theorem B.5. $\square$

Notice that $\boldsymbol{H}(z^R)$ is paraunitary if and only if $\boldsymbol{H}(e^{j\omega})$ is unitary for all frequency $\omega \in \mathbb{R}$, which is the same as $\boldsymbol{X}Hz$ being paraunitary. In other words, any filter bank that is orthogonal for a standard convolution is also orthogonal for a dilated convolution and vice versa.

**(3) Group convolutional layer** is proposed to reduce the parameters and computations and used in many efficient architectures, including MobileNet, ShuffleNet. The layer divides both input/output channels into multiple groups and restricts the connections within each group.

Formally, a group convolutional layer with $G$ groups (abbrev. as $G$-group convolutions) is parameterized by $G$ filter banks $\{\boldsymbol{h}^g\}_{g \in [G]}$, each consists of $(T/G) \times (S/G)$ filters. The layer maps an $S$ channels input $\boldsymbol{x}$ to a $T$ channels output $\boldsymbol{y}$ according to

$$\boldsymbol{y}[i] = \sum_{n \in \mathbb{Z}} \mathsf{blkdiag}\left( \{\boldsymbol{h}^g[n]\}_{g \in [G]} \right) \boldsymbol{x}[i - n], \tag{C.48}$$

where $\mathsf{blkdiag}(\{\cdot\})$ computes a block diagonal matrix from a set of matrices.

**Proposition C.6** (Orthogonality of group convolutional layer). *For a $G$-group convolution, the spatial convolution in Equation (C.48) leads a spectral representation as , their z-transforms satisfy*

$$Y(z) = \text{blkdiag}\left(\{H^g(z)\}_{g\in[G]}\right)X(z),\tag{C.49}$$

*Furthermore, a $G$-group convolutional layer is orthogonal if and only if the block diagonal matrix is paraunitary, i.e., each $h^g(z)$ is paraunitary.*

*Proof of Proposition C.6.* Due to the convolution theorem, it suffices to prove that the Z-transform of a sequence of block diagonal matrices is also block diagonal in the spectral domain.

$$\sum_{n\in\mathbb{Z}}\underbrace{\begin{bmatrix}h^0[n] & & \\ & \ddots & \\ & & h^{G-1}[n]\end{bmatrix}}_{\text{blkdiag}\left(\{h^g[n]\}_{g\in[G]}\right)}z^{-n} = \begin{bmatrix}\sum_{n\in\mathbb{Z}}h^0[n]z^{-n} & & \\ & \ddots & \\ & & \sum_{n\in\mathbb{Z}}h^{G-1}[n]z^{-n}\end{bmatrix}\tag{C.50}$$

$$= \underbrace{\begin{bmatrix}H^0(z) & & \\ & \ddots & \\ & & H^{G-1}(z)\end{bmatrix}}_{\text{blkdiag}\left(\{H^g(z)\}_{g\in[G]}\right)}.\tag{C.51}$$

As a result, we can write the orthogonality condition as

$$\begin{bmatrix}H^0(z) & & \\ & \ddots & \\ & & h^{G-1}(z)\end{bmatrix}^{\dagger}\begin{bmatrix}H^0(z) & & \\ & \ddots & \\ & & H^{G-1}(z)\end{bmatrix}$$

$$= \begin{bmatrix}H^0(z)^{\dagger}H^0(z) & & \\ & \ddots & \\ & & H^{G-1}(z)^{\dagger}H^{G-1}(z)\end{bmatrix} = I, \ \forall|z|=1.\tag{C.52}$$

The equation implies $H^g(z)^{\dagger}H^g(z) = I, \forall|z|=1, \forall g\in[G]$, i.e., each $H^g(z)$ is paraunitary. $\square$

## C.3 REALIZATIONS OF PARAUNITARY SYSTEMS

In this subsection, we first prove that all finite-length 1D-paraunitary systems can be represented in a factorized form. Next, we show how we can construct MD-paraunitary systems using 1D systems. Lastly, we study the relationship of existing approaches to paraunitary systems.

### C.3.1 COMPLETE FACTORIZATION OF 1D-PARAUNITARY SYSTEMS

The classic theorem for spectral factorization of paraunitary systems is traditionally developed for causal systems (Vaidyanathan, 1993; Kautsky & Turcajová, 1994). Given a causal paraunitary system of length $L$ (i.e., polynomial in $z^{-1}$), there always exists a factorization such that

$$H(z) = QV(z^{-1}; U^{(1)})\cdots V(z^{-1}; U^{(L-1)}),\tag{C.53}$$

where $Q$ is an orthogonal matrix, $U^{(\ell)}$ is a column-orthogonal matrix, and $V(z; U)$ is defined as

$$V(z; U) = (I - UU^{\top}) + UU^{\top}z.\tag{C.54}$$

In Theorem C.7, we extends this theorem from causal systems to finite-length (but non-causal) ones.

**Theorem C.7** (Complete factorization for 1D-paraunitary systems). *Suppose that a paraunitary system $H(z)$ is finite-length, i.e., it can be written as $\sum_n h[n]z^{-n}$ for some sequence $\{h[n], n\in[-\underline{L}, \overline{L}]\}$, then it can be factorized in the following form:*

$$H(z) = V(z; U^{(-\underline{L})})\cdots V(z; U^{(-1)})QV(z^{-1}; U^{(1)})\cdots V(z^{-1}; U^{(\overline{L})}),\tag{C.55}$$

*where $Q$ is an orthogonal matrix, $U^{(\ell)}$ is a column-orthogonal matrix, and $V(z; U)$ is defined in Equation (C.54). Consequently, the paraunitary system $H(z)$ is parameterized by $\underline{L} + \overline{L} + 1$ (column-)orthogonal matrices $Q$ and $U^{(\ell)}$'s.*

*Proof for Theorem C.7.* Given a non-causal paraunitary system $\boldsymbol{H}(z)$, we can always find a causal counterpart $\hat{\boldsymbol{H}}(z)$ such that $\boldsymbol{H}(z) = z^{\underline{L}}\hat{\boldsymbol{H}}(z)$ (This can be done by shifting the causal system backward by $\underline{L}$ steps, which is equivalent to multiplying $z^{\underline{L}}$ in the spectral domain). Since the causal system $\hat{\boldsymbol{H}}(z)$ admits a factorization in Equation (C.55), we can write the non-causal system $\boldsymbol{H}(z)$ as

$$\boldsymbol{H}(z) = z^{\underline{L}}\boldsymbol{Q}\boldsymbol{V}(z^{-1};\hat{\boldsymbol{U}}^{(1)})\cdots\boldsymbol{V}(z^{-1};\hat{\boldsymbol{U}}^{(\underline{L}+\overline{L})}). \tag{C.56}$$

Therefore, it suffices to show that for an orthogonal matrix $\boldsymbol{Q}$ and any column-orthogonal matrix $\hat{\boldsymbol{U}}$, we can always find another column-orthogonal matrix $\boldsymbol{U}$ such that

$$z\boldsymbol{Q}\boldsymbol{V}(z^{-1};\hat{\boldsymbol{U}}) = \boldsymbol{V}(z;\boldsymbol{U})\boldsymbol{Q}. \tag{C.57}$$

If the equation above is true, we can set $\boldsymbol{U}^{(\ell)} = \hat{\boldsymbol{U}}^{(\ell-1-\underline{L})}$ for $\ell < 0$ and $\boldsymbol{U}^{(\ell)} = \hat{\boldsymbol{U}}^{(\ell-\underline{L})}$ for $\ell > 0$, which will convert Equation (C.56) into Equation (C.55).

Now we start to prove Equation (C.57). Note that any column-orthogonal $\hat{\boldsymbol{U}}$ has a complement $\bar{\boldsymbol{U}}$ such that $[\hat{\boldsymbol{U}}, \bar{\boldsymbol{U}}]$ is orthogonal and $\boldsymbol{I} = \hat{\boldsymbol{U}}\hat{\boldsymbol{U}}^\top + \bar{\boldsymbol{U}}\bar{\boldsymbol{U}}^\top$. We then rewrite Equation (C.57) as

$$z\boldsymbol{Q}\boldsymbol{V}(z^{-1};\hat{\boldsymbol{U}}) = z\boldsymbol{Q}(\boldsymbol{I} - \hat{\boldsymbol{U}}\hat{\boldsymbol{U}}^\top + \hat{\boldsymbol{U}}\hat{\boldsymbol{U}}^\top z^{-1}) \tag{C.58}$$

$$= \boldsymbol{Q}(\boldsymbol{I} - \bar{\boldsymbol{U}}\bar{\boldsymbol{U}}^\top + \bar{\boldsymbol{U}}\bar{\boldsymbol{U}}^\top z) \tag{C.59}$$

$$= (\boldsymbol{I} - \boldsymbol{Q}\bar{\boldsymbol{U}}\bar{\boldsymbol{U}}^\top\boldsymbol{Q}^\top + \boldsymbol{Q}\bar{\boldsymbol{U}}\bar{\boldsymbol{U}}^\top\boldsymbol{Q}^\top z)\boldsymbol{Q} \tag{C.60}$$

$$= (\boldsymbol{I} - \boldsymbol{U}\boldsymbol{U}^\top + \boldsymbol{U}\boldsymbol{U}^\top z)\boldsymbol{Q} \tag{C.61}$$

$$= \boldsymbol{V}(z;\boldsymbol{U})\boldsymbol{Q}, \tag{C.62}$$

where in Equation (C.61) we set $\boldsymbol{U} = \boldsymbol{Q}\bar{\boldsymbol{U}}$. This completes the proof. $\square$

### C.3.2 MULTI-DIMENSIONAL (MD) PARAUNITARY SYSTEMS

If the data are multi-dimensional (MD), we will need MD-convolutional layers in neural networks. Analogously, we can prove the equivalence between orthogonal MD-convolutions in the spatial domain and MD-paraunitary systems in the spectral domain, i.e.,

$$\boldsymbol{H}(\boldsymbol{z})^\dagger\boldsymbol{H}(\boldsymbol{z}) = \boldsymbol{I}, \boldsymbol{z} = (z_1, \cdots, z_D), |z_d| = 1, \forall d \in [D], \tag{C.63}$$

where $D$ is the data dimension. In this work, we adopt a parameterization based on separable systems.

**Definition C.8** (Separable MD-paraunitary system). *A MD-paraunitary system $\boldsymbol{H}(\boldsymbol{z})$ is separable if there exists $D$ 1D-paraunitary systems $\boldsymbol{H_1}(z_1), \cdots, \boldsymbol{H_D}(z_D)$ such that*

$$\boldsymbol{H}(\boldsymbol{z}) = \boldsymbol{H}(z_1, \cdots, z_D) \triangleq \boldsymbol{H_1}(z_1)\cdots\boldsymbol{H_D}(z_D). \tag{C.64}$$

Therefore, we can construct an MD-paraunitary system with $D$ number of 1D-paraunitary systems, each of which is represented in Equation (C.55). Notice that *not* all MD-paraunitary systems are separable, thus the parameterization in Equation (C.64) is *not* complete (see Section 5 for a discussion). However, we can guarantee that our parameterization realizes all separable MD-paraunitary systems — each separable paraunitary system admits a factorization in Equation (C.64), where each 1D-system admits a factorization in Equation (C.55).

### C.3.3 INTERPRETATIONS OF PREVIOUS APPROACHES

In Theorem B.5, we have shown that a paraunitary transfer matrix is both necessary and sufficient for a convolution to be orthogonal. Therefore, we can interpret all approaches for orthogonal convolutions as implicit constructions of paraunitary systems, including *singular value clipping and masking (SVCM)* (Sedghi et al., 2019), *block convolution orthogonal parameterization (BCOP)* (Li et al., 2019b), *Cayley convolution (CayleyConv)* (Trockman & Kolter, 2021), *skew orthogonal convolution (SOC)* (Singla & Feizi, 2021). Furthermore, we prove how *orthogonal regularization* (Wang et al., 2019; Qi et al., 2020) encourages the transfer matrix to be unitary for all frequencies.

**(1) Singular value clipping and masking (SVCM)** (Sedghi et al., 2019) clips all singular values of $\boldsymbol{H}(e^{j\omega})$ to ones for each frequency $\omega$ after gradient update. Since the clipping step can arbitrarily

enlarge the filter length, SVCM subsequently masks out the coefficients outside the filter length. However, the masking step breaks the orthogonality, as we have seen in the experiments (Section 6.1).

**(2) Block convolution orthogonal parameterization (BCOP)** (Li et al., 2019b) tries to generalize the *spectral factorization* of 1D-paraunitary systems in Equation (C.55) to 2D-paraunitary systems.

$$\boldsymbol{H}(z_1, z_2) = z_1^{\frac{K-1}{2}} z_2^{\frac{K-1}{2}} \boldsymbol{Q} \boldsymbol{V}(z_1, z_2; \boldsymbol{U}_1^{(1)}, \boldsymbol{U}_2^{(1)}) \cdots \boldsymbol{V}(z_1, z_2; \boldsymbol{U}_1^{(K-1)}, \boldsymbol{U}_2^{(K-1)}), \qquad \text{(C.65)}$$

where $\boldsymbol{V}(z_1, z_2; \boldsymbol{U}_1^{(\ell)}, \boldsymbol{U}_2^{(\ell)}) = \boldsymbol{V}(z_1; \boldsymbol{U}_1^{(\ell)}) \boldsymbol{V}(z_2; \boldsymbol{U}_2^{(\ell)})$. In other words, this approach makes each $V$-block, instead of the whole paraunitary system, separable. This factorization in BCOP is incomplete for 2D-paraunitary system — unlike 1D-paraunitary systems, not every 2D-paraunitary system admits a factorized form (Lin & Vaidyanathan, 1996).

**(2) Calyey convolution (CayleyConv)** (Trockman & Kolter, 2021) aims to generalize the *Cayley transform* for orthogonal matrices in Equation (C.88) to 2D-paraunitary systems $\boldsymbol{H}(z_1, z_2)$:

$$\boldsymbol{H}(z_1, z_2) = (\boldsymbol{I} - \boldsymbol{A}(z_1, z_2)) (\boldsymbol{I} + \boldsymbol{A}(z_1, z_2))^{-1}, \qquad \text{(C.66)}$$

where $\boldsymbol{A}(z_1, z_2)$ is a skew-Hermitian matrix for $|z_1| = 1, |z_2| = 1$ (i.e., $\boldsymbol{A}(e^{j\omega_1}, e^{j\omega_2})^\dagger = \boldsymbol{A}(e^{j\omega_1}, e^{j\omega_2})$ for any $\omega_1, \omega_2$). Since Cayley transform cannot parameterize a matrix with singular value $-1$ for any frequency, the CayleyConv is not a complete parameterization.

**(4) Skew orthogonal convolution (SOC)** (Singla & Feizi, 2021) aims to generalize the *matrix exponential* for orthogonal matrices (Equation (C.90)) to *convolution exponential* for 2D-paraunitary systems $\boldsymbol{H}(z_1, z_2)$:

$$\boldsymbol{H}(z_1, z_2) = \exp(\boldsymbol{A}(z_1, z_2)) \triangleq \sum_{k=0}^\infty \frac{\boldsymbol{A}(z_1, z_2)}{k!} = \boldsymbol{I} + \boldsymbol{A}(z_1, z_2) + \frac{\boldsymbol{A}(z_1, z_2)^2}{2} + \cdots, \quad \text{(C.67)}$$

where $\boldsymbol{A}(z_1, z_2)$ is skew-Hermitian matrix for any matrix for $|z_1| = 1, |z_2| = 1$ (in other words, $\boldsymbol{A}(e^{j\omega_1}, e^{j\omega_2})^\dagger = \boldsymbol{A}(e^{j\omega_1}, e^{j\omega_2})$ for any $\omega_1, \omega_2$). It is not resolved whether all 2D-paraunitary systems can be represented in terms of convolution exponential.

**(5) Orthogonal regularization (Ortho-Reg)** (Wang et al., 2019; Qi et al., 2020) is developed to encourage orthogonality in convolutional layers. We show that such orthogonal regularization is equivalent to a unitary regularization of the paraunitary system with uniform weights on all frequencies.

$$\sum_{i \in \mathbb{Z}} \left\| \sum_{n \in \mathbb{Z}} \boldsymbol{h}[n] \boldsymbol{h}[n - Ri]^\top - \boldsymbol{\delta}[i] \right\|^2 = \frac{1}{2\pi} \int_{-\pi}^{\pi} \left\| \boldsymbol{H}^{[R]}(e^{j\omega})^\dagger \boldsymbol{H}^{[R]}(e^{j\omega}) - \boldsymbol{I} \right\|^2 d\omega, \qquad \text{(C.68)}$$

where $\boldsymbol{\delta}[0] = \boldsymbol{I}$ is an identity matrix, $\boldsymbol{\delta}[n] = \boldsymbol{0}$ is a zero matrix for $n \neq 0$. We prove the equivalence more generally in Proposition C.9. However, this approach cannot enforce exact orthogonality and in practice requires hyperparameter search for a proper regularizer coefficient.

**Proposition C.9** (Parseval's theorem for ridge regularization). *Given a sequence of matrices $\boldsymbol{h} = \{\boldsymbol{h}[n], n \in \mathbb{Z}\}$, the following four expressions are equivalent:*

$$\frac{1}{2\pi} \int_{-\pi}^{\pi} \left\| \boldsymbol{H}^{[R]}(e^{j\omega})^\dagger \boldsymbol{H}^{[R]}(e^{j\omega}) - \boldsymbol{I} \right\|^2 d\omega, \qquad \text{(C.69a)}$$

$$\sum_{i \in \mathbb{Z}} \left\| \sum_{n \in \mathbb{Z}} \boldsymbol{h}[n]^\top \boldsymbol{h}[n - Ri] - \boldsymbol{\delta}[i] \right\|^2, \qquad \text{(C.69b)}$$

$$\frac{1}{2\pi} \int_{-\pi}^{\pi} \left\| \boldsymbol{H}^{[R]}(e^{j\omega}) \boldsymbol{H}^{[R]}(e^{j\omega})^\dagger - \boldsymbol{I} \right\|^2 d\omega, \qquad \text{(C.69c)}$$

$$\sum_{i \in \mathbb{Z}} \left\| \sum_{n \in \mathbb{Z}} \boldsymbol{h}[n] \boldsymbol{h}[n - Ri]^\top - \boldsymbol{\delta}[i] \right\|^2, \qquad \text{(C.69d)}$$

*where $\| \cdot \|$ denotes the Frobenius norm of a matrix.*

*Proof of Proposition C.9.* We first prove the equivalence between Equations (C.69a) and (C.69c).

$$\left\| \boldsymbol{H}^{[R]}(e^{\mathrm{j}\omega})^{\dagger} \boldsymbol{H}^{[R]}(e^{\mathrm{j}\omega}) - \boldsymbol{I} \right\|^2$$

$$= \mathsf{tr}\left( \left( \boldsymbol{H}^{[R]}(e^{\mathrm{j}\omega})^{\dagger} \boldsymbol{H}^{[R]}(e^{\mathrm{j}\omega}) - \boldsymbol{I} \right)^{\dagger} \left( \boldsymbol{H}^{[R]}(e^{\mathrm{j}\omega})^{\dagger} \boldsymbol{H}^{[R]}(e^{\mathrm{j}\omega}) - \boldsymbol{I} \right) \right) \tag{C.70}$$

$$= \mathsf{tr}\left( \boldsymbol{H}^{[R]}(e^{\mathrm{j}\omega})^{\dagger} \boldsymbol{H}^{[R]}(e^{\mathrm{j}\omega}) \boldsymbol{H}^{[R]}(e^{\mathrm{j}\omega})^{\dagger} \boldsymbol{H}^{[R]}(e^{\mathrm{j}\omega}) \right) - 2\mathsf{tr}\left( \boldsymbol{H}^{[R]}(e^{\mathrm{j}\omega})^{\dagger} \boldsymbol{H}^{[R]}(e^{\mathrm{j}\omega}) \right) + \boldsymbol{I} \tag{C.71}$$

$$= \mathsf{tr}\left( \boldsymbol{H}^{[R]}(e^{\mathrm{j}\omega}) \boldsymbol{H}^{[R]}(e^{\mathrm{j}\omega})^{\dagger} \boldsymbol{H}^{[R]}(e^{\mathrm{j}\omega}) \boldsymbol{H}^{[R]}(e^{\mathrm{j}\omega})^{\dagger} \right) - 2\mathsf{tr}\left( \boldsymbol{H}^{[R]}(e^{\mathrm{j}\omega}) \boldsymbol{H}^{[R]}(e^{\mathrm{j}\omega})^{\dagger} \right) + \boldsymbol{I} \tag{C.72}$$

$$= \mathsf{tr}\left( \left( \boldsymbol{H}^{[R]}(e^{\mathrm{j}\omega}) \boldsymbol{H}^{[R]}(e^{\mathrm{j}\omega})^{\dagger} - \boldsymbol{I} \right)^{\dagger} \left( \boldsymbol{H}^{[R]}(e^{\mathrm{j}\omega}) \boldsymbol{H}^{[R]}(e^{\mathrm{j}\omega})^{\dagger} - \boldsymbol{I} \right) \right) \tag{C.73}$$

$$= \left\| \boldsymbol{H}^{[R]}(e^{\mathrm{j}\omega}) \boldsymbol{H}^{[R]}(e^{\mathrm{j}\omega})^{\dagger} - \boldsymbol{I} \right\|^2, \tag{C.74}$$

where Equations (C.70) and (C.74) make use of $\|\boldsymbol{A}\|^2 = \mathsf{tr}(\boldsymbol{A}^{\dagger}\boldsymbol{A})$, Equations (C.71) and (C.73) are due to the linearity of $\mathsf{tr}(\cdot)$, and Equation (C.72) utilizes $\mathsf{tr}(\boldsymbol{AB}) = \mathsf{tr}(\boldsymbol{BA})$.

Next, we prove the equivalence between Equations (C.69a) and (C.69b).

$$\boldsymbol{H}^{[R]}(e^{\mathrm{j}\omega})^{\dagger} \boldsymbol{H}^{[R]}(e^{\mathrm{j}\omega}) - \boldsymbol{I} = \sum_{r \in [R]} \boldsymbol{H}^{r|R}(e^{\mathrm{j}\omega})^{\dagger} \boldsymbol{H}^{r|R}(e^{\mathrm{j}\omega}) - \boldsymbol{I} \tag{C.75}$$

$$= \sum_{r \in [R]} \sum_{i \in \mathbb{Z}} \left( \sum_{m \in \mathbb{Z}} \boldsymbol{h}^{r|R}[m]^{\top} \boldsymbol{h}^{r|R}[m - i] - \boldsymbol{\delta}[i] \right) e^{-\mathrm{j}\omega i} \tag{C.76}$$

$$= \sum_{r \in [R]} \sum_{i \in \mathbb{Z}} \left( \sum_{m \in \mathbb{Z}} \boldsymbol{h}[Rm + r]^{\top} \boldsymbol{h}[R(m - i) + r] - \boldsymbol{\delta}[i] \right) e^{-\mathrm{j}\omega i} \tag{C.77}$$

$$= \sum_{i \in \mathbb{Z}} \left( \sum_{r \in [R]} \sum_{m \in \mathbb{Z}} \boldsymbol{h}[Rm + r]^{\top} \boldsymbol{h}[Rm + r - Ri] - \boldsymbol{\delta}[i] \right) e^{-\mathrm{j}\omega i} \tag{C.78}$$

$$= \sum_{i \in \mathbb{Z}} \left( \sum_{n \in \mathbb{Z}} \boldsymbol{h}[n]^{\top} \boldsymbol{h}[n - Ri] - \boldsymbol{\delta}[i] \right) e^{-\mathrm{j}\omega i}, \tag{C.79}$$

where Equation (C.75) follows from the definition of polyphase matrix in Equation (C.15). Equation (C.76) uses a number of properties of Fourier transform: a Hermitian in the spectral domain is a transposed reflection in the spatial domain, a frequency-wise multiplication in the spectral domain is a convolution in the spatial domain, and an identical mapping in the spectral domain is an impulse sequence in the spatial domain. Equation (C.77) follows from the definition of polyphase components in Equation (C.7), and Equation (C.79) makes a change of variables $n = Rm + r$. In summary, we show that the LHS (denoted $\boldsymbol{D}(e^{\mathrm{j}\omega})$) is a Fourier transform of the RHS (denoted as $\boldsymbol{d}[i]$):

$$\underbrace{\boldsymbol{H}^{[R]}(e^{\mathrm{j}\omega})^{\dagger} \boldsymbol{H}^{[R]}(e^{\mathrm{j}\omega}) - \boldsymbol{I}}_{\boldsymbol{D}(e^{\mathrm{j}\omega})} = \sum_{i \in \mathbb{Z}} \underbrace{\left( \sum_{n \in \mathbb{Z}} \boldsymbol{h}[n]^{\top} \boldsymbol{h}[n - Ri] - \boldsymbol{\delta}[n] \right)}_{\boldsymbol{d}[i]} e^{-\mathrm{j}\omega i}. \tag{C.80}$$

Applying Parseval's theorem (Theorem B.3) to the sequence $\boldsymbol{d} = \{\boldsymbol{d}[i], i \in \mathbb{Z}\}$, we have

$$\frac{1}{2\pi} \int_{-\pi}^{\pi} \left\| \boldsymbol{H}^{[R]}(e^{\mathrm{j}\omega})^{\dagger} \boldsymbol{H}^{[R]}(e^{\mathrm{j}\omega}) - \boldsymbol{I} \right\|^2 d\omega = \sum_{i \in \mathbb{Z}} \left\| \sum_{n \in \mathbb{Z}} \boldsymbol{h}[n]^{\top} \boldsymbol{h}[n - Ri] - \boldsymbol{\delta}[i] \right\|^2, \tag{C.81}$$

which proves the equivalence between Equations (C.69a) and (C.69b). With almost identical arguments, we can prove the equivalence between Equations (C.69c) and (C.69d), that is

$$\frac{1}{2\pi} \int_{-\pi}^{\pi} \left\| \boldsymbol{H}^{[R]}(e^{\mathrm{j}\omega}) \boldsymbol{H}^{[R]}(e^{\mathrm{j}\omega})^{\dagger} - \boldsymbol{I} \right\|^2 d\omega = \sum_{i \in \mathbb{Z}} \left\| \sum_{n \in \mathbb{Z}} \boldsymbol{h}[n] \boldsymbol{h}[n - Ri]^{\top} - \boldsymbol{\delta}[i] \right\|^2, \tag{C.82}$$

which completes the proof. □

Table 5: Computational complexities of different approaches for orthogonal convolutions.

| Approach | Computational Complexity |
|---|---|
| Normal | $O\left(L^2 N^2 C^2\right)$ |
| SVCM | $O\left(L^2 N^2 C^2 + N^2 C^3\right)$ |
| Ortho-Reg | $O\left(L^2 N^2 C^2 + L^4 C^2\right)$ |
| CayleyConv | $O\left(N^2 \log(N)C^2 + N^2 C^3\right)$ |
| SOC | $O\left(K L^2 N^2 C^2\right)$ |
| BCOP | $O\left(L^2 N^2 C^2 + K L^2 C^3\right)$ |
| **SC-Fac** | $O\left(L^2 N^2 C^2 + L^2 C^3\right)$ |

In Table 5, we compare the computational complexities (forward pass) of orthogonal convolutions against normal convolution. For simplicity, we assume the feature maps have size $N \times N$, the convolution filters have size $L \times L$, and the maximum of input/output channels is $C$. We use $K$ to denote the number of iterations for Björck's algorithm in BCOP, or Taylor's series order in SOC.

- For SC-Fac, BCOP, SVCM, or Ortho-Reg, the first term $O(L^2 N^2 C^2)$ is the base cost of a normal convolution, and the second term is the overhead for reconstruction, projection, or regularization. Note that the overhead in SC-Fac is comparable to Ortho-Reg, which is lower than SVCM or BCOP. If $C < N^2$, the overhead in SC-Fac is negligible compared to the base cost.
- For CayleyConv, the first term $O(N^2 \log(N)C^2)$ is the cost for fast Fourier transform (FFT), and the second term $O(N^2 C^3)$ is the cost for matrix inversion for all frequencies. The cost of SOC is exactly $K$ times as a normal convolution. The computational complexities of CayleyConv and SOC are significantly higher than the one of normal convolution.
- For those approaches whose filters can be explicitly obtained (SC-Fac, BCOP, SVCM, and Ortho-Reg), the inference time of an orthogonal convolution is no different from a normal convolution, i.e., $O(L^2 N^2 C^2)$. On the other hand, for those approaches whose filters are implicitly defined (CayleyConv and SOC), the inference time is the same as the forward pass in training.

## C.4 CONSTRAINED OPTIMIZATION OVER ORTHOGONAL MATRICES

In Theorem C.7, we have shown how orthogonal matrices characterize paraunitary systems. Our remaining goal, therefore, is to parameterize orthogonal matrices using unconstrained parameters. In this part, we review popular parameterization methods: *Householder reflections* (Mhammedi et al., 2017; Mathiasen et al., 2020), *Givens rotations* (Dorobantu et al., 2016; Jing et al., 2017), *Björck orthogonalization* (Anil et al., 2019), *Cayley transform* (Helfrich et al., 2018; Maduranga et al., 2019), and *exponential map* (Lezcano-Casado & Martínez-Rubio, 2019; Lezcano Casado, 2019).

**(1) Householder reflections (Mhammedi et al., 2017; Mathiasen et al., 2020).** The parameterization represents an orthogonal matrix using a product of Householder matrices. Given $N \in \mathbb{N}$ and $n \in \{1, \cdots, N\}$, we define $\boldsymbol{H}^{(n)}$ as a mapping from a vector $\boldsymbol{v} \in \mathbb{R}^n$ (a scalar $v \in \mathbb{R}$ for $n = 1$) to a block-diagonal matrix $\boldsymbol{H}^{(n)}(\boldsymbol{v}) \in \mathbb{R}^{N \times N}$:

$$\boldsymbol{H}^{(n)}(\boldsymbol{v}) = \begin{bmatrix} \boldsymbol{I}_{N-n} & \\ & \boldsymbol{I}_n - 2\frac{\boldsymbol{v}\boldsymbol{v}^\top}{\|\boldsymbol{v}\|^2} \end{bmatrix}, \ \boldsymbol{H}^{(1)}(v) = \begin{bmatrix} \boldsymbol{I}_{N-1} & \\ & v \end{bmatrix}, \tag{C.83}$$

where $\boldsymbol{I}_n \in \mathbb{R}^{n \times n}$ denotes an identity matrix. For $n \geq 2$, $\boldsymbol{H}^{(n)}(\boldsymbol{v})$ is the Householder matrix that represents the reflection with respect to the hyperplane orthogonal to $\mathsf{concat}(\boldsymbol{0}_{N-n}, \boldsymbol{v}) \in \mathbb{R}^N$ and passing through the origin. For $n = 1$, the scalar $v$ takes values $\{-1, +1\}$, which makes $\boldsymbol{H}^{(1)}(v)$ either an identity matrix (for $v = 1$) or a Householder matrix (for $v = -1$). This method parameterizes an orthogonal matrix as a product of $\boldsymbol{H}^{(n)}$'s:

$$\boldsymbol{U} = \boldsymbol{H}^{(n)}(\boldsymbol{v}^{(n)}) \cdots \boldsymbol{H}^{(2)}(\boldsymbol{v}^{(2)}) \boldsymbol{H}^{(1)}(v^{(1)}), \tag{C.84}$$

where each $\boldsymbol{v}^{(n)} \in \mathbb{R}^n$ is a learnable vector for $n \geq 2$, and $v^{(1)}$ is a fixed constant that is generated at initialization and fixed afterward. Observing that Equation (C.84) is serial (unfriendly to parallel computing), Mathiasen et al. (2020) proposes to increase its parallelism using *WY transform*.

**(2) Givens rotations (Dorobantu et al., 2016; Jing et al., 2017).** The parameterization represents a special orthogonal matrix using a product of Given rotations matrices. Given $N \in \mathbb{N}$ and $i, j \in \{1, \cdots, N\}$ with $i \neq j$, we define $\boldsymbol{G}^{(i,j)}$ as a mapping from an angle $\theta \in \mathbb{R}$ to the corresponding rotation matrix $\boldsymbol{G}^{(i,j)} \in \mathbb{R}^{N \times N}$:

$$
\boldsymbol{G}^{(i,j)}(\theta) = \begin{bmatrix}
1 & \cdots & 0 & \cdots & 0 & \cdots & 0 \\
\vdots & \ddots & \vdots & & \vdots & & \vdots \\
0 & \cdots & \cos\theta & \cdots & -\sin\theta & \cdots & 0 \\
\vdots & & \vdots & \ddots & \vdots & & \vdots \\
0 & \cdots & \sin\theta & \cdots & \cos\theta & \cdots & 0 \\
\vdots & & \vdots & & \vdots & \ddots & \vdots \\
0 & \cdots & 0 & \cdots & 0 & \cdots & 1
\end{bmatrix}. \tag{C.85}
$$

This method parameterizes a special orthogonal matrix $\boldsymbol{U}$ as a product of $\boldsymbol{G}^{(i,j)}$'s:

$$
\boldsymbol{U} = \boldsymbol{G}^{(0,1)}(\theta^{(0,1)}) \, \boldsymbol{G}^{(0,2)}(\theta^{(0,2)}) \, \boldsymbol{G}^{(1,2)}(\theta^{(1,2)}) \cdots \boldsymbol{G}^{(N-2,N-1)}(\theta^{(N-2,N-1)}), \tag{C.86}
$$

where $\theta^{(i,j)}$'s are $N(N-1)$ unconstrained parameters. Since the determinant of each rotation matrix is $+1$, their product $\boldsymbol{U}$ also has $+1$ determinant, thus is a special orthogonal matrix. Again, Equation (C.86) is highly serial, thus both Dorobantu et al. (2016) and Jing et al. (2017) propose to group $N/2$ Given rotations into a *packed rotation* to increase the parallelism.

**(2) Björck orthogonalization (Anil et al., 2019).** The algorithm was first introduced in Björck & Bowie (1971) to compute the closest orthogonal matrix of a given matrix. Given an initial matrix $\boldsymbol{U}_0$, this algorithm *iteratively* approaches its closest orthogonal matrix as:

$$
\boldsymbol{U}_{k+1} = \boldsymbol{U}_k \left( \boldsymbol{I} + \frac{1}{2}\boldsymbol{P}_k + \cdots + (-1)^p \begin{pmatrix} -\frac{1}{2} \\ p \end{pmatrix} \boldsymbol{P}_k^p \right), \forall k \in [K], \tag{C.87}
$$

where $K$ is the iterative steps, $\boldsymbol{P}_k = \boldsymbol{I} - \boldsymbol{U}_k^\top \boldsymbol{U}_k$, and $p$ controls the trade-off between efficiency and accuracy at each step. When the algorithm is used for parameterization, it maps an unconstrained matrix $\boldsymbol{U}_0$ to an approximately orthogonal matrix $\boldsymbol{U}_K$ in $K$ steps. Although Björck parameterization is complete (since any orthogonal matrix $\boldsymbol{Q}$ can be represented by $\boldsymbol{U}_0 = \boldsymbol{Q}$), it is inexact due to the iterative approximation.

**(3) Cayley transform (Helfrich et al., 2018; Maduranga et al., 2019).** The transform provides a *bijective* parameterization of orthogonal matrices *without* $-1$ *eigenvalue* with skew-symmetric matrices (i.e., $\boldsymbol{A}^\top = -\boldsymbol{A}$)

$$
\boldsymbol{U} = (\boldsymbol{I} - \boldsymbol{A})(\boldsymbol{I} + \boldsymbol{A})^{-1}, \tag{C.88}
$$

where the skew-symmetric matrix $\boldsymbol{A}$ is represented by its upper-triangle entries. Since orthogonal matrices with $-1$ eigenvalue are out of consideration, the parameterization is incomplete. Helfrich et al. (2018); Maduranga et al. (2019) overcome this difficulty by a *scaled Cayley transform*:

$$
\boldsymbol{U} = \boldsymbol{D}(\boldsymbol{I} - \boldsymbol{A})(\boldsymbol{I} + \boldsymbol{A})^{-1}, \tag{C.89}
$$

where $\boldsymbol{D}$ is a diagonal matrix with $\pm 1$ non-zero entries, which is (randomly) generated at initialization and fixed during training.

**(4) Exponential map (Lezcano-Casado & Martínez-Rubio, 2019; Lezcano Casado, 2019).** The mapping provides a *surjective* parameterization of all special orthogonal matrices (with $+1$ determinant) with using skew-symmetry matrices (i.e., $\boldsymbol{A}^\top = -\boldsymbol{A}$).

$$
\boldsymbol{U} = \exp(\boldsymbol{A}) \triangleq \sum_{k=0}^{\infty} \frac{\boldsymbol{A}^k}{k!} = \boldsymbol{I} + \boldsymbol{A} + \frac{1}{2}\boldsymbol{A}^2 + \cdots, \tag{C.90}
$$

where the infinite sum can be computed exactly up to machine-precision (Higham, 2009). Lezcano-Casado & Martínez-Rubio (2019) derives an efficient backpropagation algorithm for the mapping, making it an exact and efficient parameterization in neural networks. To support a complete parameterization for all orthogonal matrices, Lezcano Casado (2019) extends the mapping as:

$$
\boldsymbol{U} = \boldsymbol{Q} \exp(\boldsymbol{A}), \tag{C.91}
$$

where $\boldsymbol{Q}$ is an orthogonal matrix, which is generated at initialization and fixed during training.

In principle, we can use any of these approaches to parameterize the orthogonal matrices. In this work, we choose *exponential map* due to its exactness, efficiency, and completeness.

## D LEARNING DEEP ORTHOGONAL NETWORKS WITH LIPSCHITZ BOUNDS

In this section, we first discuss the properties of GroupSort and Lipschitz networks. Subsequently, we prove Proposition 4.1, which exhibits two approaches to construct Lipschitz residual blocks. Lastly, we prove in Proposition D.1 when a paraunitary system (represented by a complete factorization as in Theorem C.7) reduces to an orthogonal matrix. The reduction allows us to apply the initialization methods for orthogonal matrices to paraunitary systems.

**GroupSort and orthogonality.** The GroupSort activation separates inputs into groups and sorts each group into ascending order (Anil et al., 2019). It guarantees two properties: **(1) The activation is norm preserving in the forward pass** — a sorting function does not change the norm of any input vector. **(2) The activation is gradient norm preserving in the backward pass** — a sorting function acts as a permutation, which is orthogonal. GroupSort with the group size of two is a spacial case that we use in the paper followed by (Chernodub & Nowicki, 2016; Anil et al., 2019).

A Lipschitz network with orthogonal layers and GroupSort activations is *locally orthogonal*. Given any input $\mathbf{x}$ to the network, there exists a neighborhood $N(x, r)$ with radius $r$ such that the sorting order in each activation does not change. In this neighborhood $N(x, r)$, each GroupSort layer operates as a constant permutation (thus orthogonal); consequently, the whole network is locally orthogonal.

**Lipschitz residual blocks.** In Proposition 4.1, we prove the Lipschitzness of two types of residual blocks, one based on additive skip-connection and another based on concatenative one (See Figure 6).

*Proof for Proposition 4.1.* We first prove the Lipschitzness for the additive residual block $f : f(\boldsymbol{x}) \triangleq \alpha f^1(\boldsymbol{x}) + (1 - \alpha)f^2(\boldsymbol{x})$. Let $\boldsymbol{x}$, $\boldsymbol{x}'$ be two inputs to $f$ and $f(\boldsymbol{x})$, $f(\boldsymbol{x}')$ be their outputs, we have

$$\|f(\boldsymbol{x}') - f(\boldsymbol{x})\| = \left\|\left(\alpha f^1(\boldsymbol{x}') + (1 - \alpha)f^2(\boldsymbol{x}')\right) - \left(\alpha f^1(\boldsymbol{x}) + (1 - \alpha)f^2(\boldsymbol{x})\right)\right\| \tag{D.1}$$

$$= \left\|\alpha\left(f^1(\boldsymbol{x}') - f^1(\boldsymbol{x})\right) + (1 - \alpha)\left(f^2(\boldsymbol{x}') - f^2(\boldsymbol{x})\right)\right\| \tag{D.2}$$

$$\leq \alpha\left\|f^1(\boldsymbol{x}') - f^1(\boldsymbol{x})\right\| + (1 - \alpha)\left\|f^2(\boldsymbol{x}') - f^2(\boldsymbol{x})\right\| \tag{D.3}$$

$$\leq \alpha L\|\boldsymbol{x}' - \boldsymbol{x}\| + (1 - \alpha)L\|\boldsymbol{x}' - \boldsymbol{x}\| \tag{D.4}$$

$$= L\|\boldsymbol{x}' - \boldsymbol{x}\|, \tag{D.5}$$

where Equation (D.3) makes uses of the triangle inequality, and Equation (D.4) is due to the $L$-Lipschitzness of both $f^1$, $f^2$. Therefore, we have shown that $\|f(\boldsymbol{x}') - f(\boldsymbol{x})\| \leq L\|\boldsymbol{x}' - \boldsymbol{x}\|$.

Similarly, we prove the Lipschitzness for the concatenative residual block $g : g(\boldsymbol{x}) \triangleq \boldsymbol{P}\left[g^1(\boldsymbol{x}^1); g^2(\boldsymbol{x}^2)\right]$. Let $\boldsymbol{x}$, $\boldsymbol{x}'$ be two inputs to $g$ and $g(\boldsymbol{x})$, $g(\boldsymbol{x}')$ be their outputs, we have

$$\|g(\boldsymbol{x}') - g(\boldsymbol{x})\|^2 = \left\|\boldsymbol{P}\left([g^1(\boldsymbol{x}^{1'}); g^2(\boldsymbol{x}^{2'})] - [g^1(\boldsymbol{x}^1); g^2(\boldsymbol{x}^2)]\right)\right\| \tag{D.6}$$

$$= \left\|[g^1(\boldsymbol{x}'); g^2(\boldsymbol{x}')] - [g^1(\boldsymbol{x}); g^2(\boldsymbol{x})]\right\|^2 \tag{D.7}$$

$$= \left\|g^1(\boldsymbol{x}^{1'}) - g^1(\boldsymbol{x})\right\|^2 + \left\|g^2(\boldsymbol{x}^{2'}) - g^2(\boldsymbol{x}^2)\right\|^2 \tag{D.8}$$

$$\leq L^2\left\|\boldsymbol{x}^{1'} - \boldsymbol{x}^1\right\|^2 + L^2\left\|\boldsymbol{x}^{2'} - \boldsymbol{x}^2\right\|^2 \tag{D.9}$$

$$= L^2\left\|[\boldsymbol{x}^{1'}; \boldsymbol{x}^{2'}] - [\boldsymbol{x}^1; \boldsymbol{x}^2]\right\|^2 \tag{D.10}$$

$$= L^2\|\boldsymbol{x}' - \boldsymbol{x}\|^2, \tag{D.11}$$

where Equation (D.7) utilizes $\|\boldsymbol{P}\boldsymbol{x}\| = \|\boldsymbol{x}\|, \forall \boldsymbol{x}$, and Equation (D.9) is due to the $L$-Lipschitzness of $g^1$, $g^2$. The equations above implies that $\|g(\boldsymbol{x}') - g(\boldsymbol{x})\| \leq L\|\boldsymbol{x}' - \boldsymbol{x}\|$. □

**Reduction of a paraunitary system to an orthogonal matrix.** In Proposition D.1, we prove a special case when a paraunitary system reduces to an orthogonal matrix. The reduction allows us to apply the initialization methods for orthogonal matrices to paraunitary systems.

**Proposition D.1** (Reduction of a paraunitary matrix to an orthogonal matrix)**.** *Suppose a paraunitary system $\boldsymbol{H}(z)$ takes the complete factorization in Equation (C.55), and assume $\underline{L} = \overline{L}$ with $\boldsymbol{U}^{(-\ell)} = \boldsymbol{Q}\boldsymbol{U}^{(\ell)}$ for all $\ell$, then the paraunitary matrix $\boldsymbol{H}(z)$ reduces to an orthogonal matrix $\boldsymbol{Q}$,*

$$\boldsymbol{H}(z) = \boldsymbol{V}(z; \boldsymbol{U}^{(-\underline{L})}) \cdots \boldsymbol{V}(z; \boldsymbol{U}^{(-1)})\boldsymbol{Q}\boldsymbol{V}(z^{-1}; \boldsymbol{U}^{(1)}) \cdots \boldsymbol{V}(z^{-1}; \boldsymbol{U}^{(\overline{L})}) = \boldsymbol{Q}. \tag{D.12}$$

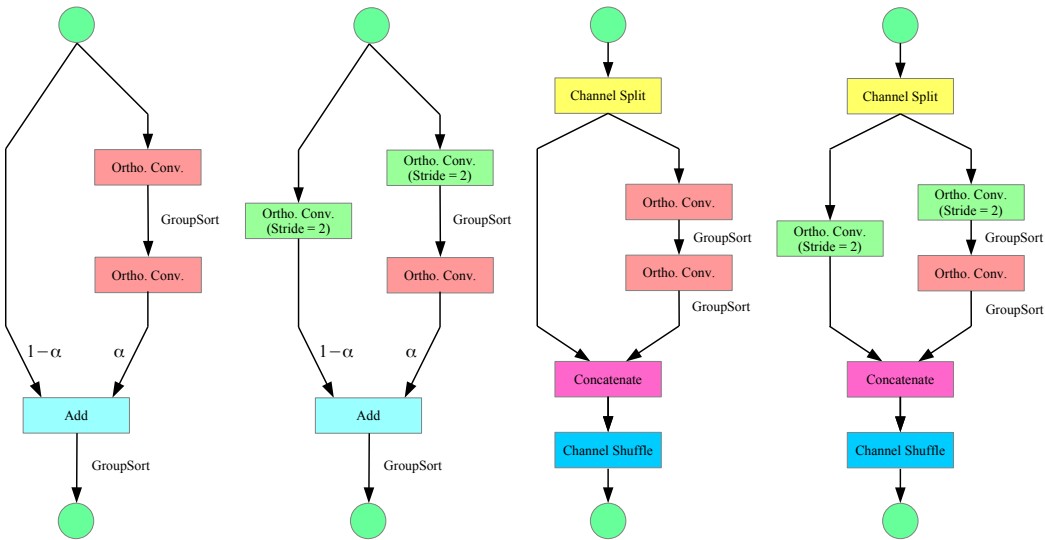

(a) Basic additive block.    (b) Strided additive block.    (c) Basic shuffling block.    (d) Strided shuffling block.

Figure 6: **Variants of residual blocks.** In our experiments, we combine **(a) & (b)** to construct an *orthogonal ResNet*, and **(c) & (d)** to construct an *orthogonal ShuffleNet*. In Proposition 4.1, we prove the Lipschitzness of these building blocks. Since composition of Lipschitz functions is still Lipschitz, it implies that a network constructed by these building blocks is also Lipschitz.

*Proof for Proposition D.1.* In order to prove Equation (D.12), it suffice to show that

$$V(z; U^{(-\ell)})QV(z^{-1}; U^{(\ell)}) = Q, \tag{D.13}$$

and Equation (D.12) will reduce recursively to the orthogonal matrix $Q$. For simplicity, we rewrite $U^{(-\ell)}$ as $L$ and $U^{(\ell)}$ as $R$, by which we have $L = QR$ (or $R = Q^\top L$) and we aim to prove $V(z; L)QV(z^{-1}; R) = Q$. By the definition of $V(z; \cdot)$ in Equation (C.54), we expand it as

$$V(z; L)QV(z^{-1}; R) = \left[ \left( I - LL^\top \right) + LL^\top z \right] Q \left[ \left( I - RR^\top \right) + RR^\top z^{-1} \right] =$$
$$\underbrace{LL^\top Q(I - RR^\top)}_{c[-1]} z + \underbrace{(I - LL^\top)Q(I - RR^\top) + LL^\top QRR^\top}_{c[0]} + \underbrace{(I - LL^\top)QRR^\top}_{c[1]} z^{-1} \quad \text{(D.14)}$$

Therefore, we will need to show that $c[-1] = 0$, $c[1] = 0$ and $c[0] = Q$.

We first show that both $c[-1]$ for $z$ and $c[1]$ for $z^{-1}$ are zero matrices.

$$c[-1] = LL^\top Q \left( I - RR^\top \right) \tag{D.15}$$
$$= LL^\top Q - LL^\top QRR^\top \tag{D.16}$$
$$= L(Q^\top L)^\top - L(Q^\top L)^\top RR^\top \tag{D.17}$$
$$= LR^\top - L(R^\top R)R^\top \tag{D.18}$$
$$= LR^\top - LR^\top = 0, \tag{D.19}$$

$$c[1] = (I - LL^\top)QRR^\top \tag{D.20}$$
$$= QRR^\top - LL^\top QRR^\top \tag{D.21}$$
$$= (QR)R^\top - LL^\top (QR)R^\top \tag{D.22}$$
$$= LR^\top - L(L^\top L)R^\top \tag{D.23}$$
$$= LR^\top - LR^\top = 0. \tag{D.24}$$

Table 6: **Comparisons of various initialization methods** on WideResNet (kernel size 5).

| Initialization | WideResNet10-10 | | WideResNet22-10 | |
|---|---|---|---|---|
| | Clean (%) | PGD (%) | Clean (%) | PGD (%) |
| uniform | **83.58** | 73.20 | 87.55 | 75.71 |
| torus | 82.40 | 72.50 | **88.12** | 75.43 |
| permutation | 83.18 | 73.16 | 87.82 | **76.46** |
| identical | 83.29 | **73.49** | 87.82 | 75.49 |

Lastly, we show that the constant coefficient $c[0]$ is equal to $Q$.

$$c[0] = (I - LL^\top)Q(I - RR^\top) + LL^\top QRR^\top \tag{D.25}$$

$$= Q - LL^\top Q - QRR^\top + 2LL^\top QRR^\top \tag{D.26}$$

$$= Q - LR^\top - LR^\top + 2LR^\top = Q \tag{D.27}$$

which completes the proof. $\qquad\square$

# E  SUPPLEMENTARY MATERIALS FOR EXPERIMENTS

## E.1  EXPERIMENTAL SETUP

**Network architectures.** For fair comparisons, we follow the architectures by Trockman & Kolter (2021) for KW-Large, ResNet9, WideResNet10-10 (i.e., shallow networks). We set the group size for GroupSort activations as 2 in all experiments. For networks deeper than 10 layers, we implement their architectures modifying from the Pytorch official implementation of ResNet. It is crucial to replace the global pooling before fully-connected layers with an average pooling with a window size of 4. For the average pooling, we multiply the output with the window size to maintain its 1-Lipschitzness. Other architectures, including ShuffleNet and plain convolutional network (ConvNet), are further modified from the ResNet, where only the skip-connections are changed or removed. We use the widen factor to indicate the channel number: we set the number of channels at each layer as base channels multiplied by the widen factor. The base channels are $16, 32, 64$ for three groups of residual blocks. More details of the ResNet architecture can be found in the official Pytorch implementation.[1]

**Learning strategies.** We use the CIFAR-10 dataset for all our experiments. We normalize all input images to $[0, 1]$ followed by standard augmentation, including random cropping and horizontal flipping. We use the Adam optimizer with a maximum learning rate of $10^{-2}$ coupled with a piece-wise triangular learning rate scheduler. We initialize all our SC-Fac layers as permutation matrices: (1) we select the number of columns for each pair $U^{(\ell)}, U^{(-\ell)}$ uniformly from $\{1, \cdots, T\}$ at initialization (the number is fixed during training); (2) for $\ell > 0$, we sample the entries in $U^{(\ell)}$ uniformly with respect to the Haar measure; (3) for $\ell < 0$, we set $U^{(-\ell)} = QU^{(\ell)}$ according to Proposition D.1.

## E.2  ADDITIONAL EMPIRICAL RESULTS ON HYPER-PARAMETERS SELECTION

**Multi-class hinge loss.** Following previous works on Lipschitz networks (Anil et al., 2019; Li et al., 2019b; Trockman & Kolter, 2021), we adopt the multi-class hinge loss in training. For each model, we perform a grid search on different margins $\epsilon_0 \in \{1 \times 10^{-3}, 2 \times 10^{-3}, 5 \times 10^{-3}, 1 \times 10^{-2}, 2 \times 10^{-2}, 5 \times 10^{-2}, 0.1, 0.2, 0.5\}$ and report the best performance in terms of robust accuracy. Notice that the margin $\epsilon_0$ controls the trade-off between clean and robust accuracy, as shown in Figure 7.

**Initialization methods.** In Proposition D.1, we have shown how to initialize our orthogonal convolutional layers as orthogonal matrices. In Table 6, we perform a study on different initialization methods, including identical, permutation, uniform, and torus (Henaff et al., 2016; Helfrich et al., 2018). We find that permutation works the best for WideResNet22-10, while all methods are similar in shallower WideResNet10-10. Therefore, we use permutation initialization for all other experiments.

---

[1] https://github.com/pytorch/vision/blob/master/torchvision/models/

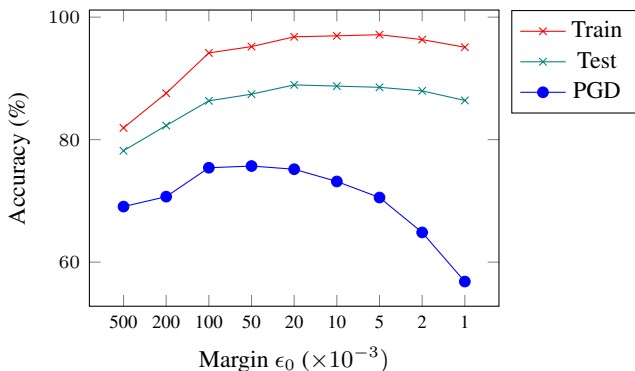

Figure 7: **Effect of the Lipschitz margin $\epsilon_0$ for WideResNet22-10**. It shows a trade-off between clean and robust accuracy with different margins for multi-class hinge loss. As shown, the training and test accuracy become higher with larger margin, but the robust accuracy decreases after $\epsilon_0 = 0.1$.

Table 7: **Comparison of different depth and width** on WideResNet (kernel size 5). Some numbers are missing due to the large memory requirement (on Tesla V100 32G). The notation width factor indicates (channels = base channels $\times$ factor).

|  | 10 layers | | | | | | | | | |
|---|---|---|---|---|---|---|---|---|---|---|
| Width | 1 | 3 | 6 | 8 | 10 | 1 | 3 | 6 | 8 | 10 |
|  | Clean (%) | | | | | PGD with $\epsilon = 36/255$ (%) | | | | |
| Ours | 79.96 | 84.17 | 84.96 | 84.61 | 84.09 | 65.92 | 69.70 | 72.18 | 72.51 | 74.29 |
| Cayley | 77.88 | 82.14 | 82.56 | **85.53** | 85.01 | 66.65 | 73.06 | 74.33 | 75.66 | 76.13 |
| RKO | 81.37 | 83.55 | 84.67 | 85.18 | 84.62 | 70.55 | 74.44 | 76.41 | 76.65 | **77.02** |
|  | 22 layers | | | | | | | | | |
| Width | 1 | 3 | 6 | 8 | 10 | 1 | 3 | 6 | 8 | 10 |
|  | Clean (%) | | | | | PGD with $\epsilon = 36/255$ (%) | | | | |
| Ours | 79.90 | 82.22 | 87.21 | **88.10** | 87.82 | 67.95 | 70.88 | 74.30 | 75.12 | **76.46** |
| Cayley | 79.11 | 84.82 | 85.85 | - | - | 69.79 | 65.61 | 74.81 | - | - |
| RKO | 82.71 | 84.19 | 84.33 | 84.55 | - | 72.40 | 74.36 | 75.66 | 76.41 | - |
|  | 34 layers | | | | | | | | | |
| Width | 1 | 3 | 6 | 8 | 10 | 1 | 3 | 6 | 8 | 10 |
|  | Clean (%) | | | | | PGD with $\epsilon = 36/255$ (%) | | | | |
| Ours | 81.24 | 88.17 | **88.92** | - | - | 69.21 | 71.85 | **75.09** | - | - |
| Cayley | 82.46 | 84.29 | - | - | - | 71.27 | 74.73 | - | - | - |
| RKO | 81.51 | 83.24 | 83.92 | - | - | 71.38 | 73.84 | 75.03 | - | - |

**Network depth and width.** Exact orthogonality is criticized for harming the expressive power of neural networks, and we find that increasing network depth/width can partially compensate for such loss. In Table 7, we perform a study on the impact of network depth/width on the predictive performance. As shown, deeper/wider architectures consistently improve both the clean and robust accuracy for our implementation. However, the best robust accuracy is achieved by a 22-layer network since we can afford a wide architecture for 34-layer architecture.

**Comparison against normal convolutional networks.** In Table 8, We perform a comparison between our orthogonal networks and normal convolutional networks. Their architecture are identical except for the activation function (GroupSort for ours and ReLU for normal convolutional networks). Since batch normalization is common in normal convolutional networks but not in Lipschitz networks,

Table 8: **Comparison of orthogonal convolutions and normal convolutions** on WideResNet (kernel size 5). The notation width factor indicates (channels = base channels $\times$ factor).

| | | 22 layers | | | | | | | | |
|---|---|---|---|---|---|---|---|---|---|---|
| Width | 1 | 3 | 6 | 8 | 10 | 1 | 3 | 6 | 8 | 10 |
| | | | Clean (%) | | | | | PGD with $\epsilon = 36/255$ (%) | | |
| **Ortho. (SC-Fac)** | 79.90 | 82.22 | 87.21 | 88.10 | 87.82 | 67.95 | 70.88 | 74.30 | 75.12 | 76.46 |
| Normal (w/o BN) | 88.81 | 90.71 | 91.59 | 91.64 | 91.57 | 54.33 | 66.36 | 69.35 | 69.94 | 74.10 |
| Normal (with BN) | 88.52 | 91.74 | 91.20 | 92.29 | 92.40 | 51.53 | 66.90 | 73.18 | 73.89 | 73.02 |
| | | | Training (s) | | | | | Inference (s) | | |
| **Ortho. (SC-Fac)** | 145.3 | 173.4 | 250.1 | 323.1 | 434.0 | 1.47 | 4.35 | 9.72 | 12.65 | 17.56 |
| Normal (w/o BN) | 13.77 | 30.71 | 69.35 | 99.94 | 153.5 | 1.01 | 3.03 | 6.77 | 9.12 | 13.03 |
| Normal (with BN) | 16.56 | 34.16 | 87.49 | 106.9 | 167.4 | 1.14 | 3.34 | 7.33 | 9.69 | 13.39 |

Table 9: **Practical robustness against $\ell_\infty$ adversarial examples** (WideResNet kernel size 5, $\ell_\infty$ perturbation radius of $\epsilon = 8/255$). BCOP+ and SOC (Singla & Feizi, 2021) results with ResNet-18 are reported by Singla & Feizi (2021).

| Model | Method | Clean (%) | PGD (%) |
|---|---|---|---|
| Resnet-18 | BCOP+ | 79.26 | 34.85 |
| | SOC | 82.24 | 43.73 |
| WideResNet22-max | BCOP | 77.57 | 46.35 |
| | Cayley | 78.27 | 45.21 |
| | Ours | 76.28 | 46.27 |

we provide both results for normal convolutional networks with or without batch normalization. In the table, we report the clean/robust accuracy, train time for epoch, and inference time for the test set.

**Robustness against $\ell_\infty$ attacks using adversarial training.** Since orthogonality only guarantees $\ell_2$ Lipschitzness, Lipschitz networks with orthogonal layers are not naturally robust to $\ell_\infty$ perturbations. To further guard Lipschitz networks against $\ell_\infty$ attacks, we follow the approach of adversarial training in Wong et al. (2020). For training, we use a FGSM variant with step size $10/255$; for evaluation, we use 50 PGD iterations with step size $2/255$ and 10 random restarts. We report the experimental results in Table 9. We observe that different orthogonal convolution methods achieve similar $\ell_\infty$ robustness on WideResNet-22. Furthermore, the Lipschitz networks with WideResNet22 architecture is consistently better than ResNet-18, which is previously used in Singla & Feizi (2021).

### E.3    ON THE NECESSITY OF EXACT ORTHOGONALITY

Due to the benefits like generalizability and robustness, achieving exact orthogonality in convolutions is the primary goal of a current research line (Sedghi et al., 2019; Li et al., 2019b; Trockman & Kolter, 2021; Singla & Feizi, 2021). However, until our work, no previous approach achieves orthogonality up to machine precision. Therefore, our proposed method serves as an extreme case in gaining insight into the trade-off between orthogonality and expressiveness.

In Section 6, we have seen that exact orthogonality is not critical in shallow Lipschitz networks for robustness, and various orthogonal convolutions (with different precision) achieve comparable results. However, our method is more favorable in deeper networks (with more than 10 layers) — we show the results in Table 7. It indicates that exact orthogonality is crucial in learning deep Lipschitz networks. However, without our implementation of exact orthogonality (which does not exist before), it is unclear whether exact orthogonality up to machine precision is needed in Lipschitz networks.

Moreover, exact orthogonality is essential for other important and timely applications. For example, reversible networks/normalizing flows (Kingma & Dhariwal, 2018; Van Den Berg et al., 2018) require exact orthogonality to compute inverse transform and determinants accurately.

## F   ORTHOGONAL CONVOLUTIONS FOR RESIDUAL FLOWS

In this section, we first review the class of *flow-based generative models* (Papamakarios et al., 2019; Kobyzev et al., 2020). We focus on *invertible residual network* (Behrmann et al., 2019), a flow-based model that relies on Lipschitz residual block, and its extended version *Residual Flow* (Chen et al., 2019). We then show how to construct improved Residual Flow using our orthogonal convolutions.

**Flow-based models.** Given an observable vector $\boldsymbol{x} \in \mathbb{R}^D$ and a latent vector $\boldsymbol{z} \in \mathbb{R}^D$, we define a bijective mapping $f : \mathbb{R}^D \to \mathbb{R}^D$ from the latent vector $\boldsymbol{z}$ to an observation $\boldsymbol{x} = f(\boldsymbol{z})$. We further define the inverse of $f$ as $F = f^{-1}$, with which we represent the likelihood of $\boldsymbol{x}$ by the one of $\boldsymbol{z}$ as:

$$\ln p_X(\boldsymbol{x}) = \ln p_Z(\boldsymbol{z}) + \ln|\det \boldsymbol{J}_F(\boldsymbol{x})|, \tag{F.1}$$

where $p_X$ is the data distribution, $p_Z$ is the base distribution (usually a normal distribution), and $\boldsymbol{J}_F(x)$ is the Jacobian of $F$ at $\boldsymbol{x}$. In practice, the bijective mapping $f$ is composed by a sequence of $K$ bijective mapping such that $f = f_K \circ \cdots \circ f_1$, where each $f_k$ is named as a *flow*. Since the inverse mapping $F = F_1 \circ \cdots F_K$ transforms the data distribution $p_X$ into a normal distribution $p_Z$, flow-based models are also known as *normalizing flows*. Accordingly, we rewrite Equation (F.1) as:

$$\ln p_X(\boldsymbol{x}) = \ln p_Z(\boldsymbol{z}) + \sum_{k=1}^{K} \ln|\det \boldsymbol{J}_{F_k}(\boldsymbol{x})|, \tag{F.2}$$

In a practical flow-based model, we require efficient computations of **(a)** each bijective mapping $f_k$, **(b)** its inverse mapping $F_k = f_k^{-1}$, and **(c)** the corresponding log-determinant $\ln|\det \boldsymbol{J}_F(\cdot)|$.

**Invertible residual networks (i-ResNets).** Behrmann et al. (2019) proposes a flow-based model based on residual network (ResNet). Note that a block in ResNet is defined as $F(\boldsymbol{x}) = \boldsymbol{x} + g(\boldsymbol{x})$, where $g$ is a convolutional network. In (Behrmann et al., 2019), the authors prove that $F$ is a bijective mapping if $g$ is 1-Lipschitz, and its inverse mapping can be computed by *fixed-point iterations*:

$$\boldsymbol{x}_{k+1} = \boldsymbol{y} - g(\boldsymbol{x}_k), \tag{F.3}$$

where $\boldsymbol{y} = g(\boldsymbol{x})$ is the output of $F$ and the initialization of the iterative algorithm is $\boldsymbol{x}_0 := \boldsymbol{y}$. From the Banach fixed-point theorem, we have

$$\|\boldsymbol{x} - \boldsymbol{x}_k\|_2 = \frac{\mathrm{Lip}(g)^k}{1 - \mathrm{Lip}(g)} \|\boldsymbol{x}_1 - \boldsymbol{x}_0\|, \tag{F.4}$$

i.e., the convergence rate is exponential in the number of iterations and smaller Lipschitz constant will yield faster convergence. Furthermore, the log-determinant can be computed as:

$$\ln p_X(\boldsymbol{x}) = \ln p_Z(\boldsymbol{z}) + \mathrm{tr}\left(\ln\left(\boldsymbol{I} + \boldsymbol{J}_g(\boldsymbol{x})\right)\right) \tag{F.5}$$

$$= \ln p_Z(\boldsymbol{z}) + \mathrm{tr}\left(\sum_{k=1}^{\infty} \frac{(-1)^{k+1}}{k} \left[\boldsymbol{J}_g(\boldsymbol{x})\right]^k\right), \tag{F.6}$$

where the infinite sum is approximated by truncation and the trace is efficiently estimated using the *Hutchinson trace estimator* $\mathrm{tr}(\boldsymbol{A}) = \mathbb{E}_{\boldsymbol{v} \sim \mathcal{N}(\boldsymbol{0}, \boldsymbol{I})}[\boldsymbol{v}^\top \boldsymbol{A} \boldsymbol{v}]$.

To constrain the Lipschitz constant, i-ResNet uses *spectral normalization* on each linear layer in the block. Moreover, to improve optimization stability, i-ResNet changes the activation function from ReLU to ELU, ensuring nonlinear activations have continous derivatives.

As summarized in the conclusion of Behrmann et al. (2019), there are two remaining problems in this model: **(1)** The estimator of the log-determinant is biased and inefficient; **(2)** Designing and learning networks with a Lipschitz constraint are challenging — one needs to constrain each linear layer in the block instead of being able to control the Lipschitz constant of a block.

**Residual Flow.** Chen et al. (2019) address **problem {(1)** by proposing an unbiased *Russian roulette estimator* for Equation (F.6):

$$\mathrm{tr}\left(\ln\left(\boldsymbol{I} + \boldsymbol{J}_g(\boldsymbol{x})\right)\right) = \mathbb{E}_{n,\boldsymbol{v}}\left[\sum_{k=1}^{n} \frac{(-1)^k}{k} \frac{\boldsymbol{v}\left[\boldsymbol{J}g(\boldsymbol{x})^k\right]\boldsymbol{v}}{\mathbb{P}(N \geq k)}\right], \tag{F.7}$$

Table 10: **Comparisons of various flow-based models on the MNIST dataset.** We report the performance in bits per dimension (bpm), where a smaller number indicates a better performance.

| Model | MNIST |
|---|---|
| Glow (Kingma & Dhariwal, 2018) | 1.05 |
| FFJORD (Grathwohl et al., 2018) | 0.99 |
| i-ResNet (Behrmann et al., 2019) | 1.05 |
| Residual Flow (Chen et al., 2019) | 0.97 |
| SC-Fac Residual Flow (Ours) | 0.896 |

where $n \sim p(N)$ and $v \sim \mathcal{N}(0, I)$. Residual Flow further changes the activation function from ELU to LipSwish. The LipSwich activation avoids derivative saturation, which occurs when the second derivative is zero in a large region. However, **problem (2)** remains unresolved.

**Residual flows with orthogonal convolutions.** We propose to address **problem (2)** by replacing the spectral normalized layers by our orthogonal convolutional layers (SC-Fac). Note that orthogonal convolutions directly control the Lipschitz constant of a ResNet block. We keep all other components unchanged — in particular, we use LipSwish activation instead of GroupSort, as GroupSort suffers from derivative saturation. We experiment our model on MNIST dataset. As shown in Table 10, our model substantially improve the performance over the original Residual Flow. We display some images generated by our model in Figure 8.

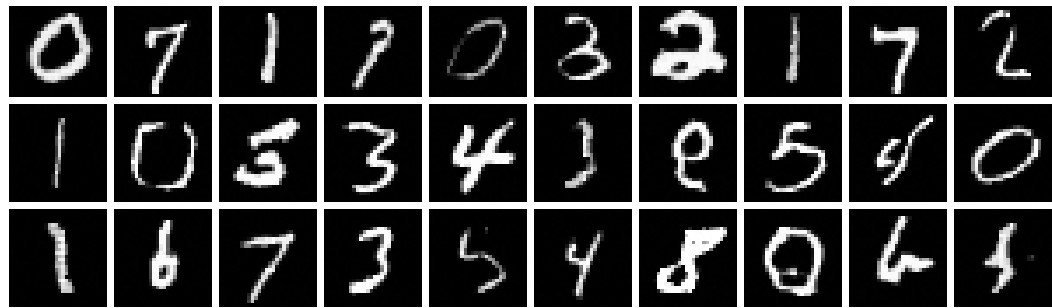

Figure 8: Random samples from SC-Fac Residual Flow trained on MNIST.

