# OpenReview forum: "Scaling-up Diverse Orthogonal Convolutional Networks by a Paraunitary Framework"
_ICLR.cc/2022/Conference — ICLR 2022 Submitted_

### Official Review · Reviewer_EX1P · 2021-10-31

**Correctness:** 4
**Technical Novelty And Significance:** 3
**Empirical Novelty And Significance:** 2
**Recommendation:** 6
**Confidence:** 4

**Main Review:**

This work proposes a new method for orthogonalizing the convolutional layers by exploring the equivalence between spatial orthogonal and spectral paraunitary. The experiments are conducted on various networks including the shallow KW-large networks and slightly deeper WideResNet22. Although the reviewer does not check the submitted code in detail, the code is well-written and clearly commented.
The major concerns are (1) the work seems to have made some overstatement of the contributions, claiming that all the previous work are heuristic, and the proposed approach is systematic with theoretical justification. The reviewer does not quite buy this point, and better explanation on this is needed; (2) the experimental results are not consistently showing the advantages of the proposed method, also the improvement in terms of computational efficiency seems to be marginal.

Below are some more detailed comments.

1. The reviewer found the implementation of the proposed method somewhat hard to follow, it could be better to incorporate an algorithm view of the method to clearly present it.

2. In the paper, the Q matrix is defined as an orthogonal matrix that is randomly initialized and fixed during training. But the reviewer didn’t find the associated implementation in the code (correct me if I missed anything), so the reviewer is wondering how the Q matrix is constructed in the experiments.

3. When demonstrating the results of adversarial robustness, the paper devotes to $\ell_2$ norm based attacks. The reviewer is curious about the results of  $\ell_\infty$ based attacks.

4. The reviewer notices that in the code, when considering striding, the authors include 2 use cases [stride_wide, stride_slim], the reviewer is curious about the actual definition of the different use cases. Besides, the code of the proposed method mentions that the kernel size should be a multiple of stride (in the stride_wide case, this constraint is bypassed by letting kernel_size = kernel_size * stride), the reviewer would appreciate it if this part is presented in more detail. (An algorithm for handling different cases would be nice).

5. This paper is missing quite a few citations on related work:

https://arxiv.org/abs/1810.09102

https://arxiv.org/abs/2103.00673

https://arxiv.org/abs/1911.12207

https://arxiv.org/abs/1905.11926

Please refer and discuss the relationship

**Summary Of The Paper:**

This work proposes a new method for orthogonalizing the convolutional layers by exploring the equivalence between spatial orthogonal and spectral paraunitary. The work then empirically demonstrates the effectiveness of the proposed methods by comparing (1) the Lipschitzness (2) the results of adversarial robustness and (3) the time and memory cost among different methods.


**Summary Of The Review:**

The paper is well-presented overall. However, better positioning of the work, and more convincing experimental results are needed.

---

> ### Author Response · Authors · 2021-11-18
> **Technical Details of our proposed method**
>
> We thank reviewer EX1P for the constructive feedback. We are updating our manuscript according to your suggestions. In a meanwhile, we address reviewer's concerns and questions in the following responses.
>
> > Q1:The work seems to have made some overstatement of the contributions, claiming that all the previous work are heuristic, and the proposed approach is systematic with theoretical justification.
>
> - Our statement in the abstract, "many previous methods are heuristics," mainly refers to the methods that enforce/encourage orthogonality on the flattened kernel [1, 2, 3]. It has been proved that these methods do not lead to orthogonal transformation (i.e., norm preserving). On the other hand, some previous methods guarantee an orthogonal layer, such as BCOP and CayleyConv. Our work studies the spectral properties of orthogonal convolutions, providing a framework that includes all methods (including ours) as special cases. We will clarify this point in the revised version.
>     [1] Jia et al. *Improving training of deep neural networks via singular value bounding.*
>     [2] Cisse et al. *Parseval networks: Improving robustness to adversarial examples.*
>     [3] Bansal et al. *Can We Gain More from Orthogonality Regularizations in Training Deep CNNs?*
>
> >Q2: The experimental results are not consistently showing the advantages of the proposed method, also the improvement in terms of computational efficiency seems to be marginal.
>
> - This is the first work that constructs the exact orghogonal convolution up to machine precision. The primary goal of our experiments is gaining insight into the tradeoff between orthogonality and expressiveness. **The experiments show that our method is beneficial for deeper networks (over 10 layers) compared to the exisitng methods.** The detailed results are shown in Table 7 and the last column in Table 3.
>
> >Q3: The reviewer found the implementation of the proposed method somewhat hard to follow, it could be better to incorporate an algorithm view of the method to clearly present it.
>
> - Thanks for the suggestion! We will add an algorithmic pseudocode to illustrate our method. Following Figure 2, our algorithm consists of three levels: **(1)** Given an oracle for paraunitary systems, we construct orthogonal convolutions according to the hyperparameters of stride, dilation, and groups (Section 3); **(2)** Given an oracle for orthogonal matrices, we construct paraunitary systems according to Equation 2 and Figure 1; **(3)** Lastly, we parameterize orthogonal matrices into unconstrained parameters using matrix exponential.
>
> >Q4: The reviewer is wondering how the Q matrix is constructed in the experiments.
>
> - The computation of $\mathbf{U} = \mathbf{Q} \exp(\mathbf{A})$ is integrated in the Geotorch library [1], which does not expose the underlying parameters $\mathbf{Q}$ and $\mathbf{A}$ to end users.
>     [1] https://github.com/Lezcano/geotorch

---

> > ### Author Response · Authors · 2021-11-18
> > **Technical Details of our proposed method**
> >
> > >Q5: The reviewer is curious about the results of $\ell_\infty$ based attacks.
> >
> > - Thank you for the suggestion! Since orthogonality only guarantees $\ell_2$ Lipschitzness, Lipschitz networks with orthogonal layers are not naturally robust to $\ell_\infty$ perturbations. Therefore, we further use adversarial training [1] to guard these models against $\ell_\infty$ attacks. We are working on these additional experiments that we will add in the revised version.
> >     [1] Wong et al. *Fast is better than free: Revisiting adversarial training.*
> >
> > >Q6: The reviewer notices that in the code, when considering striding, the authors include 2 use cases [stride_wide, stride_slim], the reviewer is curious about the actual definition of the different use cases.
> >
> > - For stride_slim, we choose kernle_size = stride, which mimics the receptive field of a skip connection in an ordinary residual network.
> > - For stride_wide, we choose kernel_size = stride * kernel_size', where kernel_size' is the kernel size for the main branch of the network. Thus, the stride_wide setting mimics the receptive field of a skip connection with average pooling.
> > - Therefore, these two cases bridge the ordinary residual network and our final architecture.
> >
> > >Q7: This paper is missing quite a few citations on related work.
> >
> > - Thanks for pointing out these papers. We want to bring to the reviewer's attention that our paper is significantly different from the papers pointed out. We discuss them in detail below (we cited OCNN [3] in our appendix). We will add these discussions to the revised paper.
> >     - **Orthogonal regularization [1].** The paper proposed first to flatten the 4th-order kernel into a matrix and enforce orthogonality of the flattened matrix with advanced regularization. Unlike [3], even if the regularization term reduces to zero, the layer is not orthogonal.
> >     - **Convolutional normalization [2].** The method orthogonalizes the 3rd-order kernel for each output channel. However, it does not enforce mutual orthogonality for different output channels. Therefore, the holistic 4th-order kernel is not orthogonal, and its Lipschitz constant is only upper bound by the number of output channels.
> >     - **Orthogonal convolutional neural networks [3].** The paper proposed a regularization to encourage orthogonality of the 4th-order kernel --- only when the regularization term reduces to zero, the layer is precisely orthogonal. As a result, the method can not guarantee exact orthogonality, and the approximation depends on the additional hyper-parameter of the regularization coefficient.
> >     - **Network deconvolution [4].** The paper proposed a novel normalization layer that whitens the input feature distribution for each layer. The method focuses on the orthogonality of the data distribution instead of the convolutional layer.
> >
> >     [1] Bansal et al. *Can We Gain More from Orthogonality Regularizations in Training Deep CNNs?*
> >
> >     [2] Liu et al. *Convolutional Normalization: Improving Deep Convolutional Network Robustness and Training.*
> >
> >     [3] Wang et al. *Orthogonal Convolutional Neural Networks.*
> >
> >     [4] Ye et al. *Network Deconvolution.*

---

> > > ### Author Response · Authors · 2021-11-29
> > > **A reminder of the discussion period deadline**
> > >
> > > Dear Reviewer EX1P,
> > >
> > > This is a kind reminder that the discussion period is ending soon. Since we will not be able to reply to you after the discussion period, we want to double-check with you whether we have fully addressed your concerns. We are happy to answer further questions or clarify unclear issues.
> > >
> > > Best,
> > >
> > > Paper 4019 Authors

---

### Official Review · Reviewer_5fbR · 2021-11-01

**Correctness:** 2
**Technical Novelty And Significance:** 2
**Empirical Novelty And Significance:** 2
**Recommendation:** 3
**Confidence:** 5

**Main Review:**

The paper proposes a nice framework which covers different orthonormalization methods for convolutional kernels. In the analyses, the proposed framework performs on par with the state-of-the-art.

However, there are several issues with the paper. In general, first, some of the claims should be revised since they are not verified in the analyses. Second, experimental analyses should be improved with additional analyses on additional datasets and tasks in comparison with the other state-of-the-art methods.

More detailed comments are given as follows:

1. The paper states that “However, many previous approaches are heuristic, and the orthogonality of convolutional layers is not systematically studied.” However, there is a nice literature on orthogonal CNNs, which explores these CNNs from various aspects including generalization and convergence properties of models, in various tasks including image recognition, speech processing and NLP, including adversarial robustness studied in this paper. Then, could you please describe the systematic study referred to and proposed in this paper, which is not covered in the literature?

2. Please explain the statement “There are mature methods that represent orthogonal matrices via unconstrained parameters.” How does this enable to optimize parameters of orthogonal convolutions using standard optimizers instead of optimizers designed for optimization on orthogonal convolutions?

3. There are some issues with the definition and interpretation of orthogonal layers. First, an orthogonal layer is defined according to preservation of norm of input and output. This can be achieved using different types of parameters of convolutions, even with scaled Gaussian parameters. In addition, orthogonal convolutions proposed in the literature satisfy some particular orthogonality properties of matrices of parameters. Second, the orthogonal convolution proposed in this paper is associated with paraunitary property of the transfer matrix.

4. In the experimental analyses, in most of the results, state-of-the-art outperforms the proposed method, while in some results, the proposed method outperforms them. To show a more clear benefit of the proposed method over the state-of-the-art, could you please perform analyses on additional larger scale datasets such as Imagenet? Could you please also compare the proposed method with the methods which employ orthogonal matrices?

5. It is proposed that “related works proposing orthogonalization methods do not guarantee Lipschitzness after training”. However, the proposed orth. conv. employ deep Lipschitz networks to guarantee Lipschitzness. If the proposed orth. conv. does not employ deep Lipschitz networks, then does it  guarantee Lipschitzness?

6. How do you optimize “learnable (column)-orthogonal matrices”?

7. While training models, how do you estimate and optimize h[z], H[z], ortho. Factors, and model params? In the code, Adam is used to optimize parameters. However, it is not clear how orth. factors are also optimized. Do you also optimize them using Adam?

8. How do you apply z-transform on input, kernel and output. For instance, if an input NxN image x is convolved with a 7x7 kernel h, then how do you apply z transform on x and h? That is do you apply path wise or holistically? Also, if x’ is a feature map of size CxWxH, where C is the number of channels, W and H are weight and height of the map, then how do you apply the z-transform on the map?

9. How do you compute ortho. factors efficiently?

10. How do you calculate model parameters A?

11. In the experiments, the proposed methods perform similar to the state-of-the-art. To show superiority of the methods in comparison with the state-of-the-art, additional analyses on larger scale datasets and models should be provided.


**Summary Of The Paper:**

This paper suggests a framework for designing orthogonal convolutions using paraunitary systems. The proposed framework studies orthogonalization of different convolution operations. The proposed methods were examined on several datasets in comparison with state-of-the-art orthogonalization methods.

**Summary Of The Review:**

The proposed framework is nice, and the initial results are promising. However, there are various unclear parts in the paper. In addition, some of the claims are not verified and experimental analyses are limited. Therefore, the paper should be improved with additional analyses and in detail revision for clear acceptance.

---

> ### Author Response · Authors · 2021-11-18
> **Clarification on misunderstandings of our method**
>
> We thank reviewer 5fbR for the detailed questions. We believe most questions are from misunderstandings of our method. We address all questions in the following and add them into the revised version.
>
> > Q1: There is a nice literature on orthogonal CNNs, which explores these CNNs from various aspects including generalization and convergence properties. Then, could you please describe the systematic study referred to and proposed in this paper, which is not covered in the literature?
>
> 1. While the benefits of orthogonality in CNNs are recognized in the literature, methods to construct orthogonal convolutions only appeared recently. Most early papers first flatten the convolution kernels into matrices and then enforce orthogonality therein. However, it has been proved that these methods do not lead to orthogonal transformation (that is, norm-preserving) [1].
> 2. Our work studies the problem of constructing orthogonal convolutions. We establish the equivalence between orthogonal convolutions and paraunitary systems, which leads to three novelties that are not covered in the literature: **(1)** It allows us to design orthogonal convolutions using methods for paraunitary systems (Vaidyanathan, 1996; Lin & Vaidyanathan 1996); **(2)** It provides a framework to orthogonalize variants of convolutions (Section 3); **(3)** It unifies all previous approaches for orthogonal convolutions (Appendix B.3.3).
>     [1] Wang et al. *Orthogonal Convolutional Neural Networks.*
>
> > Q2: Please explain the statement “There are mature methods that represent orthogonal matrices via unconstrained parameters.” How does this enable to optimize parameters of orthogonal convolutions using standard optimizers instead of optimizers designed for optimization on orthogonal convolutions?
>
> 1. We have included a survey on various methods that parameterize orthogonal matrices using unconstrained parameters in Appendix B.4. In general, these approaches represent an orthogonal matrix $\mathbf{U}$ as a function $f$ on unconstrained parameters $\mathbf{A}$, i.e., $\mathbf{U} = f(\mathbf{A})$. We can compute the gradient of $\mathbf{A}$ from the one of $\mathbf{U}$, i.e., $\partial \mathcal{L}/\partial \mathbf{A} = (\partial \mathcal{L}/\partial \mathbf{U}) (\partial \mathbf{U}/\partial \mathbf{A}$), which allows us to use standard optimizers for orthogonal matrices. In our work, we parameterize orthogonal matrices using matrix exponential.
> 2. In Equation (2) and Figure 1, we show how to parameterize orthogonal convolutions using orthogonal matrices. Therefore, we learn an orthogonal convolution by optimizing its underlying orthogonal matrices. In general, parameterization (a.k.a. trivialization) allows us to optimize over constrained objects using standard optimizers, avoiding customized optimizers for each constraint. Moreover, to the best of our knowledge, there is no manifold optimizer for orthogonal convolutions.
>
> >Q3: There are some issues with the definition and interpretation of orthogonal layers. First, an orthogonal layer is defined according to preservation of norm of input and output. This can be achieved using different types of parameters of convolutions, even with scaled Gaussian parameters. In addition, orthogonal convolutions proposed in the literature satisfy some particular orthogonality properties of matrices of parameters. Second, the orthogonal convolution proposed in this paper is associated with paraunitary property of the transfer matrix.
>
> 1. Our definition of orthogonal convolutions is consistent throughout the paper --- we define orthogonality in terms of norm-preservation, and we prove in Theorem A.5 that a convolutional layer is orthogonal **if and only if** the transfer matrix is paraunitary.
> 2. We have a question regarding scaled Gaussian parameters. Could the reviewer explain what it is and provide us with some references that we can look at and cite?
>
> > Q4: To show a more clear benefit of the proposed method over the state-of-the-art, could you please perform analyses on additional larger scale datasets such as Imagenet? Could you please also compare the proposed method with the methods which employ orthogonal matrices?
>
> - To the best of our knowledge, our method is the first that allows scaling of orthogonal convolutions to deep Lipschtiz networks, such as WideResNet-22. Since no existing methods on deep Lipschtiz networks present results on ImageNet, we believe our experiments are sufficient to exhibit the superiority of our approach compared with the state-of-the-art.
> - Regarding comparisons with the methods that employ orthogonal matrices, could you provide a specific reference that we can compare? (We mainly compare with BCOP and CayleyConv in our current experiments).

---

> > ### Author Response · Authors · 2021-11-18
> > **Clarification on misunderstandings of our method**
> >
> > > Q5: The proposed orth. conv. employ deep Lipschitz networks to guarantee Lipschitzness. If the proposed orth. conv. does not employ deep Lipschitz networks, then does it guarantee Lipschitzness?
> >
> > - Quite the opposite, it is deep Lipschitz networks that employ orthogonal convolutions to guarantee Lipschitzness --- The aspiration to build deep Lipschitz networks activates the recent research of orthogonal convolutions. While an orthogonal convolution is always Lipschitz by itself, a network with orthogonal convolutions may not be Lipshitz if it contains other components that are not Lipschitz, such as batch normalization.
> >
> > >Q6: How do you optimize “learnable (column)-orthogonal matrices”?
> >
> > - In the response to Q2 above, we discuss how to optimize orthogonal matrices. To learn a column-orthogonal matrix, we can learn a full orthogonal matrix but only return its first few columns.
> >
> > >Q7: While training models, how do you estimate and optimize h[z], H[z], ortho. Factors, and model params? In the code, Adam is used to optimize parameters. However, it is not clear how orth. factors are also optimized. Do you also optimize them using Adam?
> >
> > - Recall that our method uses orthogonal matrices to parameterize orthogonal convolutions, and the orthogonal matrices are in turn represented by unconstrained parameters using matrix exponential. Therefore, we use Adam to optimize the unconstrained parameters --- the orthogonal matrices and convolutions are learned accordingly. Note that we use the z-transform H[z] for analysis, which is not explicitly computed in the implementation. We will add more discussion of the role of H[z] in Section 2.
> >
> > >Q8: How do you apply z-transform on input, kernel and output. For instance, if an input NxN image x is convolved with a 7x7 kernel h, then how do you apply z transform on x and h? That is do you apply path wise or holistically? Also, if x’ is a feature map of size CxWxH, where C is the number of channels, W and H are weight and height of the map, then how do you apply the z-transform on the map?
> >
> > - For simplicity, we analyze 1D convolutional layers in the paper. To deal with 2D filters and features, we can easily extend our analysis using 2D z-transform.
> >     - Given a kernel $\mathbf{h}$ of size $\mathbb{R}^{C \times C \times K \times K}$, we denote each entry as $h_{ts}[i,j]$, where $(t,s)$ index output/input channels and $(i,j)$ index filter locations. Then we group all entries from each location $(i,j)$ into a matrix $\mathbf{h}[i, j] \in \mathbb{R}^{C \times C}$, and we compute the 2D z-transform holistically $\mathbf{H}(z_1, z_2) = \sum_{i = -(K-1)/2}^{(K - 1)/2} \sum_{j = -(K-1)/2}^{(K-1)/2} \mathbf{h}[i, j] z_1^{-i} z_2^{-j}$.
> >     - Similarly, given an input map $\mathbf{x}$ (or an output map $\mathbf{y}$) of size $\mathbb{R}^{C \times H \times W}$, we denote each entry as $x_s[i, j]$ (or $y_t[i, j]$) and group all entries from each location $(i, j)$ as a vector $\mathbf{x}[i, j] \in \mathbb{R}^{C}$ (or $\mathbf{y}[i, j] \in \mathbb{R}^{C}$). Then we compute its z-transform as $\mathbf{X}(z_1, z_2) = \sum_{i = 0}^{H - 1} \sum_{j = 0}^{W - 1} \mathbf{x}[i, j] z_1^{-i} z_2^{-j}$ (or $\mathbf{Y}(z_1, z_2) = \sum_{i = 0}^{H - 1} \sum_{j = 0}^{W - 1} \mathbf{y}[i, j] z_1^{-i} z_2^{-j}$).
> >     - The convolution theorem states that $\mathbf{Y}(z_1, z_2) = \mathbf{H}(z_1, z_2) \mathbf{X}(z_1, z_2)$.
> >
> > >Q9: How do you compute ortho. factors efficiently?
> >
> > - To compute orthogonal matrices efficiently, we use the Geotorch library for constrained optimization [1], which itself relies on the scale-squaring method to calculate the matrix exponential to the machine precision [2].
> >     [1] https://github.com/Lezcano/geotorch
> >     [2] Higman. *The scaling and squaring method for the matrix exponential revisited.*
> >
> > >Q10: How do you calculate model parameters A?
> >
> > - With recursive parameterization (Figure 2), we reduce the problem of learning orthogonal convolutions to the one of learning skew-symmetric matrices. In the figure, we omit the last step, where each skew-symmetric matrix A is determined by its upper triangle entries. Therefore, we can use backpropagation and standard optimizer to learn these entries.
> >
> > >Q11: To show superiority of the methods in comparison with the state-of-the-art, additional analyses on larger scale datasets and models should be provided.
> >
> > - To the best of our knowledge, our method is the first that allows scaling of orthogonal convolutions to deep Lipschitz networks, such as WideResNet-22. Furthermore, previous approaches do not support variants of convolutions, such as strided and dilated convolutions. Therefore, one focus of our paper is to deal with these convolution variants. Since no existing methods on orthogonal convolutions present results on ImageNet, we believe our experiments are sufficient to exhibit the superiority of our approach compared with the state-of-the-art.

---

> > > ### Comment · Reviewer_5fbR · 2021-11-22
> > > **Thank you for the response.**
> > >
> > > Thank you very much for the detailed explanations.
> > > - For the related work, you can check the following:
> > > https://openaccess.thecvf.com/content_ICCV_2019/papers/Roy_Siamese_Networks_The_Tale_of_Two_Manifolds_ICCV_2019_paper.pdf
> > > https://www.aaai.org/ocs/index.php/AAAI/AAAI18/paper/viewFile/17072/16695
> > > https://openaccess.thecvf.com/content/CVPR2021/papers/Liu_Orthogonal_Over-Parameterized_Training_CVPR_2021_paper.pdf
> > > https://arxiv.org/pdf/2002.01113.pdf
> > > https://openaccess.thecvf.com/content_CVPR_2020/papers/Huang_Controllable_Orthogonalization_in_Training_DNNs_CVPR_2020_paper.pdf
> > > https://openaccess.thecvf.com/content_CVPR_2020/papers/Wang_Orthogonal_Convolutional_Neural_Networks_CVPR_2020_paper.pdf (also orth. conv. on imagenet)
> > > Some additional works were discussed in the following paper:
> > > https://arxiv.org/pdf/2009.12836.pdf
> > > The proposed methods should be compared with at least some of these optimization methods on orthogonal kernels.
> > >
> > > - I implemented some tests using your code with ResNet9 by only changing the parameter --conv and keeping the other params default, and the results are as follows:
> > >
> > > Paraunitary
> > >
> > > [ResNet9] --- Empirical Lipschitzity: 0.6180931925773621
> > > [ResNet9] Epoch: 199 | Train Acc: 0.6693, Test Acc: 0.6759, Time: 59.9, lr: 0.000000
> > > [ResNet9] (PROVABLE) Certifiably Robust (eps: 0.1412): 0.4877, Cert. Wrong: 0.1024, Insc. Right: 0.1882, Insc. Wrong: 0.2217
> > > [ResNet9] (EMPIRICAL) Robust accuracy (eps: 0.1412): 0.6226
> > >
> > > CayleyConv
> > >
> > > [ResNet9] --- Empirical Lipschitzity: 0.631869375705719
> > > [ResNet9] Epoch: 199 | Train Acc: 0.7363, Test Acc: 0.7207, Time: 36.8, lr: 0.000000
> > > [ResNet9] (PROVABLE) Certifiably Robust (eps: 0.1412): 0.5367, Cert. Wrong: 0.0935, Insc. Right: 0.1840, Insc. Wrong: 0.1858
> > > [ResNet9] (EMPIRICAL) Robust accuracy (eps: 0.1412): 0.6659
> > >
> > > BCOP
> > >
> > > [ResNet9] --- Empirical Lipschitzity: 0.6167868971824646
> > > [ResNet9] Epoch: 199 | Train Acc: 0.7135, Test Acc: 0.7050, Time: 46.4, lr: 0.000000
> > > [ResNet9] (PROVABLE) Certifiably Robust (eps: 0.1412): 0.5179, Cert. Wrong: 0.0969, Insc. Right: 0.1873, Insc. Wrong: 0.1979
> > > [ResNet9] (EMPIRICAL) Robust accuracy (eps: 0.1412): 0.6507
> > >
> > >
> > > RKO
> > >
> > > [ResNet9] --- Empirical Lipschitzity: 0.5633834004402161
> > > [ResNet9] Epoch: 199 | Train Acc: 0.7077, Test Acc: 0.6901, Time: 36.4, lr: 0.000000
> > > [ResNet9] (PROVABLE) Certifiably Robust (eps: 0.1412): 0.4924, Cert. Wrong: 0.0992, Insc. Right: 0.1976, Insc. Wrong: 0.2108
> > > [ResNet9] (EMPIRICAL) Robust accuracy (eps: 0.1412): 0.6443
> > >
> > > These results show that:
> > > - Paraunitary underperforms all these competitors (in terms of acc.).
> > > - There is not a direct relationship between Empirical Lipschitzity and train/test accuracy.
> > >
> > > Could you please further elaborate these results?
> > >
> > > Thank you.

---

> > > > ### Author Response · Authors · 2021-11-23
> > > > **Discussion of additional references and clarification of experimental setup**
> > > >
> > > > ### Discussion of additional references
> > > >
> > > > Thank you for providing the references. After carefully checking them, we think many of these works are not directly related to our proposed method. We clarify the details below:
> > > >
> > > > - **Orthogonal Convolutional Neural Networks (OCNN) [6].** We already added this reference in our paper (in Sections 1 and 5).
> > > >     - This paper proposes a regularization method for orthogonal convolutions, which does not lead to strict orthogonality  (Lipschitzness). Therefore, it is not suitable for applications such as adversarial robustness --- the robustness accuracy drops to 5% after training, unlike ours and comparing methods in our paper.
> > > >     - The reviewer pointed out that this paper includes the experiment on ImageNet but **their adversarial robustness experiment is only on CIFAR-100** (Figure 8 and Section 4.7 in this paper).
> > > >
> > > > - **Learning Orthogonality on Matrices [2, 4, 5] rather than on Convolutions**. These three papers aim to learn orthogonal matrices, where [2, 5] are based on parameterization using algorithmic unrolling and [4] uses Riemannian gradient descent (RGD).
> > > >     - For the convolutional layer, these papers enforce orthogonality on the flattened kernel. However, as shown in OCNN [6], flattened kernel orthogonality does not lead to the orthogonality of the convolution. Moreover, to the best of our knowledge, there is no algorithm unrolling or RGD method for orthogonal convolution due to its complicated structure.
> > > >     - Since our proposed method constructs orthogonal convolutions using orthogonal matrices, these methods can also be used in our implementation. We leave this part as future work. We added these references in our revised version (related works, Section 5).
> > > >
> > > > - **[1], [3], and [7] are directly related to orthogonal networks.**
> > > >     - **Siamese Networks [1].** The paper provides a method to update the factorization $M=L L^\top$, which has invariance to the orthogonal group. The purpose and result of this paper are significantly different from the goal of our paper.
> > > >     - **Over-Parameteried Training (OPT) [3].** This paper does not learn orthogonal weights/kernels in convolutional neural networks. Instead, OPT learns an orthogonal transformation for each layer in the neural network while keeping the randomly initialized neuron weights fixed.
> > > >     - **Normalization Techniques [7].** The operations in normalization layers (such as batch normalization) are not orthogonal (nor Lipschitz). Therefore, all Lipschitz networks do not use the normalization layer. Some previous models like OCNN [6] keep normalization layers in their architecture; as a result, they still suffer from adversarial attacks.
> > > >
> > > > [1] Roy et al. *Siamese Networks: The Tale of Two Manifolds.*
> > > >
> > > > [2] Huang *Orthogonal Weight Normalization: Solution to Optimization over Multiple Dependent Stiefel Manifolds in Deep Neural Networks.*
> > > >
> > > > [3] Liu et al. *Orthogonal Over-Parameterized Training.*
> > > >
> > > > [4] Li et al. *Efficient Riemannian Optimization on the Stiefel Manifold via the Cayley Transform.*
> > > >
> > > > [5] Huang et al. *Controllable Orthogonalization in Training DNNs.*
> > > >
> > > > [6] Wang et al. *Orthogonal Convolutional Neural Networks.*
> > > >
> > > > [7] Huang et al. *Normalization Techniques in Training DNNs: Methodology, Analysis, and Application.*
> > > >
> > > > ### Clarification of our experimental setup
> > > >
> > > > Thank you for trying out our code and providing us with the comparisons. However, there are two major misunderstandings.
> > > > - **Model performance highly depends on the training strategies.** Following (Trockman & Kolter, 2021), we performed a hyper-parameter search for the Lipschitz margin $\epsilon_0$ of the loss function for each model. In Figure 7 (Appendix E), we show the effect of Lipschitz margin $\epsilon_0$ for WideResNet22-10. We believe that the performance gap between these results and the ones we reported comes from this parameter.
> > > > - **Our method has better performance for deeper architectures.** Our proposed method performs better for networks with more than ten layers. We compared shallow vs. deeper networks in Table 3 and compared different depth and widths (especially with 22 and 34 layers) in Table 7. We discuss the relationship between our exact orthogonal convolutions and expressiveness in Appendix E.3.'
> > > >
> > > > **Regarding the relationship between empirical Lipschitz and training/test accuracy.** We also found no direct relationship between empirical Lipschitz/robust accuracy and training/test accuracy for all models. Our experiments focus on obtaining the best robust accuracy with the best possible test accuracy (close to the standard model).

---

> > > > > ### Author Response · Authors · 2021-11-29
> > > > > **A reminder of the discussion period deadline**
> > > > >
> > > > > Dear Reviewer 5fbR,
> > > > >
> > > > > This is a kind reminder that the discussion period is ending soon. Since we will not be able to reply to you after the discussion period, we want to double-check with you whether we have fully addressed your concerns. We are happy to answer further questions or clarify unclear issues.
> > > > >
> > > > > Best,
> > > > >
> > > > > Paper 4019 Authors

---

> > > > > > ### Comment · Reviewer_5fbR · 2021-11-29
> > > > > > **Additional comments to improve the paper.**
> > > > > >
> > > > > > Dear Authors,
> > > > > >
> > > > > > Thank you for the response and the reminder.
> > > > > >
> > > > > > Regarding experiments: I further performed additional analyses using your code with some hyperparameter search, while the accuracy gap is still high (even for clean data). If you performed hyperparameter search, then could you please provide the optimal hyperparameters used in the experiments as well? Also, how does Lipschitzness change as you train models using optimal hyperparameters (I did not see much difference)?
> > > > > >
> > > > > > Regarding Lipschitzness: As you mentioned, a relationship between empirical Lipschitzness and training/test accuracy could not be obtained in the analyses. However, existence of this relationship is the main motivation for your proposed Lipschitz networks. Then, there seems to be an inconsistency between the motivation and experimental results.
> > > > > >
> > > > > > Regarding the related work: In the paper, it is stated that “existing architectures for Lipschitz networks remain simple and shallow, and a Lipschitz network is typically an interleaving cascade of orthogonal layers and GroupSort activations.” In weight normalization methods, weight matrices can be identified on Stiefel manifolds. Therefore, the claim “Therefore, all Lipschitz networks do not use the normalization layer.” does not sound correct. In addition, the proposed methods need to be compared with some these orthogonal networks in more detailed large scale analyses (see the following first claim).
> > > > > >
> > > > > > Regarding the overall motivation and claims: I checked all the discussions and results/claims in the paper. There are several related but slightly different claims proposed in the paper. First, as stated in the title, a major claim is “scaling-up diverse orthogonal CNNs”, which is mainly considered in the algorithmic/method part. In this case, the proposed methods should be compared with the related works mentioned by the reviewers. Another claim is the improvement of the robustness of the networks against adversarial attacks using a variation of Lipschitz networks, which is mainly examined in the experimental analyses. In this case, there seems to be an inconsistency between some of the experimental results and claims. I recommend authors to elucidate both of these claims in detail to improve the paper. Some simple analyses using larger models on larger datasets such as ImageNet, in comparison with the related work can show superiority of the proposed methods more clearly.

---

> > > > > > > ### Author Response · Authors · 2021-11-30
> > > > > > > **Additional clarifications**
> > > > > > >
> > > > > > > **Experimental setup.** Following other state-of-the-art approaches, we searched for hyperparameters, such as the loss margin $\epsilon_0$ and kernel size, and reported the best models for all. **1) The loss margin $\epsilon_0$** with ResNet9 is $\epsilon_0 = 0.2$  for our model and $\epsilon_0 = 0.1$ for CayleyConv, **2)** The best **kernel size** we found is $5$ which is used for all experiments. **3)** Following (Trockman & Kolter, 2021), the **inputs are standardized** except for KW-Large (Table 3, top). **4)** To eliminate fluctuations by seeds, we report the **best performance among five seeds** for all models. To reproduce our ResNet9 experiments (Table 3, bottom left) using our code, below are the commands with the correct hyperparameters:
> > > > > > >
> > > > > > >     python3 train.py --model ResNet --layers 8 --factor 10 --kernel 5 --linear CayleyLinear --conv Paraunitary --stddev --eps_0 0.2
> > > > > > >     python3 train.py --model ResNet --layers 8 --factor 10 --kernel 5 --linear CayleyLinear --conv CayleyConv --stddev --eps_0 0.1
> > > > > > >     python3 train.py --model ResNet --layers 8 --factor 10 --kernel 5 --linear CayleyLinear --conv RKO --stddev --eps_0 0.01
> > > > > > >     python3 train.py --model ResNet --layers 8 --factor 10 --kernel 5 --linear BjorckLinear --conv BCOP --stddev --eps_0 0.2
> > > > > > >
> > > > > > > **Empirical Lipschizness.** In theory, the Lipschitz constant is closely related to robust accuracy, as explained in **Lipschitz networks** (Section 4). Since computing the exact Lipschitz constant is NP-hard, the recent method relies on empirical estimation [1]. However, the estimation may not be accurate enough to determine the robustness alone. BCOP (Li et al., 2019) and CayleyConv (Trockman & Kolter, 2021) show that the estimated Lipschitzness is not directly related to clean/robust accuracy. Therefore, we also do not use this metric in the paper.
> > > > > > >
> > > > > > > [1] Scaman & Virmaux. *Lipschitz regularity of deep neural networks: analysis and efficient estimation.*
> > > > > > >
> > > > > > > **Normalization layers.** We would like to point out that *weight normalization* is not the normalization layer we talk about --- weight normalization applies **regularization** on the weights instead of **normalization** on the features (like *layer normalization* or *batch normalization*). To the best of our knowledge, there is no normalization layer on the features that is Lipschitz.
> > > > > > >
> > > > > > > **Motivations for orthogonal convolutions.** While our work provides a framework for general orthogonal convolutions, we focus on deep Lipschitz networks for adversarial robustness in our experiments. We show how our framework can be used to scale-up deep Lipschitz networks, and exact orthogonality affects the performance compared to other orthogonal networks. In the third paragraph of the introduction, we explain why other methods for orthogonal convolutions, such as regularization and initialization, are not suitable for this scenario. As shown in Appendix F, our framework can also be used for other scenarios, such as the invertible residual network.
> > > > > > >
> > > > > > > We hope that our clarifications address your concerns. We would very much appreciate it if you could consider increasing the review score of our paper.

---

### Official Review · Reviewer_yYMs · 2021-11-02

**Correctness:** 3
**Technical Novelty And Significance:** 3
**Empirical Novelty And Significance:** 3
**Recommendation:** 6
**Confidence:** 4

**Main Review:**

**Strengths**
- The proposed method is theoretically grounded and relatively efficient in computations.
- The analysis of strided, dilated convolution layers is inspiring.
- Numerical evidence on orthogonality evaluation of different designs for standard convolution shows that exact orthogonality is achieved.

**Weaknesses**
- It is nice to see that exact orthogonality is achieved however it remains unclear exact orthogonality is actually helpful or needed. For example, from Table 2, the proposed SC-Fac achieves the most "accurate" orthogonality result in worse performance in both certified and practical robustness. Even though the author claims that the method achieves "comparable" results with other baseline methods, the results are consistently worse than the baselines.

- The authors can compare their core idea with related work that is more heuristic, such as [1] which also considers achieving orthogonality in the spectral domain, as well as [2],[3].

[1] Liu et al. Convolutional Normalization: Improving Deep Convolutional Network Robustness and Training

[2] Wang et al. Orthogonal Convolutional Neural Networks

[3] Bansai et al. Can We Gain More from Orthogonality Regularizations in Training Deep CNNs?

- Even though the method is more computationally efficient, it is only compared with methods such as Cayley which is known to be computationally heavy. The method is still much computationally heavier than the original networks. It would be nice to have an extra line in Figure 4 showing the train time of the ordinary network.

**Summary Of The Paper:**

This paper proposed a theoretical framework for orthogonal convolutional layers based on the equivalency of orthogonal convolution in the spatial domain and the paraunitary systems in the spectral domain. The proposed method parametrizes the orthogonal convolution layer as compositions of multiple convolutions in the spatial domain, resulting in exact orthogonality. The layers are also more memory and computationally efficient than most previous methods.

**Summary Of The Review:**

The proposed method achieves exact orthogonal convolutional layers through re-parametrization. The method is theoretically grounded and easy to understand. Numerical proofs are provided to show exact orthogonality is achieved by composing a sequence of learnable convolutions.

---

> ### Author Response · Authors · 2021-11-18
> **Comparison with previous methods and ordinary network**
>
> We thank reviewer yYMs for the positive feedback. Our response to your comments are as follows:
>
> > Q1: It remains unclear exact orthogonality is actually helpful or needed.
>
> 1. Due to the benefits like generalizability and robustness, achieving exact orthogonality in convolutions is the primary goal of a current research line (Sedghi et al., 2018; Li et al., 2019; Trockman & Kolter, 2021). However, until our work, no previous approach achieves orthogonality up to machine precision. Therefore, our proposed method serves as an extreme case in gaining insight into the tradeoff between orthogonality and expressiveness.
> 2. In Section 6.2 (Adversarial Robustness), we discussed that exact orthogonality is not critical in shallow Lipschitz networks for robustness, and various orthogonal convolutions (with different precision) achieve comparable results. However, our method is more favorable than other ones in deeper networks (> 10 layers). We show the results in Table 7 and the last column in Table 3. It indicates that exact orthogonality is necessary. However, without our implementation of exact orthogonality (which does not exist before), it is unclear whether exact orthogonality up to machine precision is crucial in shallow or deep Lipschitz networks.
> 3. Moreover, exact orthogonality is essential for other important and timely applications. For example, reversible networks/normalizing flows [a,b] require exact orthogonality to compute inverse transform and determinants accurately.
>     [a] Kingma, Diederik P., and Prafulla Dhariwal. "Glow: generative flow with invertible 1× 1 convolutions." Proceedings of the 32nd International Conference on Neural Information Processing Systems. 2018
>     [b] Berg, Rianne van den, et al. "Sylvester normalizing flows for variational inference." arXiv preprint arXiv:1803.05649 (2018).
>
> > Q2: The authors can compare their core idea with related work that is more heuristic, such as [1] which also considers achieving orthogonality in the spectral domain, as well as [2],[3].
>
> - Thanks for pointing out these papers. We want to bring to the reviewer's attention that our paper is significantly different from the papers pointed out, and we discuss them in detail below (we cite OCNN [2] in our appendix). We will add the discussion to the revised manuscript.
>     - **Convolutional normalization (ConvNorm) [1]**. This method orthogonalizes the 3rd-order kernel for each output channel. However, it does not enforce the mutual orthogonality of kernels for different output channels. Therefore, the holistic 4th-order kernel is not orthogonal, and its Lipschitz constant is only upper bound by the number of output channels.
>     - **Orthogonal convolutional neural network (OCNN) [2]**. This paper proposed a regularization to encourage orthogonality of the 4th-order kernel --- only when the regularization term reduces to zero, the layer is exactly orthogonal. As a result, the method can not guarantee exact orthogonality, and the approximation depends on the additional hyper-parameter of the regularization coefficient.
>     - **Orthogonal regularization [3]**. This paper proposed first flattening the 4th-order kernel into a matrix and encouraging orthogonality of the flattened matrix using regularization. Unlike [2], even if the regularization term reduces to zero, the layer is not orthogonal.
>
>     [1] Liu et al. *Convolutional Normalization: Improving Deep Convolutional Network Robustness and Training.*
>
>     [2] Wang et al. *Orthogonal Convolutional Neural Network.*
>
>     [3] Bansai et al. *Can We Gain More from Orthogonality Regularizations in Training Deep CNNs?*
>
> > Q3: It would be nice to have an extra line in Figure 4 showing the train time of the ordinary network.
>
> - Thanks for the suggestion. There is a training and inference time comparison between ours and the ordinary network in Table 8 (Appendix D). We will add them in Figure 4 in the revision.

---

> ### Author Response · Authors · 2021-11-29
> **A reminder of the discussion period deadline**
>
> Dear Reviewer yYMs,
>
> This is a kind reminder that the discussion period is ending soon. Since we will not be able to reply to you after the discussion period, we want to double-check with you whether we have fully addressed your concerns. We are happy to answer further questions or clarify unclear issues.
>
> Best,
>
> Paper 4019 Authors

---

### Official Review · Reviewer_rQN6 · 2021-11-03

**Correctness:** 4
**Technical Novelty And Significance:** 4
**Empirical Novelty And Significance:** 4
**Recommendation:** 8
**Confidence:** 4

**Main Review:**

Orthogonality is an important problem in the design of neural network architecture that relates to many fundamental properties of the network such as trainability, generalizability and robustness. While the study of orthogonality in fully connected layers (or convolution layers with 4D kernels treated as 2D matrices) has a long history, it is only until very recent (in the past 2-3 years) that work on orthogonality of *convolution* layers emerges. This paper provides a solid study in this area by providing a method of enforcing orthogonality in convolutions, revealing its technical connections with previous methods, designing a deep residual Lipschitz network architectures and conducting solid experiments. I find the presentation to be mostly clear and easy to follow, though I feel that there is a tendency of overclaiming the contribution in the abstract & intro, see below.

- Complete parameterization of orthogonal convolution. The paper claims that it offers a complete parameterization of orthogonal convolution, but this is not really the case. As stated in Sec. 2, it only offers complete design for *separable* orthogonal 2D convolutions. This puts the technique in an unfavorable position compared to previous methods that does not require separability (e.g. Trockman & Kolter).

- "Orthogonal Networks". The paper frequently use the phrase "orthogonal networks", but it is not clear what that term entails. For example, it is claimed that "Our versatile framework, for the first time, enables the study of architecture designs for deep orthogonal networks", which seems an overclaim since orthogonality in neural networks have already been extensively studied before. In addition, if "orthogonal network" means that the entire network is an orthogonal transformation, then this is kind of a useless network since orthogonality implies linearity (as long as it is surjective). If it means approximately orthogonal then it should consider, in addition to the convolutional layers, the effect of the nonlinear layers - right now there is no discussion of whether the GroupSort that is used as the nonlinear layer is approximately orthogonal or not.


**Summary Of The Paper:**

This paper presents a method for enforcing strict orthogonality of convolutional layers, by means of a factorization in the spectral domain. It shows that the technique can be extended to dealing with practical convolutions with strides, dilations and groups. The experiments demonstrated the superior performance in terms of adversarial robustness.

**Summary Of The Review:**

Solid work on orthogonality of convolution, though there seem to be some overclaiming / imprecise statements in the intro/abstract that may be misleading.

---

> ### Author Response · Authors · 2021-11-18
> **Clarification of imprecise and ambiguous concepts**
>
> We thank reviewer rQN5 for the positive feedback and valuable suggestions. We are revising the manuscript based on your advice and will upload the updated version soon. In the meanwhile, we'd like to address all concerns in the following.
>
> > Q1: The paper claims that it offers a complete parameterization of orthogonal convolution, but this is not really the case. As stated in Sec. 2, it only offers complete design for separable orthogonal 2D convolutions. This puts the technique in an unfavorable position compared to previous methods that does not require separability (e.g. Trockman & Kolter).
>
> 1. Thank you for pointing it out. Our design is complete for 1D orthogonal convolutions and relies on separability to construct multi-dimensional (MD) orthogonal convolutions. However, the separability does not necessarily make our method less expressive compared to CayleyConv (Trockman & Kolter) --- since CayleyConv is also incomplete for MD orthogonal convolutions), and there are orthogonal convolutions realized by our method that are not realizable by CayleyConv.
> 2. Also, methods such as CayleyCov are often memory heavy due to Fourier transforms and matrix inversions on all pixels. Therefore, our method is favorable for deeper and wider models. We will clarify this point in the updated version.
>
> > Q2: The paper frequently use the phrase "orthogonal networks", but it is not clear what that term entails.
> > If it means approximately orthogonal then it should consider, in addition to the convolutional layers, the effect of the nonlinear layers - right now there is no discussion of whether the GroupSort that is used as the nonlinear layer is approximately orthogonal or not.
>
> - By "orthogonal networks," we mean deep Lipschitz networks that are approximately orthogonal. For example, in Section 4, we write, "a Lipschitz network is typically composed of orthogonal layers and GroupSort activations." Therefore, a Lipschitz network is approximately orthogonal --- it is orthogonal at each linear region. We will clarify the definition in the revision.

---

> > ### Comment · Reviewer_rQN6 · 2021-11-19
> > **"Orthogonal Network"**
> >
> > > "a Lipschitz network is typically composed of orthogonal layers and GroupSort activations." Therefore, a Lipschitz network is approximately orthogonal.
> >
> > I don't see how the statement is valid. GroupSort has the property of preserving gradient norm in backward pass but does not preserve the norm in the forward pass in general. So, there is no any guarantee that such a Lipschitz network is orthogonal or in any sense "approximately" orthogonal. In fact, designing orthogonal or approximately orthogonal network is a separate topic of interest as in the reference Pennington '17, Qi '20 etc where the design of nonlinearity is a factor to be carefully considered.

---

> > > ### Author Response · Authors · 2021-11-20
> > > **Clarification of GroupSort and Orthogonality**
> > >
> > > Let us clarify the GroupSort activation more in detail.
> > >
> > > - **GroupSort not only perserves the gradient norm in the backward pass but also the norm in the forward pass.** Since GroupSort is a sorting function, input and output always have the same $\ell_2$ norm in general. GroupSort with the group size of $2$ is a spacial case that we use in the paper followed by [1, 2]. Given a length-2 vector $\mathbf{x} = (x_1, x_2)$ in a window, the output $\mathbf{y} = \textrm{GroupSort}(\mathbf{x})$ either returns $(x_1, x_2)$ or $(x_2, x_1)$ --- in both cases, $\|\mathbf{y}\| = \|\mathbf{x}\| = \sqrt{x_1^2 + x_2^2}$, i.e., the input and output norms are equal. We will clarify this detail in the revised version.
> > >
> > >     [1] Chernodub & Nowicki. *Norm-preserving orthogonal permutation linear unit activation functions (oplu).*
> > >
> > >     [2] Anil et al. * Sorting out lipschitz function approximation.*
> > >
> > > - **By "approximate orthogonal," we mean a Lipschitz network with orthogonal layers and GroupSort activations is locally orthogonal**. Given any input $\mathbf{x}$ to the network, there exists a neighborhood $N(x, r)$ with radius $r$ such that the sorting ordering in each GroupSort activation does not change. In this neighborhood $N(x, r)$, each GroupSort layer operates as a constant permutation (thus orthogonal); consequently, the whole network is locally orthogonal.
> > >
> > > - **Comparison with dynamic isometry.** We discuss dynamical isometry [3, 4] in Section 5. These works ensure orthogonality at initialization by orthogonal initialization of linear layers and modification of activations. In contrast, deep Lipschitz networks ensure local orthogonality.
> > >
> > >     [3] Pennington et al. *Resurrecting the sigmoid in deep learning through dynamical isometry: theory and practice.*
> > >
> > >     [4] Qi et al.*Deep isometric learning for visual recognition.*

---

> > > > ### Comment · Reviewer_rQN6 · 2021-11-20
> > > > **Final notes**
> > > >
> > > > Thanks, yes I meant to say GroupSort does not preserve inner product (instead of norm) in forward pass , so that it is not orthogonal. Nonetheless, I find the notion of "locally orthogonal" much more precise than "approximately orthogonal".

---

> > > > > ### Author Response · Authors · 2021-11-22
> > > > > **Answer to the final notes**
> > > > >
> > > > > Thank you for your quick response! We will update the paper according to your suggestion.

---

### Author Response · Authors · 2021-11-23
**Revision of the paper**

Dear reviewers and area chairs,

Thanks to all reviewers for your efforts in reviewing our paper. We have revised our paper according to the valuable suggestions. To make our updated draft more accessible, we have summarized the revisions in the following.
- **[Reviewer rQN6] Clarification of separability in abstract and introduction.** According to the suggestion from Reviewer rQN6, we have clarified that our method is complete *orthogonal 1D-convolutions* and *separable orthogonal 2D-convolutions*. We also emphasize that no existing method achieves completeness for *orthogonal 2D-convolutions*.
- **[Reviewer rQN6] Clarification of orthogonal networks and GroupSort.** We have added a discussion on the GroupSort activation and its relation to orthogonal networks in Appendix D. We discuss two crucial properties of GroupSort --- it preserves the norm in the forward pass and the gradient norm in the backward pass. We also clarify that a Lipschitz network with orthogonal layers and GroupSort activations is locally orthogonal.
- **[Reviewer yYMs] Necessity of exact orthogonality.** We have added a discussion on the necessity of exact orthogonality in Appendix E.3.
- **[Reviewer EX1P] Modification of discussions of previous methods in the abstract.** We modified our discussions of previous methods. For example, we removed our claim that "many previous approaches are heuristic"; instead, we state that enforcing orthogonality on the flattened kernel does not lead to the orthogonality of the convolution. We also removed "the proposed approach is systematic". We instead explained what our approach does (more in details) in the abstract.
- **[Reviewer EX1P] Pseudocode for our proposed method.** We added an algorithm box in Appendix A. The algorithm presents a pseudo code for Figure 2 in Section 2.
- **[Reviewer EX1P] Robustness against $\ell_\infty$ attacks using adversarial training.** We added an experiment on the robustness of the Lipschitz network against $\ell_\infty$ in Table 9 (Appendix E.2). We observe that all orthogonal convolution methods including ours, BCOP, Cayley achieve similar $\ell_\infty$ robustness in deep Lipschitz networks when trained with adversarial examples.
- **[Reviewer rQN6, 5fbR, EX1P] Discussion on additional references.** We added all of the references suggested by reviewers in related works (Section 5).
- **[Reviewer EX1P and yYMs] Clarification on some technical details.** (1) We clarify the definitions of 'stride_wide' and 'stride_slim' in Table 4. (2) We add a line for ordinary convolutional networks in Figure 4.

Best,

Paper 4019 Authors

---

### Decision · Program_Chairs · 2022-01-20

**Decision:**

Reject

**Comment:**

This paper proposes a method for parameterizing orthogonal convolutional layers that derives from paraunitary systems in the spectral domain and performs a comparison with other state-of-the-art orthogonalization methods. The paper argues that the approach is more computationally efficient than most previous methods and that the exact orthogonality is important to ensure robustness in some applications.

The reviewers had diverging opinions about the paper, with most reviewers appreciating the theoretical grounding and empirical analysis, but with some reviewers finding weakness in the clarity, reproducibility, and discussion of prior work. The revisions addressed many, but not all, of the reviewers' criticisms.

One point that was highlighted in the discussion is that the method is restricted to separable convolutions. The authors acknowledged this limitation, justifying the expressivity of the method with a comparison to CayleyConv (Trockman & Kolter) and a suggestion that more expressive parameterizations are not necessarily available in 2D. I am not sure this is entirely accurate. In the discussion of related work, the paper briefly mentions dynamical isometry and the prior work of Xiao et al. 2018, who develop a method for initializing orthogonal convolutional layers. What the current paper fails to recognize is that Algorithm 1 of Xiao et al. 2018 actually provides a method for parameterizing non-separable 2D convolutions: simply represent every orthogonal matrix in that algorithm in a standard way, e.g. via the exponential map. While I think there is certainly value in the connection to paraunitarity systems, it seems to me that the above approach would yield a simpler and more expressive representation, and is at minimum worth discussing.

Overall, between the mixed reviewer opinions and their lingering concerns and the existence of relevant prior art that was not discussed in sufficient depth, I believe this paper is not quite suitable for publication at this time.